# Quickly Finding a Benign Region via Heavy Ball Momentum in Non-Convex Optimization

## Abstract

The Heavy Ball Method (Polyak, 1964), proposed by Polyak over five decades ago, is a first-order method for optimizing continuous functions. While its stochastic counterpart has proven extremely popular in training deep networks, there are almost no known functions where deterministic Heavy Ball is provably faster than the simple and classical gradient descent algorithm in non-convex optimization. The success of Heavy Ball has thus far eluded theoretical understanding. Our goal is to address this gap, and in the present work we identify two non-convex problems where we provably show that the Heavy Ball momentum helps the iterate to enter a benign region that contains a global optimal point faster. We show that Heavy Ball exhibits simple dynamics that clearly reveal the benefit of using a larger value of momentum parameter for the problems. The first of these optimization problems is the phase retrieval problem, which has useful applications in physical science. The second of these optimization problems is the cubic-regularized minimization, a critical subroutine required by Nesterov-Polyak cubic-regularized method (Nesterov & Polyak (2006)) to find second-order stationary points in general smooth non-convex problems.

## 1 Introduction

Poylak's Heavy Ball method (Polyak (1964)) has been very popular in modern non-convex optimization and deep learning, and the stochastic version (a.k.a. SGD with momentum) has become the de facto algorithm for training neural nets. Many empirical results show that the algorithm is better than the standard SGD in deep learning (see e.g. Hoffer et al. (2017); Loshchilov & Hutter (2019); Wilson et al. (2017); Sutskever et al. (2013)), but there are almost no corresponding mathematical results that show a benefit relative to the more standard (stochastic) gradient descent. Despite its popularity, we still have a very poor justification theoretically for its success in non-convex optimization tasks, and Kidambi et al. (2018) were able to establish a negative result, showing that Heavy Ball momentum cannot outperform other methods in certain problems. Furthermore, even for convex problems it appears that strongly convex, smooth, and twice differentiable functions (e.g. strongly convex quadratic functions) are one of just a handful examples for which a provable speedup over standard gradient descent can be shown (e.g (Lessard et al., 2016; Goh, 2017; Ghadimi et al., 2015; Gitman et al., 2019; Loizou & Richtárik, 2017; 2018; Gadat et al., 2016; Scieur & Pedregosa, 2020; Sun et al., 2019; Yang et al., 2018a; Can et al., 2019; Liu et al., 2020; Sebbouh et al., 2020; Flammarion & Bach, 2015)). There are even some negative results when the function is strongly convex but not twice differentiable. That is, Heavy Ball momentum might lead to a divergence in convex optimization (see e.g. (Ghadimi et al., 2015; Lessard et al., 2016)). The algorithm's apparent success in modern non-convex optimization has remained quite mysterious.

In this paper, we identify two non-convex optimization problems for which the use of Heavy Ball method has a provable advantage over vanilla gradient descent. The first problem is phase retrieval. It has some useful applications in physical science such as microscopy or astronomy (see e.g. (Candés et al., 2013), (Fannjiang & Strohmer, 2020), and (Shechtman et al., 2015)). The objective is

$$\min_{w \in \mathbb{R}^d} f(w) := \frac{1}{4n} \sum_{i=1}^{n} \left( (x_i^\top w)^2 - y_i \right)^2, \tag{1}$$

where $x_i \in \mathbb{R}^d$ is the design vector and $y_i = (x_i^\top w_*)^2$ is the label of sample $i$. The goal is to recover $w_*$ up to the sign that is not recoverable (Candés et al., 2013). Under the Gaussian design

setting (i.e. $x_i \sim N(0, I_d)$), it is known that the empirical risk minimizer (1) is $w_*$ or $-w_*$, as long as the number of samples $n$ exceeds the order of the dimension $d$ (see e.g. Bandeira et al. (2014)). Therefore, solving (1) allows one to recover the desired vector $w_* \in \mathbb{R}^d$ up to the sign. Unfortunately the problem is non-convex which limits our ability to efficiently find a minimizer. For this problem, there are many specialized algorithms that aim at achieving a better computational complexity and/or sample complexity to recover $w_*$ modulo the unrecoverable sign (e.g. (Cai et al., 2016; Candés & Li, 2014; Candés et al., 2015; 2013; Chen & Candés, 2017; Duchi & Ruan, 2018; Ma et al., 2017; 2018; Netrapalli et al., 2013; Qu et al., 2017; Tu et al., 2016; Wang et al., 2017a;b; Yang et al., 2018b; Zhang et al., 2017a;b; Zheng & Lafferty, 2015)). Our goal is not about providing a state-of-the-art algorithm for solving (1). Instead, we treat this problem as a starting point of understanding Heavy Ball momentum in non-convex optimization and hope for getting some insights on why Heavy Ball (Algorithm 1 and 2) can be faster than the vanilla gradient descent in non-convex optimization and deep learning in practice. If we want to understand why Heavy Ball momentum leads to acceleration for a complicated non-convex problem, we should first understand it in the simplest possible setting.

We provably show that Heavy Ball recovers the desired vector $w_*$, up to a sign flip, given a random isotropic initialization. Our analysis divides the execution of the algorithm into two stages. In the first stage, the ratio of the projection of the current iterate $w_t$ on $w_*$ to the projection of $w_t$ on the perpendicular component keeps growing, which makes the iterate eventually enter a benign region which is strongly convex, smooth, twice differentiable, and contains a global optimal point. Therefore, in the second stage, Heavy Ball has a linear convergence rate. Furthermore, up to a value, a larger value of the momentum parameter has a faster linear convergence than the vanilla gradient descent in the second stage. Yet, most importantly, we show that Heavy Ball momentum also has an important role in reducing the number of iterations in the first stage, which is when the iterate might be in a non-convex region. We show that the higher the momentum parameter $\beta$, the fewer the iterations spent in the first stage (see also Figure 1). Namely, momentum helps the iterate to enter a benign region faster. Consequently, using a non-zero momentum parameter leads to a speedup over the standard gradient descent ($\beta = 0$). Therefore, our result shows a provable acceleration relative to the vanilla gradient descent, for computing a *global optimal* solution in non-convex optimization.

The second of these is solving a class of cubic-regularized problems,

$$\min_w f(w) := \tfrac{1}{2} w^\top A w + b^\top w + \tfrac{\rho}{3} \|w\|^3, \tag{2}$$

where the matrix $A \in \mathbb{R}^{d \times d}$ is symmetric and possibly indefinite. Problem (2) is a sub-routine of the Nesterov-Polyak cubic-regularized method (Nesterov & Polyak (2006)), which aims to minimize a non-convex objective $F(\cdot)$ by iteratively solving

$$w_{t+1} = \arg\min_{w \in \mathbb{R}^d} \{\nabla F(w_t)^\top (w - w_t) + \tfrac{1}{2}(w - w_t)^\top \nabla^2 F(w)(w - w_t) + \tfrac{\rho}{3}\|w - w_t\|^3\}, \tag{3}$$

With some additional post-processing, the iterate $w_t$ converges to an $(\epsilon_g, \epsilon_h)$ second order stationary point, defined as $\{w : \|\nabla f(w)\| \le \epsilon_g \text{ and } \nabla^2 f(w) \succeq -\epsilon_h I_d\}$ for any small $\epsilon_g, \epsilon_h > 0$. However, their algorithm needs to compute a matrix inverse to solve (2), which is computationally expensive when the dimension is high. A very recent result due to Carmon & Duchi (2019) shows that vanilla gradient descent approximately finds the global minimum of (2) under mild conditions, which only needs a Hessian-vector product and can be computed in the same computational complexity as computing gradients (Pearlmutter, 1994), and hence is computationally cheaper than the matrix inversion of the Hessian. Our result shows that, similar to the case of phase retrieval, the use of Heavy Ball momentum helps the iterate to enter a benign region of (3) that contains a global optimal solution faster, compared to vanilla gradient descent. For certain non-convex problems, e.g. dictionary learning (Sun et al., 2015), matrix completion (Chi et al., 2019), robust PCA (Ge et al., 2017), and learning a neural network (Ge et al. (2019); Bai & Lee (2020)), where it suffices to find a second-order stationary point, our result consequently might have application.

To summarize, our theoretical results of the two non-convex problems provably show the benefit of using Heavy Ball momentum. Compared to the vanilla gradient descent, the use of momentum helps to accelerate the optimization process. The key to showing the acceleration in getting into benign regions of these problems is a family of simple dynamics due to Heavy Ball momentum. We will argue that the simple dynamics are not restricted to the two main problems considered in this paper. Specifically, the dynamics also naturally arise when solving the problem of top eigenvector computation (Golub & Loan, 1996) and the problem of saddle points escape (e.g. (Jin et al., 2017;

Wang et al., 2020)), which might imply the broad applicability of the dynamics for analyzing Heavy Ball in non-convex optimization.

| **Algorithm 1:** Heavy Ball method (Polyak, 1964) (Equivalent version 1) | **Algorithm 2:** Heavy Ball method (Polyak, 1964) (Equivalent version 2) |
|---|---|
| 1: Required: step size $\eta$ and momentum parameter $\beta \in [0,1]$. | 1: Required: step size $\eta$ and momentum parameter $\beta \in [0,1]$. |
| 2: Init: $w_0 = w_{-1} \in \mathbb{R}^d$ | 2: Init: $w_0 \in \mathbb{R}^d$ and $m_{-1} = 0$. |
| 3: **for** $t = 0$ to $T$ **do** | 3: **for** $t = 0$ to $T$ **do** |
| 4:    Update iterate $w_{t+1} = w_t - \eta \nabla f(w_t) + \beta(w_t - w_{t-1})$. | 4:    Update momentum $m_t := \beta m_{t-1} + \nabla f(w_t)$. |
| 5: **end for** | 5:    Update iterate $w_{t+1} := w_t - \eta m_t$. |
| | 6: **end for** |

## 2 MORE RELATED WORKS

**Heavy Ball (HB):** HB has two *exactly* equivalent presentations in the literature (see Algorithm 1 and 2). Given the same initialization, both algorithms generate the same sequence of $\{w_t\}$. In Algorithm 2, we note that the momentum $m_t$ can be written as $m_t = \sum_{s=0}^t \beta^{t-s} \nabla f(w_s)$ and can be viewed as a weighted sum of gradients. As we described in the opening paragraph, there is little theory of showing a provable acceleration of the method in non-convex optimization. The only exception that we are aware of is (Wang et al., 2020). They show that HB momentum can help to escape saddle points faster and find a second-order stationary point faster for smooth non-convex optimization. They also observed that stochastic HB solves (1) and that using higher values of the momentum parameter $\beta$ leads to faster convergence. However, while their work focused on the stochastic setting, their main result required some assumptions on the statistical properties of the sequence of observed gradients; it is not clear whether these would hold in general. In appendix A, we provide a more detailed literature review of HB. To summarize, current results in the literature imply that we are still very far from understanding deterministic HB in non-convex optimization, let alone understanding the success of stochastic HB in deep learning. Hence, this work aims to make progress on a simple question: can we give a precise advantage argument for the acceleration effect of Heavy Ball in the deterministic setting?

**Phase retrieval:** The optimization landscape of problem (1) and its variants has been studied by (Davis et al., 2018; Soltanolkotabi, 2014; Sun et al., 2016; White et al., 2016), which shows that as long as the number of samples is sufficiently large, it has no spurious local optima. We note that the problem can also be viewed as a special case of matrix sensing (e.g. Li et al. (2018); Gunasekar et al. (2017); Li & Lin (2020); Li et al. (2019); Gidel et al. (2019); You et al. (2020)); in Appendix A, we provide a brief summary of matrix sensing. For solving phase retrieval, Mannellia et al. (2020) study gradient flow, while Chen et al. (2018) show that the standard gradient descent with a random initialization like Gaussian initialization solves (1) and recovers $w_*$ up to the sign. Tan & Vershynin (2019) show that online gradient descent with a simple random initialization can converge to a global optimal point in an online setting where fresh samples are required for each step. In this paper, we show that Heavy Ball converges even faster than the vanilla gradient descent. Zhou et al. (2016) propose leveraging Nesterov's momentum to solve phase retrieval. However, their approach requires delicate and computationally expensive initialization like spectral initialization so that the initial point is already within the neighborhood of a minimizer. Similarly, Xiong et al. (2018; 2020) show local convergence of Nesterov's momentem and Heavy Ball momentum for phase retrieval, but require the initial point to be in the neighborhood of an optimal point. Jin et al. (2018) propose an algorithm that uses Nesterov's momentum together as a subroutine with perturbation for finding a second-order stationary point, which could be applied for solving phase retrieval. Compared to (Zhou et al., 2016; Jin et al., 2018; Xiong et al., 2018; 2020), we consider *directly* applying gradient descent with *Heavy Ball* momentum (i.e. HB method) to the objective function with *simple* random initialization, e.g. Gaussian initialization, which is what people do in practice and is what we want to understand. The goals of the works are different. Finally, we note that there are some efforts in integrating the technique of generative models and phase retrieval, which could help the task of image recovery (e.g. Hand et al. (2018)). Phase retrieval might also be a good entry point of understanding some observations in optimization and neural net training (e.g. Mannellia et al. (2020)).

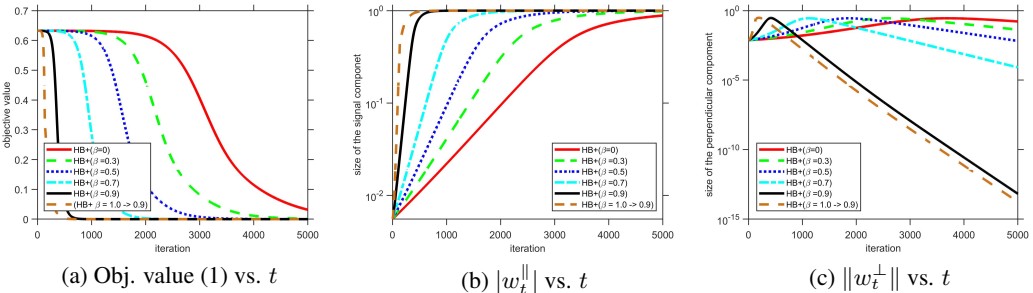

(a) Obj. value (1) vs. $t$        (b) $|w_t^{\|}|$ vs. $t$        (c) $\|w_t^{\perp}\|$ vs. $t$

Figure 1: Performance of HB with different $\beta = \{0, 0.3, 0.5, 0.7, 0.9, 1.0 \rightarrow 0.9\}$ for phase retrieval. (Enlarged figures are available in Appendix H.) Here "$1.0 \rightarrow 0.9$" stands for using parameter $\beta = 1.0$ in the first few iterations and then switching to using $\beta = 0.9$ after that. In our experiment, for the ease of implementation, we let the criteria of the switch be $\mathbb{1}\{\frac{f(w_1) - f(w_t)}{f(w_1)} \geq 0.5\}$, i.e. if the relative change of objective value compared to the initial value has been increased to 50%. Algorithm 3 and Algorithm 4 in Appendix H describe the procedures. All the lines are obtained by initializing the iterate at the same point $w_0 \sim \mathcal{N}(0, \mathcal{I}_d/(10000d))$ and using the same step size $\eta = 5 \times 10^{-4}$. Here we set $w_* = e_1$ and sample $x_i \sim \mathcal{N}(0, \mathcal{I}_d)$ with dimension $d = 10$ and number of samples $n = 200$. We see that the higher the momentum parameter $\beta$, the faster the algorithm enters the linear convergence regime. (a): Objective value (1) vs. iteration $t$. We see that the higher the momentum parameter $\beta$, the faster the algorithm enters the linear convergence regime. (b): The size of projection of $w_t$ on $w_*$ over iterations (i.e. $|w_t^{\|}|$ vs. $t$), which is non-decreasing throughout the iterations until reaching an optimal point (here, $\|w_*\| = 1$). (c): The size of the perpendicular component over iterations (i.e. $\|w_t^{\perp}\|$ vs. $t$), which is increasing in the beginning and then it is decreasing towards zero after some point. We see that the slope of the curve corresponding to a larger momentum parameter $\beta$ is steeper than that of a smaller one, which confirms Lemma 1 and Lemma 3.

## 3  PHASE RETRIEVAL

### 3.1  PRELIMINARIES

Following the works of Candés et al. (2013); Chen et al. (2018), we assume that the design vectors $\{x_i\}$ (which are known a priori) are from Gaussian distribution $x_i \sim N(0, I_d)$. Furthermore, without loss of generality, we assume that $w_* = e_1$ (so that $\|w_*\| = 1$), where $e_1$ is the standard unit vector whose first element is 1. We also denote $w_t^{\|} := w_t[1]$ and $w_{t,\perp} := [w_t[2], \ldots, w_t[d]]^{\top}$. That is, $w_t^{\|}$ is the projection of the current iterate $w_t$ on $w_*$, while $w_t^{\perp}$ is the perpendicular component. Throughout the paper, the subscript $t$ is an index of the iterations while the subscript $i$ is an index of the samples.

Before describing the main results, we would like to provide a preliminary analysis to show how momentum helps. Applying gradient descent with Heavy Ball momentum (Algorithm 1) to objective (1), we see that the iterate is generated according to $w_{t+1} = w_t - \eta \frac{1}{n} \sum_{i=1}^{n} ((x_i^{\top} w_t)^3 - (x_i^{\top} w_t) y_i) x_i + \beta(w_t - w_{t-1})$. On the other hand, the population counterpart (i.e. when the number of samples $n$ is infinite) of the update rule turns out to be the key to understanding momentum. The population gradient $\nabla F(w) := \mathbb{E}_{x \sim N(0, I_n)}[\nabla f(w)]$ is (proof is available in appendix B)

$$\nabla F(w) = (3\|w\|^2 - 1)w - 2(w_*^{\top} w)w_*. \tag{4}$$

Using the population gradient (4), we have the population update, $w_{t+1} = w_t - \eta \nabla F(w_t) + \beta(w_t - w_{t-1})$, which can be decomposed as follows:

$$
\begin{aligned}
w_{t+1}^{\|} &= (1 + 3\eta(1 - \|w_t\|^2))w_t^{\|} + \beta(w_t^{\|} - w_{t-1}^{\|}) \\
w_{t+1}^{\perp} &= (1 + \eta(1 - 3\|w_t\|^2))w_t^{\perp} + \beta(w_t^{\perp} - w_{t-1}^{\perp}).
\end{aligned}
\tag{5}
$$

Assume that the random initialization satisfies $\|w_0\|^2 \ll \frac{1}{3}$. From the population recursive system (5), both the magnitude of the signal component $w_t^{\|}$ and the perpendicular component $w_t^{\perp}$ grow exponentially in the first few iterations.

**Lemma 1.** *For a positive number $\theta$ and the momentum parameter $\beta \in [0,1]$, if a non-negative sequence $\{a_t\}$ satisfies $a_0 \geq a_{-1} > 0$ and that for all $t \leq T$,*

$$a_{t+1} \geq (1+\theta)a_t + \beta(a_t - a_{t-1}), \tag{6}$$

*then $\{a_t\}$ satisfies*

$$a_{t+1} \geq \left(1 + \left(1 + \tfrac{\beta}{1+\theta}\right)\theta\right) a_t, \tag{7}$$

*for every $t = 1, \ldots, T+1$. Similarly, if a non-positive sequence $\{a_t\}$ satisfies $a_0 \leq a_{-1} < 0$ and that for all $t \leq T$, $a_{t+1} \leq (1+\theta)a_t + \beta(a_t - a_{t-1})$, then $\{a_t\}$ satisfies*

$$a_{t+1} \leq \left(1 + \left(1 + \tfrac{\beta}{1+\theta}\right)\theta\right) a_t, \tag{8}$$

*for every $t = 1, \ldots, T+1$.*

One can view $a_t$ in Lemma 1 as the projection of the current iterate $w_t$ onto a vector of interest. The lemma says that with a larger value of the momentum parameter $\beta$, the magnitude of $a_t$ is increasing faster. It also implies that if the projection due to vanilla gradient descent satisfies $a_{t+1} \geq (1+\theta)a_t$, then the magnitude of the projection only grows faster with the use of Heavy Ball momentum. The dynamics in the lemma are the keys to showing that Heavy Ball momentum accelerates the process of entering a benign (convex) region. The factor $\frac{\beta}{1+\theta}\theta$ in (7) and (8) represents the contribution due to the use of momentum, and the contribution is larger with a larger value of momentum parameter $\beta$. Now let us apply Lemma 1 to the recursive system (5) and pretend that the magnitude of $\|w_t\|$ was a constant for a moment. Denote $\theta_t := 3\eta(1 - \|w_t\|^2)$ and $\tilde{\theta}_t := \eta(1 - 3\|w_t\|^2)$ and notice that $\theta_t > \tilde{\theta}_t > 0$ when $\|w_t\|^2 < \frac{1}{3}$. We can rewrite the recursive system as

$$
\begin{aligned}
w_{t+1}^{\parallel} &= (1+\theta_t)w_t^{\parallel} + \beta(w_t^{\parallel} - w_{t-1}^{\parallel}) \\
w_{t+1}^{\perp} &= (1+\tilde{\theta}_t)w_t^{\perp} + \beta(w_t^{\perp} - w_{t-1}^{\perp}).
\end{aligned}
\tag{9}
$$

Since the above system is in the form of (6), the dynamics (7) and (8) in Lemma 1 suggest that the larger the momentum parameter $\beta$, the faster the growth rate of the magnitude of the signal component $w_t^{\parallel}$ and the perpendicular component $w_t^{\perp}$. Moreover, the magnitude of the signal component $w_t^{\parallel}$ grows faster than that of the perpendicular component $w_t^{\perp}$. Both components will grow until the size of iterate $\|w_t\|^2$ is sufficiently large (i.e. $\|w_t\|^2 > \frac{1}{3}$). After that, the magnitude of the perpendicular component $w_t^{\perp}$ starts decaying, while $|w_t^{\parallel}|$ keeps growing until it approaches 1. Furthermore, we have that the larger the momentum parameter $\beta$, the faster the decay rate of the (magnitude of the) perpendicular component $w_t^{\perp}$. In other words, $|w_t^{\parallel}|$ converges to 1 and $w_t^{\perp}$ converges to 0 quickly. Lemma 3 in Appendix C, which is a counterpart of Lemma 1, can be used to explain the faster decay of the magnitude due to a larger value of the momentum parameter $\beta$.

By using the population recursive system (5) and Lemma 1 & 3 as the tool, we obtain a high-level insight on how momentum helps (see also Figure 1). The momentum helps to drive the iterate to enter the neighborhood of a $w_*$ (or $-w_*$) faster.

## 3.2 MAIN RESULTS

We denote $\text{dist}(w_t, w_*) := \min\{\|w_t - w_*\|_2, \|w_t + w_*\|_2\}$ as the distance between the current iterate $w_t$ and $w_*$, modulo the unrecoverable sign. Note that both $\pm w_*$ achieve zero testing errors. Furthermore, as long as the number of samples is sufficiently large, i.e. $n \gtrsim d \log d$, there exists a constant $\vartheta > 0$ so that the Hessian satisfies $\nabla^2 f(w) \succeq \vartheta I_d$ for all $w \in \mathbb{B}_\zeta(\pm w_*)$ with high probability, where $\mathbb{B}_\zeta(\pm w_*)$ represents the balls centered at $\pm w_*$ with a radius $\zeta$ (e.g. (Ma et al., 2017)). So in this paper we consider the case that the local strong convexity holds in the neighborhood $\mathbb{B}_\zeta(\pm w_*)$.

We will divide the iterations into two stages. The first stage consists of those iterations that satisfy $0 \leq t \leq T_\zeta$, where $T_\zeta$ is defined as

$$T_\zeta := \min\{t : |\|w_t^{\parallel}\| - 1| \leq \tfrac{\zeta}{2} \text{ and } \|w_t^{\perp}\| \leq \tfrac{\zeta}{2}\}, \tag{10}$$

and $\zeta > 0$ is sufficiently small so that it makes $w_{T_\zeta}$ be in the neighborhood of $w_*$ or $-w_*$ which is smooth, twice differentiable, strongly convex, see e.g. (Ma et al., 2017; Soltanolkotabi, 2014).

Observe that if $\|w_t^{\|}\| - 1| \leq \frac{\zeta}{2}$ and $\|w_t^{\perp}\| \leq \frac{\zeta}{2}$, we have that $\text{dist}(w_t, w_*) \leq \|w_t - w_*\| \leq \|w_t^{\|}\| - 1| + \|w_t^{\perp}\| \leq \zeta$. The second stage consists of those iterations that satisfy $t \geq T_\zeta$. Given that $\text{dist}(w_{T_\zeta}, w_*) \leq \zeta$ and that the local strong convexity holds in $\mathbb{B}_\zeta(\pm w_*)$, the iterate $w_t$ would be in a benign region at the start of this stage, which allows linear convergence to a global optimal point. That is, we have that $\text{dist}(w_t, w_*) \leq (1 - \nu)^{t - T_\zeta} \zeta$ for all $t \geq T_\zeta$, where $1 > \nu > 0$ is some number. Since the behavior of the momentum method in the second stage can be explained by the existing results (e.g. Section 3 of Saunders (2018) or Xiong et al. (2020)), the goal is to understand why momentum helps to drive the iterate into the benign region faster.

To deal with the case that only finite samples are available in practice, we will consider some perturbations from the population dynamics (5). In particular, we consider

$$
\begin{aligned}
w_{t+1}^{\|} &= \left(1 + 3\eta(1 - \|w_t\|^2) + \eta\xi_t\right)w_t^{\|} + \beta(w_t^{\|} - w_{t-1}^{\|}) \\
w_{t+1}^{\perp}[j] &= \left(1 + \eta(1 - 3\|w_t\|^2) + \eta\rho_{t,j}\right)w_t^{\perp}[j] + \beta(w_t^{\perp}[j] - w_{t-1}^{\perp}[j]),
\end{aligned}
\tag{11}
$$

where $\{\xi_t\}$ and $\{\rho_{t,j}\}$ for $1 \leq j \leq d-1$ are the perturbation terms. The perturbation terms are used to model the deviation from the population dynamics. In this paper, we assume that there exists a small number $c_n > 0$ such that for all iterations $t \leq T_\zeta$ and all $j \in [d-1]$, $\max\{|\xi_t|, |\rho_{t,j}|\} \leq c_n$, where the value $c_n$ should decay when the number of samples $n$ is increasing and $c_n \cong 0$ when there are sufficiently large number of samples $n$.

**Theorem 1.** *Suppose that the approximated dynamics (11) holds with $\max\{|\xi_t|, |\rho_{t,j}|\} \leq c_n$ for all iterations $t \leq T_\zeta$ and all dimensions $j \in [d-1]$, where $c_n \leq \zeta$. Suppose that the local strong convexity holds in $\mathbb{B}_\zeta(\pm w_*)$. Assume that the initial point $w_0$ satisfies $|w_0^{\|}| \gtrsim \frac{1}{\sqrt{d}\log d}$ and $\|w_0\| < \frac{1}{3}$. Set the momentum parameter $\beta \in [0, 1]$. Assume that the norm of the momentum $\|m_t\|$ is bounded for all $t \leq T_\zeta$, i.e. $\|m_t\| \leq c_m$ for some constant $c_m > 0$. If the step size $\eta$ satisfies $\eta \leq \frac{1}{36(1+\zeta)\max\{c_m, 1\}}$, then Heavy Ball (Algorithm 1 & 2) takes at most*

$$
T_\zeta \lesssim \frac{\log d}{\eta\left(1 + \boxed{c_\eta\beta}\right)}
$$

*number of iterations to enter the benign region $\mathbb{B}_\zeta(w_*)$ or $\mathbb{B}_\zeta(-w_*)$, where $c_\eta := \frac{1}{1 + \eta/2} \leq 1$. Furthermore, for all $t \geq T_\zeta$, the distance is shrinking linearly for some values of $\eta$ and $\beta < 1$. That is, we have that $\text{dist}(w_t, w_*) \leq (1 - \nu)^{t - T_\zeta}\zeta$, for some number $1 > \nu > 0$.*

The theorem states that the number of iterations required for gradient descent to enter the linear convergence is reduced by a factor of $(1 + c_\eta\beta)$, which clearly demonstrates that momentum helps to drive the iterate into a benign region faster. The constant $c_\eta$ suggests that the smaller the step size $\eta$, the acceleration due to the use of momentum is more evident. The reduction can be about $\cong 1 + \beta$ for a small $\eta$. After $T_\zeta$, Heavy Ball has a locally linear convergence to $w_*$ or $-w_*$. Specifically, if $w_0^{\|} > 0$, then we will have that $w_t^{\|} > 0$ for all $t$ and that the iterate will converge to $w_*$; otherwise, we will have $w_t^{\|} < 0$ for all $t$ and that the iterate will converge to $-w_*$. The proof of Theorem 1 is in Appendix D.

**Remark 1:** The initial point is required to satisfy $\langle w_0, w_* \rangle \gtrsim \frac{1}{\sqrt{d}\log d}$ and $\|w_0\| < \frac{1}{3}$. The first condition can be achieved by generating $w_0$ from a Gaussian distribution or uniformly sampling from a sphere with high probability (see e.g. Chapter 2 of (Blum et al., 2018)). The second condition can then be satisfied by scaling the size appropriately. We note that the condition that the norm of the momentum is bounded is also assumed in (Wang et al., 2020).

**Remark 2:** Our theorem indicates that in the early stage of the optimization process, the momentum parameter $\beta$ can be as large as 1, which is also verified in the experiment (Figure 1). However, to guarantee convergence after the iterate is in the neighborhood of a global optimal solution, the parameter $\beta$ must satisfy $\beta < 1$ (Polyak, 1964; Lessard et al., 2016).

**Remark 3:** The number $\nu$ of the local linear convergence rate due to Heavy Ball momentum actually depends on the smoothness constant $L$ and strongly convexity constant $\mu$ in the neighborhood of a global solution, as well as the step size $\eta$ and the momentum parameter $\beta$ (see e.g. Section 3 of (Saunders, 2018) or the original paper (Polyak, 1964)). By setting the step size $\eta = 4/(\sqrt{L} + \sqrt{\mu})^2$

and the momentum parameter $\beta = \max\{|1 - \sqrt{\eta L}|, |1 - \sqrt{\eta \mu}|\}^2$, $\nu$ will depend on the squared root of the condition number $\sqrt{\kappa} := \sqrt{L/\mu}$ instead of $\kappa := L/\mu$, which means that an optimal local convergence rate is achieved (e.g. (Bubeck, 2014)). In general, up to a certain threshold, a larger value of $\beta$ leads to a faster rate than that of standard gradient descent.

## 4 CUBIC-REGULARIZED PROBLEM

### 4.1 NOTATIONS

We begin by introducing the notations used in this section. For the symmetric but possibly indefinite matrix $A$, we denote its eigenvalue in the increasing order, where any $\lambda^{(i)}$ might be negative. We denote the eigen-decomposition of $A$ as $A := \sum_{i=1}^{d} \lambda^{(i)}(A) v_i v_i^\top$, where each $v_i \in \mathbb{R}^d$ is orthonormal. We also denote $\gamma := -\lambda^{(1)}(A)$, and $\gamma_+ = \max\{\gamma, 0\}$, and $\|A\|_2 := \max\{|\lambda^{(1)}(A)|, |\lambda^{(d)}(A)|\}$. For any vector $w \in \mathbb{R}^d$, we denote $w^{(i)}$ as the projection on the eigenvector of $A$, $w^{(i)} = \langle w, v_i \rangle$. Denote $w_*$ as a global minimizer of the cubic-regularized problem (2) and denote $A_* := A + \rho\|w_*\|I_d$. Previous works of Nesterov & Polyak (2006); Carmon & Duchi (2019) show that the minimizer $w_*$ has a characterization; it satisfies $\rho\|w_*\| \geq \gamma$ and $\nabla f(w_*) = A_* w_* + b = 0$. Furthermore, the minimizer $w_*$ is unique if $\rho\|w_*\| > \gamma$. In this paper, we assume that the problem has a unique minimizer so that $\rho\|w_*\| > \gamma$. The gradient of the cubic-regularized problem (2) is

$$\nabla f(w) = Aw + b + \rho\|w\|w = A_*(w - w_*) - \rho(\|w_*\| - \|w\|)w. \tag{12}$$

By applying the Heavy Ball algorithm (Algorithm 1) to the cubic-regularized problem (2), we see that it generates the iterates via

$$w_{t+1} = w_t - \eta\nabla f(w_t) + \beta(w_t - w_{t-1}) = (I_d - \eta A - \rho\eta\|w_t\|I_d)w_t - \eta b + \beta(w_t - w_{t-1}). \tag{13}$$

### 4.2 ENTERING A BENIGN REGION FASTER

For the cubic-regularized problem, we define a different notion of a benign region from that for the phase retrieval. The benign region here is smooth, contains the unique global optimal point $w_*$, and satisfies a notion of *one-point strong convexity* (Kleinberg et al., 2018; Li & Yuan, 2017; Safran et al., 2020),

$$w \in \mathbb{R}^d : \langle w - w_*, \nabla f(w) \rangle \geq \vartheta\|w - w_*\|^2, \text{ where } \vartheta > 0. \tag{14}$$

We note that the standard strong convexity used in the definition of a benign region for phase retrieval could imply the one-point strong convexity here, but not vice versa (Hinder et al., 2020).

Previous work of Carmon & Duchi (2019) shows that if the norm of the iterate is sufficiently large, i.e. $\rho\|w\| \geq \gamma - \delta$ for any sufficiently small $\delta > 0$, the iterate is in the benign region that contains the global minimizer $w_*$. To see this, by using the gradient expression of $\nabla f(w)$, we have that

$$\langle w - w_*, \nabla f(w) \rangle = (w - w_*)^\top \left(A_* + \frac{\rho}{2}(\|w\| - \|w_*\|)I_d\right)(w - w_*) \tag{15}$$
$$+ \frac{\rho}{2}(\|w_*\| - \|w\|)^2(\|w\| + \|w_*\|).$$

The first term on the r.h.s. of the equality becomes nonnegative if the matrix $A_* + \frac{\rho}{2}(\|w\| - \|w_*\|)I_d$ becomes PSD. Since $A_* \succeq (-\gamma + \rho\|w_*\|)I_d$, it means that if $\rho\|w\| \geq \gamma - (\rho\|w_*\| - \gamma)$, the matrix becomes PSD (note that by the characterization, $\rho\|w_*\| - \gamma > 0$). Furthermore, if the size of iterate $\rho\|w\|$ satisfies $\rho\|w\| > \gamma - (\rho\|w_*\| - \gamma)$, the matrix becomes positive definite and consequently, we have that $A_* + \frac{\rho}{2}(\|w\| - \|w_*\|)I_d \succ \vartheta I_d$ for a number $\vartheta > 0$. Therefore, (15) becomes

$$\langle w - w_*, \nabla f(w) \rangle \geq \vartheta\|w - w_*\|^2, \qquad \vartheta > 0. \tag{16}$$

Therefore, the benign region of the cubic regularized problem can be characterized as

$$\mathbb{B} := \{w \in \mathbb{R}^d : \rho\|w\| > \gamma - (\rho\|w_*\| - \gamma)\}. \tag{17}$$

What we are going to show is that HB with a larger value of the momentum parameter $\beta$ enters the benign region $\mathbb{B}$ faster. We have the following theorem, which shows that the size of the iterate $\rho\|w_t\|$ will grow very fast to exceed any level below $\gamma$. Furthermore, the larger the momentum parameter $\beta$, the faster the growth, which shows the advantage of Heavy Ball over vanilla gradient descent.

**Theorem 2.** *Fix any number $\delta$ that satisfies $\delta > 0$. Define $T_\delta := \min\{t : \rho\|w_{t+1}\| \geq \gamma - \delta\}$. Suppose that the initialization satisfies $w_0^{(1)} b^{(1)} \leq 0$. Set the momentum parameter $\beta \in [0,1]$. If the step size $\eta$ satisfies $\eta \leq \frac{1}{\|A\|_2 + \rho\|w_*\|}$, then Heavy Ball (Algorithm 1 & 2) takes at most*

$$T_\delta \leq \frac{2}{\eta\delta(1 + \boxed{\beta/(1+\eta\delta)})} \log\left(1 + \frac{\gamma_+^2(1 + \beta/(1+\eta\delta))}{4\rho|b^{(1)}|}\right)$$

*number of iterations required to enter the benign region $\mathbb{B}$.*

Note that the case of $\beta = 0$ in Theorem 2 reduces to the result of Carmon & Duchi (2019)which analyzes vanilla gradient descent. The lemma implies that the higher the momentum parameter $\beta$, the faster that the iterate enters the benign region for which a linear convergence is possible; see also Figure 2 for the empirical results. Specifically, $\beta$ reduces the number of iterations $T_\delta$ by a factor of $(1 + \beta/(1+\eta\delta))$ (ignoring the $\beta$ in the log factor as its effect is small), which also implies that for a smaller step size $\eta$, the acceleration effect due to the use of momentum is more evident. The factor can be approximately $1 + \beta$ for a small $\eta$. Lastly, the condition $w_0^{(1)} b^{(1)} \leq 0$ in Theorem 2 can be satisfied by $w_0 = w_{-1} = -r\frac{b}{\|b\|}$ for any $r > 0$.

*Proof.* (sketch; detailed proof is available in Appendix E) The theorem holds trivially when $\gamma \leq 0$, so let us assume $\gamma > 0$. Recall the notation that $w^{(1)}$ represents the projection of $w$ on the eigenvector $v_1$ of the least eigenvalue $\lambda^{(1)}(A)$, i.e. $w^{(1)} = \langle w, v_1 \rangle$. From the update, we have that

$$\frac{w_{t+1}^{(1)}}{-\eta b^{(1)}} = (I_d + \eta\gamma - \rho\eta\|w_t\|)\frac{w_t^{(1)}}{-\eta b^{(1)}} + 1 + \frac{\beta(w_t^{(1)} - w_{t-1}^{(1)})}{-\eta b^{(1)}}. \tag{18}$$

Denote $a_t := \frac{w_t^{(1)}}{-\eta b^{(1)}}$ and in the detailed proof we will show that $a_t \geq 0$ for all $t \leq T_\delta$. We can rewrite (18) as $a_{t+1} = (1 + \eta\gamma - \rho\eta\|w_t\|)a_t + 1 + \beta(a_t - a_{t-1}) \geq (1 + \eta\delta)a_t + 1 + \beta(a_t - a_{t-1})$, where the inequality is due to that $\rho\|w_t\| \leq \gamma - \delta$ for $t \leq T_\delta$. So we can now see that the dynamics is essentially in the form of (6) except that there is an additional 1 on the r.h.s of the inequality. Therefore, we can invoke Lemma 1 to show that the higher the momentum, the faster the iterate enters the benign region. In Appendix E, we consider the presence of 1 on the r.h.s and obtain a tighter bound than what Lemma 1 can provide. $\square$

We have shown that the simple dynamic results in entering a benign region that is *one-point strongly convex* to $w_*$ faster. However, different from the case of phase retrieval, we are not aware of any prior results of showing that the iterate generated by Heavy Ball keeps staying in a region that has the property (16) once the iterate is in the region. Carmon & Duchi (2019) show that for the cubic regularized problem, the iterate generated by vanilla gradient descent stays in the region under certain conditions, which leads to a linear convergence rate after it enters the benign region. Showing that the property holds for HB is not in the scope of this paper, but we empirically observe that Heavy Ball stays in the region. Subfigure (a) on Figure 2 shows that the norm $\|w_t\|$ is monotone increasing for a wide range of $\beta$, which means that the iterate stays in the benign region according to (17). Assuming that the iterate stays in the region, in Appendix F, we show a locally linear convergence of HB for which up to a certain threshold of $\beta$, the larger the $\beta$ the better the convergence rate.

## 5 DISCUSSION AND CONCLUSION

Let us conclude by a discussion about the applicability of the simple dynamics to other non-convex optimization problems. Let $A$ be a positive semi-definite matrix. Consider applying HB to top eigenvector computations, i.e. solving $\min_{w \in \mathbb{R}^d : \|w\| \leq 1} -\frac{1}{2}w^\top A w$, which is a non-convex optimization problem as it is about maximizing a convex function. The update of HB for this objective is $w_{t+1} = (I_d + \eta A)w_t + \beta(w_t - w_{t-1})$. By projecting the iterate $w_{t+1}$ on an eigenvector $u_i$ of the matrix $A$, we have that

$$\langle w_{t+1}, u_i \rangle = (1 + \eta\lambda_i)\langle w_t, u_i \rangle + \beta(\langle w_t, u_i \rangle - \langle w_{t-1}, u_i \rangle). \tag{19}$$

We see that this is in the form of the simple dynamics in Lemma 1 again. So one might be able to show that the larger the momentum parameter $\beta$, the faster the top eigenvector computation. This

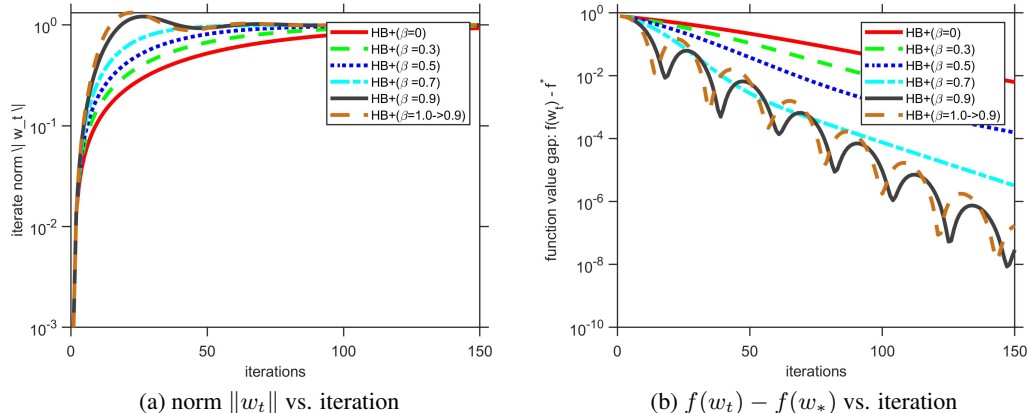

|   |   |
|---|---|
| (a) norm $\|w_t\|$ vs. iteration | (b) $f(w_t) - f(w_*)$ vs. iteration |

Figure 2: Solving (2) with different values of momentum parameter $\beta$. The empirical result shows the clear advantage of Heavy Ball momentum. Subfigure (a) shows that larger momentum parameter $\beta$ results in a faster growth rate of $\|w_t\|$, which confirms Lemma 2 and shows that it enters the benign region $\mathbb{B}$ faster with larger $\beta$. Note that here we have that $\|w_*\| = 1$. It suggests that the norm is non-decreasing during the execution of the algorithm for a wide range of $\beta$ except very large $\beta$. For $\beta = 0.9$, the norm starts decreasing only after it arises above $\|w_*\|$. Subfigure (b) show that higher $\beta$ also accelerates the linear convergence. Now let us switch to describe the setup of the experiment. We first set step size $\eta = 0.01$, dimension $d = 4$, $\rho = \|w_*\| = \|A\|_2 = 1$, $\gamma = 0.2$ and $\mathbf{gap} = 5 \times 10^{-3}$. Then we set $A = \text{diag}([-\gamma; -\gamma + \mathbf{gap}; a_{33}; a_{44}])$, where the entries $a_{33}$ and $a_{44}$ are sampled uniformly random in $[-\gamma + \mathbf{gap}; \|A\|_2]$. We draw $\tilde{w} = (A + \rho\|w_*\|I_d)^{-\xi}\theta$, where $\theta \sim \mathcal{N}(0; I_d)$ and $\log_2 \xi$ is uniform on $[-1, 1]$. We set $w_* = \frac{\|w_*\|}{\|\tilde{w}\|}\tilde{w}$ and $b = -(A + \rho\|w_*\|I_d)w_*$. The procedure makes $w_*$ the global minimizer of problem instance $(A, b, \rho)$. Patterns shown on this figure exhibit for other random problem instances as well.

connection might be used to show that the dynamics of HB momentum *implicitly* helps fast saddle points escape. In Appendix G, we provide further discussion and show some empirical evidence. We conjecture that if a non-convex optimization problem has an underlying structure like the ones in this paper, then HB might be able to exploit the structure and hence makes progress faster than vanilla gradient descent.

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

# A RELATED WORKS

## A.1 HEAVY BALL

We first note that Algorithm 1 and Algorithm 2 generate the same sequence of the iterates $\{w_t\}$ given the same initialization $w_0$, the same step size $\eta$, and the same momentum parameter $\beta$.

Lessard et al. (2016) analyze the Heavy Ball algorithm for strongly convex quadratic functions by using tools from dynamical systems and prove its accelerated linear rate. Ghadimi et al. (2015) also show an $O(1/T)$ ergodic convergence rate for general smooth convex problems, while Sun et al. (2019) show the last iterate convergence on some classes of convex problems. Nevertheless, the convergence rate of both results are not better than gradient descent. Maddison et al. (2018) and Diakonikolas & Jordan (2019) study a class of momentum methods which includes Heavy Ball by a continuous time analysis. Can et al. (2019) prove an accelerated linear convergence to a stationary distribution for strongly convex quadratic functions under Wasserstein distance. Gitman et al. (2019) analyze the stationary distribution of the iterate of a class of momentum methods that includes SGD with momentum for a quadratic function with noise, as well as studying the condition of its asymptotic convergence. Loizou & Richtárik (2017) show linear convergence results of the Heavy Ball method for a broad class of least-squares problems. Loizou & Richtárik (2018) study solving a average consensus problem by HB, which could be viewed as a strongly convex quadratic function. Sebbouh et al. (2020) show a convergence result of stochastic HB under a smooth convex setting and show that it can outperform SGD under the assumption that the data is interpolated. Chen & Kolar (2020) study stochastic HB under a growth condition. Yang et al. (2018a) show an $O(1/\sqrt{T})$ rate of convergence in expected gradient norm for smooth non-convex problems, but the rate is not better than SGD. Liu et al. (2020) provide an improved analysis of SGD with momentum. They show that SGD with momentum can converge as fast as SGD for smooth nonconvex settings in terms of the expected gradient norm. Krichene et al. (2020) show that in the continuous time regime, i.e. infinitesimal step size is used, stochastic HB converges to a stationary solution asymptotically for training a one-hidden-layer network with infinite number of neurons, but the result does not show a clear advantage compared to standard SGD. Lastly, Wang et al. (2020) show that the Heavy Ball's momentum can help to escape saddle points faster and find a second order stationary point faster for smooth non-convex optimization. However, while their work focused on the stochastic setting, their main result required two assumptions on the statistical properties of the sequence of observed gradients; it is not clear whether these would hold in general. Specifically, they make an assumption called APAG (*Almost Positively Aligned with Gradient*), i.e. $\mathbb{E}_t[\langle \nabla f(w_t), m_t - g_t \rangle] \geq -\frac{1}{2}\|\nabla f(w_t)\|^2$, where $\nabla f(w_t)$ is the deterministic gradient, $g_t$ is the stochastic gradient, and $m_t$ is the stochastic momentum. They also make an assumption called APCG (*Almost Positively Correlated with Gradient*), i.e. $\mathbb{E}_t[\langle \nabla f(w_t), M_t m_t \rangle] \geq -c'\eta \sigma_{\max}(M_t)\|\nabla f(w_t)\|^2$, where $M_t$ is a PSD matrix that is related to a local optimization landscape.

We also note that there are negative results regarding Heavy Ball (see e.g. Lessard et al. (2016); Ghadimi et al. (2015); Kidambi et al. (2018)).

## A.2 MATRIX SENSING

Problem (1) can also be viewed as a special case of matrix factorization or matrix sensing. To see this, one can rewrite (1) as $\min_{U \in \mathbb{R}^{d \times 1}} \frac{1}{4n} \sum_{i=1}^{n} (y_i - \langle A_i, UU^\top \rangle)^2$, where $A_i = x_i x_i^\top \in \mathbb{R}^{d \times d}$ and the dot product represents the matrix trace. Li et al. (2018) show that when the matrices $\{A_i\}$ satisfy restricted isometry property (RIP) and $U \in \mathbb{R}^{d \times d}$, gradient descent can converge to a global solution with a close-to-zero random initialization. Yet, if the matrix $A_i$ is in the form of a rank-one matrix product, the matrix might not satisfy RIP and a modification of the algorithm might be required (Li et al. (2018)). Li et al. (2019), a different group of authors, show that with a carefully-designed initialization (e.g. spectral initialization), gradient descent will be in a benign region in the beginning and will converge to a global optimal point. In our work, we do not assume that $x_i x_i^\top$ satisfies RIP neither do we assume a carefully-designed initialization like spectral initialization is available. Li & Lin (2020) show a *local* convergence to an optimal solution by Nesterov's momentum for a matrix factorization problem; the initial point needs to be in the neighborhood of an optimal solution. In contrast, we study Polyak's momentum and are able to establish *global* convergence to an optimal solution with a simple random initialization. Gunasekar et al. (2017); Gidel et al. (2019); You et al.

(2020) study implicit regularization of gradient descent for the matrix sensing/matrix factorization problem. The directions are different from ours.

## B  POPULATION GRADIENT

**Lemma 2.** *Assume that* $\|w_*\| = 1$.

$$\mathbb{E}_{x \sim N(0,I_d)}\big[(x^\top w)^3 x - (x^\top w_*)^2(x^\top w)x\big] = \big(3\|w\|^2 - 1\big)w - 2(w_*^\top w)w_*.$$

*Proof.* In the following, denote

$$Q := \mathbb{E}[(x^\top w)^3 x]$$
$$R := \mathbb{E}[(x^\top w_*)^2(x^\top w)x].$$

Now define $h(w) := \mathbb{E}[(x^\top w)^4]$. We have that $h(w) = 3\|w\|^4$ as $x^\top w \sim N(0, \|w\|^2)$ (i.e. the fourth order moment). Then,

$$\nabla h(w) = 4\mathbb{E}[(x^\top w)^3 x] = 4Q.$$

So $Q = \frac{1}{4}\nabla h(w) = 3\|w\|^2 w$.

For the other term, define $g(w) := \mathbb{E}[(x^\top w_*)^2(x^\top w)^2]$. Given that

$$\begin{bmatrix} x^\top w_* \\ x^\top w \end{bmatrix} \sim N\Big( \begin{bmatrix} 0 \\ 0 \end{bmatrix}, \begin{bmatrix} \|w_*\|^2 & w_*^\top w \\ w_*^\top w & \|w\|^2 \end{bmatrix} \Big), \tag{20}$$

we can write

$$x^\top w_* \overset{d}{\sim} \theta_1 x^\top w + \theta_2 z,$$

where $z \sim N(0,1)$, $\theta_1 := \frac{w_*^\top w}{\|w\|^2}$, and $\theta_2^2 := \|w_*\|^2 - \frac{(w_*^\top w)^2}{\|w\|^2}$. Then,

$$\begin{aligned} g(w) &= \theta_1^2 \mathbb{E}[(x^\top w)^4] + 2\theta_1\theta_2 \mathbb{E}[(x^\top w)^3 z] + \theta_2^2 \mathbb{E}[z^2(x^\top w)^2] \\ &= 3(w_*^\top w)^2 + \theta_2^2 \mathbb{E}[z^2]\mathbb{E}[(x^\top w)^2] \\ &= 2(w_*^\top w)^2 + \|w\|^2\|w_*\|^2. \end{aligned}$$

So we have that

$$\nabla g(w) = 2\mathbb{E}[(x^\top w_*)^2(x^\top w)x] = 2R,$$

which in turn implies that

$$R = \frac{1}{2}\nabla g(w) = \frac{1}{2}\big(4(w_*^\top w)w_* + 2\|w_*\|w\big) = 2(w_*^\top w)w_* + \|w_*\|^2 w.$$

Combining the above results, we have that

$$\nabla F(w) = Q - R = 3\|w\|^2 w - \|w_*\|^2 w - 2(w_*^\top w)w_* = \big(3\|w\|^2 - 1\big)w - 2(w_*^\top w)w_*. \tag{21}$$

$\square$

## C  SIMPLE LEMMAS

**Lemma 1:**  *For a positive number $\theta$ and the momentum parameter $\beta \in [0,1]$, if a non-negative sequence $\{a_t\}$ satisfies $a_0 \geq a_{-1} > 0$ and that for all $t \leq T$,*

$$a_{t+1} \geq (1+\theta)a_t + \beta(a_t - a_{t-1}), \tag{22}$$

*then $\{a_t\}$ satisfies*

$$a_{t+1} \geq \Big(1 + \big(1 + \tfrac{\beta}{1+\theta}\big)\theta\Big)a_t, \tag{23}$$

*for every $t = 1, \ldots, T+1$. Similarly, if a non-positive sequence $\{a_t\}$ satisfies $a_0 \leq a_{-1} < 0$ and that for all $t \leq T$, $a_{t+1} \leq (1+\theta)a_t + \beta(a_t - a_{t-1})$, then $\{a_t\}$ satisfies*

$$a_{t+1} \leq \Big(1 + \big(1 + \tfrac{\beta}{1+\theta}\big)\theta\Big)a_t, \tag{24}$$

*for every $t = 1, \ldots, T+1$.*

*Proof.* Let us first prove the first part of the statement. In the following, we denote $c := \frac{1}{1+\theta}$. For the base case $t = 1$, we have that

$$a_2 \geq (1 + \theta)a_1 + \beta(a_1 - a_0) \geq (1 + \theta)a_1 + \beta c\theta a_1, \tag{25}$$

where the last inequality holds because $a_1 - a_0 \geq \theta c a_1 \iff a_1 \geq \frac{1}{1-\theta c}a_0$, as $a_1 \geq (1 + \theta)a_0 = \frac{1}{1-\theta c}a_0$. Now suppose that it holds at iteration $t$, $a_{t+1} \geq (1 + (1+\beta c)\theta)a_t$. Consider iteration $t+1$, we have that

$$a_{t+2} \geq (1 + \theta)a_{t+1} + \beta(a_{t+1} - a_t) \geq (1 + \theta)a_{t+1} + \theta c\beta a_{t+1}, \tag{26}$$

where the last inequality holds because $a_{t+1} - a_t \geq \theta c a_{t+1} \iff a_{t+1} \geq \frac{1}{1-\theta c}a_t$ as $a_{t+1} \geq (1 + (1+\beta c)\theta)a_t \geq \frac{1}{1-\theta c}a_t$, given the assumption at $t$ and that $c := \frac{1}{1+\theta}$.

The second part of the statement can be proved similarly. $\qquad\square$

**Lemma 3.** *For a positive number $\theta < 1$ and the momentum parameter $\beta \in [0, 1]$ that satisfy $(1 + \frac{\beta}{1-\theta})\theta < 1$, if a non-negative sequence $\{b_t\}$ satisfies $b_0 \leq b_{-1}$ and that for all $t \leq T$,*

$$b_{t+1} \leq (1 - \theta)b_t + \beta(b_t - b_{t-1}), \tag{27}$$

*then $\{b_t\}$ satisfies*

$$b_{t+1} \leq \left(1 - \left(1 + \boxed{\frac{\beta}{1-\theta}}\right)\theta\right)b_t, \tag{28}$$

*for every $t = 1, \ldots, T + 1$. Similarly, if a non-positive sequence $\{b_t\}$ satisfies $b_0 \geq b_{-1}$ and that for all $t \leq T$, $b_{t+1} \geq (1 - \theta)b_t + \beta(b_t - b_{t-1})$, then $\{b_t\}$ satisfies*

$$b_{t+1} \geq \left(1 - \left(1 + \boxed{\frac{\beta}{1-\theta}}\right)\theta\right)b_t, \tag{29}$$

*for every $t = 1, \ldots, T + 1$.*

*Proof.* Let us first prove the first part of the statement. In the following, we denote $c := \frac{1}{1-\theta}$. For the base case $t = 1$, we have that

$$b_2 \leq (1 - \theta)b_1 + \beta(b_1 - b_0) \leq (1 - \theta)b_1 - \beta c\theta b_1, \tag{30}$$

where the last inequality holds because

$$b_1 - b_0 \leq -\theta c b_1 \iff b_1 \leq \frac{1}{1 + \theta c}b_0, \tag{31}$$

as $b_1 \leq (1 - \theta)b_0 = \frac{1}{1+\theta c}b_0$ due to that $c = \frac{1}{1-\theta}$. Now suppose that it holds at iteration $t$.

$$b_{t+1} \leq (1 - (1+\beta c)\theta)b_t.$$

Consider iteration $t + 1$, we have that

$$b_{t+2} \leq (1 - \theta)b_{t+1} + \beta(b_{t+1} - b_t) \leq (1 - \theta)b_{t+1} - \theta c\beta b_{t+1}, \tag{32}$$

where the last inequality holds because

$$b_{t+1} - b_t \leq -\theta c b_{t+1} \iff b_{t+1} \leq \frac{1}{1 + \theta c}b_t \tag{33}$$

as $b_{t+1} \leq (1 - (1+\beta c)\theta)b_t \leq \frac{1}{1+\theta c}b_t$, due to the induction at $t$ and that $c = \frac{1}{1-\theta}$.

The second part of the statement can be proved similarly. $\qquad\square$

# D  PROOF OF THEOREM 1

Recall the recursive system (11).

$$
\begin{aligned}
w_{t+1}^{\parallel} &= \left(1 + 3\eta(1 - \|w_t\|^2) + \eta\xi_t\right)w_t^{\parallel} + \beta(w_t^{\parallel} - w_{t-1}^{\parallel}) \\
w_{t+1}^{\perp}[j] &= \left(1 + \eta(1 - 3\|w_t\|^2) + \eta\rho_{t,j}\right)w_t^{\perp}[j] + \beta(w_t^{\perp}[j] - w_{t-1}^{\perp}[j]),
\end{aligned}
\tag{34}
$$

where $\{\xi_t\}$ and $\{\rho_{t,j}\}$ are the perturbation terms. In the analysis, we will show that the sign of $w_t^{\parallel}$ never changes during the execution of the algorithm. Our analysis divides the iterations of the first stage to several sub-stages. We assume that the size of the initial point $w_0$ is small so that it begins in Stage 1.1.

- (Stage 1.1) considers the duration when $\|w_t\| \leq \sqrt{\frac{4}{9} + c_n}$, which lasts for at most $T_0$ iterations, where $T_0$ is defined in Lemma 4. A by-product of our analysis shows that $|w_{T_0+1}^{\parallel}| \geq \sqrt{\frac{4}{9} + c_n}$.

- (Stage 1.2) considers the duration when the perpendicular component $\|w_t^{\perp}\|$ is decreasing and eventually falls below $\zeta/2$, which consists of all iterations $T_0 \leq t \leq T_0 + T_b$, where $T_b$ is defined in Lemma 5.

- (Stage 1.3) considers the duration when $|w_t^{\parallel}|$ is converging to the interval $[1 - \frac{\zeta}{2}, 1 + \frac{\zeta}{2}]$, if it was outside the interval, which consists of all iterations $T_0 + T_b \leq t \leq T_0 + T_b + T_a$, where $T_a$ is defined in Lemma 6.

In stage 1.1, both the signal component $|w_t^{\parallel}|$ and $\|w_t^{\perp}\|$ grow in the beginning (see also Figure 1). The signal component grows exponentially and consequently it only takes a logarithm number of iterations $T_0 + 1$ to reach $|w_{T_0+1}^{\parallel}| > \sqrt{\frac{4}{9} + c_n}$, which also means that $\|w_{T_0+1}\| > \sqrt{\frac{4}{9} + c_n}$. Moreover, the larger the momentum parameter $\beta$, the smaller the number of $T_0$. After $\|w_t\|$ passing the threshold, the iterate enters Stage 1.2.

**Lemma 4.** *(Stage 1.1) Denote $c_a := \frac{1}{1+\frac{\eta}{2}}$. There will be at most $T_0 := \frac{\log(\sqrt{\frac{4}{9}+c_n}/|w_0^{\parallel}|)}{\log\left(1+(1+c_a\beta)\eta\frac{5}{3}\right)}$ iterations such that $w_t^{\parallel} \leq \sqrt{\frac{4}{9} + c_n}$. Furthermore, for all $t$ in this stage, $\|w_t\| \leq \frac{5}{6} - \frac{c_n}{3}$.*

In stage 1.2, we have that $\|w_t\|^2 \geq |w_t^{\parallel}|^2 \geq \frac{4}{9} + c_n > \frac{1}{3}$ so that the perpendicular component $\|w_t^{\perp}\|$ is decaying while the signal component $|w_t^{\parallel}|$ keeps growing before reaching 1 (see also Figure 1). In particular, the perpendicular component decays exponentially so that at most additional $T_b$ iterations is needed to fall below $\frac{\zeta}{2}$ (i.e. $\|w_t^{\perp}\| \leq \frac{\zeta}{2}$), with larger $\beta$ leading to a smaller number of $T_b$. Notice that if $w_t^{\parallel}$ satisfies $||w_t^{\parallel}| - 1| \leq \frac{\zeta}{2}$ when $\|w_t^{\perp}\|$ falls below $\frac{\zeta}{2}$, we immediately have that $T_\zeta \leq T_0 + T_b$; otherwise, the iterate enters the next stage, Stage 1.3.

**Lemma 5.** *(Stage 1.2) Denote $c_b := \frac{1}{1-\frac{\eta}{3}}$ and $\bar{\omega} := \max_t \|w_t^{\perp}\| \leq \sqrt{\frac{4}{9} + c_n}$. There will be at most $T_b \leq \frac{\log(\frac{\zeta}{2\bar{\omega}})}{\log(1-\frac{\eta}{3}(1+c_b\beta))}$ iterations such that $\sqrt{\frac{4}{9} + c_n} \leq |w_t^{\parallel}|$ and $\|w_t^{\perp}\| \geq \frac{\zeta}{2}$.*

In stage 1.3, we show that $|w_t^{\parallel}|$ converges towards 1 linearly, given that $\|w_t^{\perp}\| \leq \frac{\zeta}{2}$. Specifically, after at most additional $T_a$ iterations, we have that $||w_t^{\parallel}| - 1| \leq \frac{\zeta}{2}$ and that larger $\beta$ reduces the number of iterations $T_a$.

**Lemma 6.** *(Stage 1.3) Denote $c_{\zeta,g} := \frac{1}{1+\eta\frac{\zeta}{2}}$, $c_{\zeta,d} := \frac{1}{1-\eta\zeta}$, and $\omega := \max_t |w_t^{\parallel}| \leq \sqrt{\frac{10}{9} + \frac{1}{3}c_n}$. There will be at most $T_a := \max\left(\frac{\log((1-\zeta/2)/\sqrt{\frac{4}{9}+c_n})}{\log(1+\eta\frac{\zeta}{2}(1+c_{\zeta,g}\beta))}, \frac{\log(\frac{1+\zeta/2}{\omega})}{\log(1-\eta\zeta(1+c_{\zeta,d}\beta))}\right)$ iterations such that $||w_t^{\parallel}| - 1| \geq \frac{\zeta}{2}$ and $\|w_t^{\perp}\| \leq \frac{\zeta}{2}$.*

By combining the result of Lemma 4, Lemma 5 and Lemma 6, we have that

$$
\begin{aligned}
T \leq T_0 + T_b + T_a &= \frac{\log(\frac{\sqrt{\frac{4}{9}+c_n}}{|w_0^\||})}{\log\left(1+\eta\frac{5}{3}(1+c_a\beta)\right)} + \frac{\log(\frac{\zeta}{2\bar\omega})}{\log(1-\frac{\eta}{3}(1+c_b\beta))} \\
&\quad + \max\left(\frac{\log(\frac{1-\zeta/2}{\sqrt{\frac{4}{9}+c_n}})}{\log(1+\eta\frac{\zeta}{2}(1+c_{\zeta,g}\beta))}, \frac{\log(\frac{1+\zeta/2}{\omega})}{\log(1-\eta\zeta(1+c_{\zeta,d}\beta))}\right) \\
&\lesssim \frac{6\log(\frac{\sqrt{\frac{4}{9}+c_n}}{|w_0^\||})}{5\eta(1+c_a\beta)} + \frac{6\log(\frac{2\bar\omega}{\zeta})}{\eta(1+c_b\beta)} + \max\left(\frac{2\log(\frac{1-\zeta/2}{\sqrt{\frac{4}{9}+c_n}})}{\eta\zeta(1+c_{\zeta,g}\beta)}, \frac{2\log(\frac{\omega}{1+\zeta/2})}{\eta\zeta(1+c_{\zeta,d}\beta)}\right) \\
&\lesssim \frac{\log d}{\eta\left(1+c_\eta\beta\right)},
\end{aligned}
\tag{35}
$$

where the last inequality uses that $|w_0^\|| \gtrsim \frac{1}{\sqrt{d\log d}}$ due to the random isotropic initialization. The detailed proof of Lemma 4-6 is available in the following subsections.

After $t \geq T_\zeta$, the iterate enters a benign region that is locally strong convex, smooth, twice differentiable, and contains $w_*$ (or $-w_*$) (Ma et al., 2017; White et al., 2016), which allows us to use the existing result of gradient descent with Heavy Ball momentum for showing its linear convergence. In particular, the result of local landscape (e.g. Ma et al. (2017); White et al. (2016)) and the known convergence result of gradient descent with Heavy Ball momentum (e.g. Saunders (2018); Polyak (1964); Lessard et al. (2016); Xiong et al. (2020)) can be used to show that for all $t > T_\zeta$, $\text{dist}(w_t, w_*) \leq (1-\nu)^{t-T_\zeta}\text{dist}(w_{T_\zeta}, w_*) \leq (1-\nu)^{t-T_\zeta}\zeta$, for some number $1 > \nu > 0$.

### D.1 STAGE 1.1

**Lemma: 4 (Stage 1.1)** *Denote* $c_a := \frac{1}{1+\frac{\eta}{2}}$. *There will be at most* $T_0 := \lceil\frac{\log(\sqrt{\frac{4}{9}+c_n}/|w_0^\||)}{\log\left(1+(1+c_a\beta)\eta\frac{5}{3}\right)}\rceil$ *iterations such that* $\|w_t\| \leq \sqrt{\frac{4}{9}+c_n}$. *Furthermore, we have that* $|w_{T_0+1}^\|| > \sqrt{\frac{4}{9}+c_n}$.

*Proof.* Let us first assume that $w_t^\| > 0$ and denote $a_t := w_t^\|$. By using that $\|w_t\| \leq \sqrt{\frac{4}{9}+c_n}$ in this stage, we can lower-bound the growth rate of $a_t$ as

$$
\begin{aligned}
a_{t+1} &\geq \left(1+3\eta(1-\|w_t\|^2)-\eta|\xi_t|\right)a_t + \beta(a_t - a_{t-1}) \\
&\geq \left(1+3\eta(1-\frac{4}{9}+c_n)-c_n\right)a_t + \beta(a_t - a_{t-1}) \\
&\geq \left(1+\eta\frac{5}{3}\right)a_t + \beta(a_t - a_{t-1}) \\
&\geq \left(1+(1+c_a\beta)\eta\frac{5}{3}\right)a_t,
\end{aligned}
\tag{36}
$$

where in the last inequality we use Lemma 1 and that

$$
c_a = \frac{1}{1+\eta\frac{5}{3}}.
\tag{37}
$$

Observe that $\left(1+(1+c_a\beta)\eta\frac{5}{3}\right) > 1$. Consequently, the sign of $w_t^\|$ never change in this stage. So for $w_t^\| \geq \sqrt{\frac{4}{9}+c_n}$ it takes number of iterations at most

$$
T_0 := \lceil\frac{\log(\frac{\sqrt{\frac{4}{9}+c_n}}{|w_0^\||})}{\log\left(1+(1+c_a\beta)\eta\frac{5}{3}\right)}\rceil \leq \frac{2\log(\frac{\sqrt{\frac{4}{9}+c_n}}{|w_0^\||})}{(1+c_a\beta)\eta\frac{5}{3}}.
\tag{38}
$$

Similar, when $w_t^\| < 0$, we can show that after at most $T_0$ iterations, $w_t^\|$ falls below $-\sqrt{\frac{4}{9}+c_n}$. Since $\|w_t\| > |w_t^\||$, it means that there will be at most $T_0$ iterations such that $\|w_t\| \leq \sqrt{\frac{4}{9}+c_n}$. $\quad\square$

## D.2 STAGE 1.2

**Lemma 5: (Stage 1.2)** *Denote* $c_b := \frac{1}{1-\frac{\eta}{3}}$ *and* $\bar{\omega} := \max_t \|w_t^{\perp}\| \le \sqrt{\frac{4}{9} + c_n}$. *There will be at most* $T_b := \lceil \frac{\log(\frac{\zeta}{2\bar{\omega}})}{\log(1-\frac{\eta}{3}(1+c_b\beta))} \rceil$ *iterations such that* $\sqrt{\frac{4}{9} + c_n} \le |w_t^{\|}|$ *and* $\|w_t^{\perp}\| \ge \frac{\zeta}{2}$.

*Proof.* Let $t'$ be the last iteration of the previous stage. We have that $\|w_{t'}\| \ge \|w_{t'}^{\|}\| \ge \sqrt{\frac{4}{9} + c_n}$. Denote $a_t := |w_t^{\|}|$. In this stage, we have that $a_t$ keeps increasing until $\|w_t\|^2 \gtrsim 1$. Moreover, $w_t^{\|}$ remains the same sign as the previous stage. Now fix an element $j \ne 1$ and denote $b_t := |w_t^{\perp}[j]|$. From Lemma 7, we know that the magnitude $b_t := |w_t^{\perp}[j]|$ is non-increasing in this stage. Furthermore, we can show the decay of $b_t$ as follows. If $w_t[j], w_{t-1}[j] > 0$,

$$
\begin{aligned}
w_{t+1}[j] &\le \big(1 + \eta(1 - 3\|w_t\|^2) + \eta|\rho_{t,j}|\big)w_t[j] + \beta(w_t[j] - w_{t-1}[j]) \\
&\le \big(1 + \eta\big(1 - 3(\frac{4}{9} + c_n)\big) + \eta c_n\big)w_t[j] + \beta(w_t[j] - w_{t-1}[j]) \\
&\le \big(1 - \frac{\eta}{3}\big)w_t[j] + \beta(w_t[j] - w_{t-1}[j]) \\
&\le \big(1 - \frac{\eta}{3}(1 + c_b\beta)\big)w_t[j],
\end{aligned}
\tag{39}
$$

where in the last inequality we used Lemma 3, as the condition $(1 + \frac{\beta}{1-\frac{\eta}{3}})\frac{\eta}{3} < 1$ is satisfied, and we denote that

$$
c_b := \frac{1}{1 - \eta/3}.
\tag{40}
$$

On the other hand, if $w_t[j], w_{t-1}[j] < 0$,

$$
\begin{aligned}
w_{t+1}[j] &\ge \big(1 + \eta(1 - 3\|w_t\|^2) + \eta|\rho_{t,j}|\big)w_t[j] + \beta(w_t[j] - w_{t-1}[j]) \\
&\ge \big(1 + \eta\big(1 - 3(\frac{4}{9} + c_n)\big) + \eta c_n\big)w_t[j] + \beta(w_t[j] - w_{t-1}[j]) \\
&\ge \big(1 - \frac{\eta}{3}\big)w_t[j] + \beta(w_t[j] - w_{t-1}[j]) \\
&\ge \big(1 - \frac{\eta}{3}(1 + c_b\beta)\big)w_t[j],
\end{aligned}
\tag{41}
$$

where in the last inequality we used Lemma 3, as the condition $(1 + \frac{\beta}{1-\frac{\eta}{3}})\frac{\eta}{3} < 1$ is satisfied, and we denote that

$$
c_b := \frac{1}{1 - \eta/3}.
\tag{42}
$$

The inequalities of (39) and (41) allow us to write $|w_{t+1}[j]| \le \big(1 - \frac{\eta}{3}(1 + c_b\beta)\big)|w_t[j]|$. Taking the square of both sides and summing all dimension $j \ne 1$, we have that

$$
\|w_{t+1}^{\perp}\|^2 \le \big(1 - \frac{\eta}{3}(1 + c_b\beta)\big)^2 \|w_t^{\perp}\|^2.
\tag{43}
$$

Consequently, for $\|w_t^{\perp}\|$ to fall below $\zeta/2$, it takes at most

$$
T_b := \lceil \frac{\log(\frac{\zeta/2}{\|w_{t'}^{\perp}\|})}{\log(1 - \frac{\eta}{3}(1 + c_b\beta))} \rceil \le \lceil \frac{\log(\frac{\zeta}{2\bar{\omega}})}{\log(1 - \frac{\eta}{3}(1 + c_b\beta))} \rceil
\tag{44}
$$

iterations.

Lastly, Lemma 7 implies that at the time that the magnitude of $w_t^{\perp}[j]$ starts decreasing, $\|w_t\|^2 \le \frac{4}{9} + c_n$, which in turn implies that $\bar{\omega} := \max_t \|w_t^{\perp}\| \le \sqrt{\frac{4}{9} + c_n}$.

$\square$

### D.3 STAGE 1.3

**Lemma 6: (Stage 1.3)** *Denote* $c_{\zeta,g} := \frac{1}{1+\eta\frac{\zeta}{2}}$, $c_{\zeta,d} := \frac{1}{1-\eta\zeta}$, *and* $\omega := \max_t |w_t^{\parallel}| \leq \sqrt{\frac{10}{9} + \frac{1}{3}c_n}$.
*There will be at most* $T_a := \max\left(\frac{\log((1-\zeta/2)/\sqrt{\frac{4}{9}+c_n})}{\log(1+\eta\frac{\zeta}{2}(1+c_{\zeta,g}\beta))}, \frac{\log(\frac{1+\zeta/2}{\omega})}{\log(1-\eta\zeta(1+c_{\zeta,d}\beta))}\right)$ *iterations such that*
$||w_t^{\parallel}| - 1| \geq \frac{\zeta}{2}$ *and* $\|w_t^{\perp}\| \leq \frac{\zeta}{2}$.

*Proof.* Denote $t'$ the last iteration of the previous stage. We have that $t' \leq T_0 + T_b$. Since $w_t^{\parallel}$ does not change the sign in stage 1.1 and 1.2, w.l.o.g, we assume that $w_t^{\parallel} > 0$. Denote $a_t := w_t^{\parallel}$. We consider $a_{t'}$ in two cases: $a_{t'} \leq 1 - \frac{\zeta}{2}$ and $a_{t'} \geq 1 + \frac{\zeta}{2}$. If $a_{t'} \leq 1 - \frac{\zeta}{2}$, then we have that for all $t$ in this stage, $\|w_t\|^2 \leq a_t^2 + \|w_t^{\perp}\|^2 \leq (1 - \frac{\zeta}{2})^2 + (\frac{\zeta}{2})^2 \leq 1 - \frac{\zeta}{2}$, for any sufficiently small $\zeta$. So $a_t$ grows as follows.

$$
\begin{aligned}
a_{t+1} &\geq \left(1 + 3\eta(1 - \|w_t\|^2) - \eta|\xi_t|\right)a_t + \beta(a_t - a_{t-1}) \\
&\geq \left(1 + 3\eta\frac{\zeta}{2} - \eta c_n\right)a_t + \beta(a_t - a_{t-1}) \\
&\overset{(a)}{\geq} \left(1 + \eta\frac{\zeta}{2}\right)a_t + \beta(a_t - a_{t-1}) \\
&\overset{(b)}{\geq} \left(1 + \eta\frac{\zeta}{2}(1 + c_{\zeta,g}\beta)\right)a_t,
\end{aligned}
\tag{45}
$$

where (a) is by $\zeta \geq c_n$ and (b) is due to that Lemma 1 and that

$$
c_{\zeta,g} = \frac{1}{1 + \eta\frac{\zeta}{2}}.
\tag{46}
$$

Consequently, it takes at most

$$
\lceil \frac{\log\frac{1-\zeta/2}{\sqrt{\frac{4}{9}+c_n}}}{\log(1 + \eta\frac{\zeta}{2}(1 + c_{\zeta,g}\beta))} \rceil
\tag{47}
$$

number of iterations in this stage for $a_t$ to rise above $1 - \frac{\zeta}{2}$. On the other hand, if $a_{t'} \geq 1 + \frac{\zeta}{2}$, then we can lower bound $\|w_t\|^2$ in this stage as $\|w_t\|^2 \geq a_t^2 \geq (1 + \frac{\zeta}{2})^2$. We have that

$$
\begin{aligned}
a_{t+1} &\leq \left(1 + 3\eta(1 - \|w_t\|^2) + \eta|\xi_t|\right)a_t + \beta(a_t - a_{t-1}) \\
&\leq \left(1 - 3\eta\zeta + c_n\right)a_t + \beta(a_t - a_{t-1}) \\
&\overset{(a)}{\leq} \left(1 - \eta\zeta\right)a_t + \beta(a_t - a_{t-1}) \\
&\overset{(b)}{\leq} \left(1 - \eta\zeta(1 + c_{\zeta,d}\beta)\right)a_t,
\end{aligned}
\tag{48}
$$

where (a) uses $\zeta \geq c_n$ and (b) uses Lemma 3

$$
c_{\zeta,d} := \frac{1}{1 - \eta\zeta}.
\tag{49}
$$

That is, $a_t$ is decreasing towards $1 + \zeta/2$. Denote $\omega := \max_t |w_t^{\parallel}|$. we see that it takes at most

$$
\lceil \frac{\log\frac{1+\zeta/2}{\omega}}{\log(1 - \eta\zeta(1 + c_{\zeta,d}\beta))} \rceil
\tag{50}
$$

number of iterations in this stage for $a_t$ to fall below $1 + \zeta/2$. Lastly, Lemma 8 implies that at the time that the magnitude of $w_t^{\parallel}$ starts decreasing, $\|w_t\|^2 \leq \frac{10}{9} + \frac{1}{3}c_n$, which in turn implies that $\omega := \max_t |w_t^{\parallel}| \leq \sqrt{\frac{10}{9} + \frac{1}{3}c_n}$.

On the other hand, by Lemma 10, the magnitude of the perpendicular component is non-increasing in this stage, and hence $\|w_t^{\perp}\|$ keeps staying below $\zeta/2$.

Similar analysis holds for $w_t^{\parallel} < 0$, hence we omitted the details.

$\square$

### D.4 Some supporting lemmas

**Lemma 7.** *Suppose that $\eta$ satisfies $\eta \leq \frac{1}{36\left(\frac{1}{3}+c_n\right)\max\{c_m,1\}}$. Let $t_0$ be the first time such that* $\|w_{t_0}\|^2 \geq \frac{1}{3} + c_n$. *Then, there exists a time $\tau \leq t_0$ such that $w_{\tau+1}^{\perp}[j] \leq w_{\tau}^{\perp}[j]$, if $w_{\tau}^{\perp}[j] \geq 0$; Similarly, $w_{\tau+1}^{\perp}[j] \geq w_{\tau}^{\perp}[j]$, if $w_{\tau}^{\perp}[j] \leq 0$. Furthermore, we have that $\|w_{t_0}\|^2 \leq \frac{4}{9} + c_n$.*

*Proof.* Recall that $w_{t+1}^{\perp}[j] = \left(1 + \eta(1 - 3\|w_t\|^2) + \eta\rho_{t,j}\right)w_t^{\perp}[j] + \beta(w_t^{\perp}[j] - w_{t-1}^{\perp}[j])$. W.l.o.g, let us consider $w_0^{\perp}[j] > 0$. Assume that $w_t^{\perp}[j] \geq w_{t-1}^{\perp}[j]$ for all $t \leq t_0$; otherwise, there exists a time $\tau < t_0$ such that $w_{\tau}^{\perp}[j] \leq w_{\tau-1}^{\perp}[j]$.

Denote $\lambda_{t_0,j} := \eta(1 - 3\|w_{t_0}\|^2) + \eta\rho_{t_0,j}$. Since $\|w_{t_0}\|^2 \geq \frac{1}{3} + c_n$, we have that $\lambda_{t_0,j} \leq 0$. We can rewrite the dynamics as

$$\begin{bmatrix} w_{t_0+1}^{\perp}[j] \\ w_{t_0}^{\perp}[j] \end{bmatrix} = \begin{bmatrix} 1 + \lambda_{t_0,j} + \beta & -\beta \\ 1 & 0 \end{bmatrix} \begin{bmatrix} w_{t_0}^{\perp}[j] \\ w_{t_0-1}^{\perp}[j] \end{bmatrix} \tag{51}$$

We have that

$$\left\| \begin{bmatrix} w_{t_0+1}^{\perp}[j] \\ w_{t_0}^{\perp}[j] \end{bmatrix} \right\| \leq \left\| \begin{bmatrix} 1 + \lambda_{t_0,j} + \beta & -\beta \\ 1 & 0 \end{bmatrix} \right\|_2 \cdot \left\| \begin{bmatrix} w_{t_0}^{\perp}[j] \\ w_{t_0-1}^{\perp}[j] \end{bmatrix} \right\|. \tag{52}$$

We will show that $w_{t_0+1}^{\perp}[j] \leq w_{t_0-1}^{\perp}[j]$ if $w_{t_0-1}^{\perp}[j] > 0$, which means that the magnitude of $w^{\perp}[j]$ has stopped increasing. It suffices to show that the spectral norm of the matrix $\left\| \begin{bmatrix} 1 + \lambda_{t_0,j} + \beta & -\beta \\ 1 & 0 \end{bmatrix} \right\|_2$ is not greater than 1.

Note that the roots of the characteristic equation of the matrix, $z^2 - (1 + \lambda_{t_0,j} + \beta)z + \beta$ are $\frac{\left(1+\lambda_{t_0,j}+\beta\right)\pm\sqrt{(1+\lambda_{t_0,j}+\beta)^2-4\beta}}{2}$.

If the roots are complex conjugate, then the magnitude of the roots is at most $\sqrt{\beta}$; consequently the spectral norm is at most $\sqrt{\beta} \leq 1$. On the other hand, if the roots are real, to show that the larger root is not larger than 1, it suffices to show that $\sqrt{(1 + \lambda_{t_0,j} + \beta)^2 - 4\beta} \leq 1 - \lambda_{t_0,j} - \beta$, which is guaranteed if $\lambda_{t_0,j} \leq 0$. To show that the smaller root is not greater than $-1$, we need to show that $\sqrt{(1 + \lambda_{t_0,j} + \beta)^2 - 4\beta} \leq 3 + \lambda_{t_0,j} + \beta$, which is guaranteed if $\lambda_{t_0,j} \geq -1$.

By definition, $\|w_{t_0-1}\|^2 < \frac{1}{3} + c_n$. If $\eta \leq \min\{\frac{1}{36\|w_{t_0-1}\|c_m}, \frac{1}{18c_m}\}$, then by invoking Lemma 9, we have that $\|w_{t_0}\|^2 - \|w_{t_0-1}\|^2 \leq \frac{1}{9}$ and consequently, we have that $\|w_{t_0}\|^2 \leq \frac{4}{9} + c_n$. Thus, by choosing $\eta$ satisfies $\eta \leq \frac{1}{36(1/3+c_n)c_m}$, we have that $\|w_{t_0}\|^2 \leq \frac{4}{9} + c_n$. Using the upper-bound of $\|w_{t_0}\|^2$ and the constraint of $\eta$, we have that $\lambda_{t_0,j} \geq -\eta(\frac{1}{3} + c_n) \geq -1$.

Similar analysis when $w_t^{\perp}[j]$ is negative; and hence omitted.

$\square$

**Lemma 8.** *Suppose that $\eta$ satisfies $\eta \leq \frac{1}{36\left(1+\frac{1}{3}c_n\right)\max\{c_m,1\}}$. Let $t_1$ be the first time (if exist) such that $\|w_{t_1}\|^2 \geq 1 + \frac{1}{3}c_n$. Then, there exists a time $\tau \leq t_1$ such that $w_{\tau+1}^{\|} \leq w_{\tau}^{\|}$, if $w_{\tau}^{\|} \geq 0$. Similarly, $w_{\tau+1}^{\|} \geq w_{\tau}^{\|}$, if $w_{\tau}^{\|} \leq 0$. Furthermore, we have that $\|w_{t_1}\|^2 \leq \frac{10}{9} + \frac{1}{3}c_n$.*

*Proof.* Recall that $w_{t+1}^{\|} = \left(1 + 3\eta(1 - \|w_t\|^2) + \eta\xi_t\right)w_t^{\|} + \beta(w_t^{\|} - w_{t-1}^{\|})$. W.l.o.g, let us consider $w_0^{\|} > 0$. Assume that $w_t^{\|} \geq w_{t-1}^{\|}$ for all $t \leq t_1$; otherwise, there exists a time $\tau < t_1$ such that $w_{\tau}^{\|} \leq w_{\tau-1}^{\|}$.

Denote $\lambda_{t_1} := 3\eta(1 - \|w_{t_1}\|^2) + \eta\xi_{t_1}$. Since $\|w_{t_1}\|^2 \geq 1 + \frac{1}{3}c_n$, we have that $\lambda_{t_1} \leq 0$. We can rewrite the dynamics as

$$\begin{bmatrix} w_{t_1+1}^{\|} \\ w_{t_1}^{\|} \end{bmatrix} = \begin{bmatrix} 1 + \lambda_{t_1} + \beta & -\beta \\ 1 & 0 \end{bmatrix} \begin{bmatrix} w_{t_1}^{\|} \\ w_{t_1-1}^{\|} \end{bmatrix} \tag{53}$$

We have that

$$\| \begin{bmatrix} w^{\|}_{t_1+1} \\ w^{\|}_{t_1} \end{bmatrix} \| \leq \| \begin{bmatrix} 1 + \lambda_{t_1} + \beta & -\beta \\ 1 & 0 \end{bmatrix} \|_2 \cdot \| \begin{bmatrix} w^{\|}_{t_1} \\ w^{\|}_{t_1-1} \end{bmatrix} \|. \tag{54}$$

The analysis essentially follows the same lines as Lemma 7. Specifically, to show that $w^{\|}_{t_1+1} \leq w^{\|}_{t_1-1}$, it suffices to ensure that $\lambda_{t_1} := 3\eta(1 - \|w_{t_1}\|^2) + \eta\xi_{t_1} \in [-1, 0]$. We have that $\lambda_{t_1} \leq 0$ by the definition of $t_1$. Furthermore, by the definition of $t_1 - 1$, we have that $\|w_{t_1-1}\|^2 < 1 + \frac{1}{3}c_n$. So if $\eta \leq \min\{\frac{1}{36\|w_{t_1-1}\|c_m}, \frac{1}{18c_m}\}$, then by invoking Lemma 9, we have that $\|w_{t_1}\|^2 - \|w_{t_1-1}\|^2 \leq \frac{1}{9}$ and consequently, we have that $\|w_{t_1}\|^2 \leq \frac{10}{9} + \frac{1}{3}c_n$. Thus, by choosing $\eta$ satisfies $\eta \leq \frac{1}{36\left(1+\frac{1}{3}c_n\right)c_m}$, we have that $\|w_{t_1}\|^2 \leq \frac{10}{9} + \frac{1}{3}c_n$. Using this upper-bound of $\|w_{t_1}\|^2$ and the constraint of $\eta$, we have that $\lambda_{t_1} \geq -\eta(\frac{1}{3} + 2c_n) \geq -1$. Therefore, we have completed the proof.

$\square$

**Lemma 9.** *Assume that the norm of the momentum is bounded for all $t \leq T_\zeta$, i.e. $\|m_t\| \leq c_m, \forall t \leq T_\zeta$. Set the step size $\eta$ satisfies $\eta \leq \min\{\frac{1}{36\|w_{t-1}\|c_m}, \frac{1}{18c_m}\}$. Then, we have that*

$$\|w_t\|^2 - \|w_{t-1}\|^2 \leq \frac{1}{9}.$$

*Proof.* To see this, we will use the alternative presentation Algorithm 2, which shows that $w_t = w_{t-1} - \eta m_{t-1}$, where the momentum $m_{t-1}$ stands for the weighted sum of gradients up to (and including) iteration $t - 1$, i.e. $m_{t-1} = \sum_{s=0}^{t-1} \beta^{t-1-s} \nabla f(w_s)$. Using the expression, we can expand $\|w_t\|^2 - \|w_{t-1}\|^2$ as

$$\|w_t\|^2 - \|w_{t-1}\|^2 = \|w_{t-1} - \eta m_{t-1}\|^2 - \|w_{t-1}\|^2 = -2\eta\langle w_{t-1}, m_{t-1}\rangle + \eta^2\|m_{t-1}\|^2$$
$$\leq 2\eta\|w_{t-1}\|\|m_{t-1}\| + \eta^2\|m_{t-1}\|^2 \leq \frac{1}{9}. \tag{55}$$

where the last inequality holds if $\eta \leq \min\{\frac{1}{36\|w_{t-1}\|c_m}, \frac{1}{18c_m}\}$.

$\square$

**Lemma 10.** *Fix an index $j$. Set $\eta \leq \frac{1}{36(1+\frac{1}{3}c_n)\max\{c_m,1\}}$ and $\beta \leq 1$. Suppose that $\|w_t\|^2 \geq \frac{1}{3} + c_n$. If for a number $R > 0$, we have that $\| \begin{bmatrix} w^\perp_t[j] \\ w^\perp_{t-1}[j] \end{bmatrix} \| \leq R$, then $\| \begin{bmatrix} w^\perp_{t+1}[j] \\ w^\perp_t[j] \end{bmatrix} \| \leq R$.*

*Proof.* The proof is similar to that of Lemma 7. We have that

$$\| \begin{bmatrix} w^\perp_{t+1}[j] \\ w^\perp_t[j] \end{bmatrix} \| \leq \| \begin{bmatrix} 1 + \lambda_{t,j} + \beta & -\beta \\ 1 & 0 \end{bmatrix} \|_2 \cdot \| \begin{bmatrix} w^\perp_t[j] \\ w^\perp_{t-1}[j] \end{bmatrix} \|. \tag{56}$$

Denote $\lambda_{t,j} := \eta(1 - 3\|w_t\|^2) + \eta\rho_{t,j}$. As the proof of Lemma 7, to show that the spectral norm of the matrix $\| \begin{bmatrix} 1 + \lambda_{t,j} + \beta & -\beta \\ 1 & 0 \end{bmatrix} \|_2$ is not greater than one. It suffices to have $\beta \leq 1$ and that $\lambda_{t,j} \in [-1, 0]$. By the assumption, it holds that $\|w_t\|^2 \geq \frac{1}{3} + c_n$, so we have that $\lambda_{t,j} \leq 0$. Furthermore, by Lemma 8, if the step size $\eta$ satisfies $\eta \leq \frac{1}{36\left(1+\frac{1}{3}c_n\right)\max\{c_m,1\}}$, we have that $\|w_t\|^2 \leq \frac{10}{9} + \frac{1}{3}c_n$. Therefore, using the upper-bound of the step size and the norm, we can obtain that $\lambda_{t,j} := \eta(1 - 3\|w_t\|^2) + \eta\rho_{t,j} \geq -1$. Hence, we have completed the proof.

$\square$

## E    PROOF OF THEOREM 2

To prove Theorem 2, we will need the following lemma.

**Lemma 11.** *Fix any number $\delta > 0$. Define $T_\delta := \min\{t : \rho\|w_{t+1}\| \geq \gamma - \delta\}$. Assume that $\eta \leq \frac{1}{\|A\|_2 + \rho\|w_*\|}$. Suppose that $w_0^{(1)}b^{(1)} \leq 0$. Then, we have that $w_t^{(1)}b^{(1)} \leq 0$, for all $0 \leq t \leq T_\delta$.*

*Proof.* The lemma holds trivially when $\gamma \leq 0$, so let us assume $\gamma > 0$.

Recall the Heavy Ball generates the iterates as

$$w_{t+1}^{(1)} = (1 - \eta\lambda^{(1)}(A) - \rho\eta\|w_t\|)w_t^{(1)} - \eta b^{(1)} + \beta(w_t^{(1)} - w_{t-1}^{(1)}).$$

We are going to show that for all $t$,

$$w_t^{(1)}b^{(1)} \leq 0 \text{ and } (w_t^{(1)} - w_{t-1}^{(1)})b^{(1)} \leq -c_{bw}w_t^{(1)}b^{(1)}, \tag{57}$$

for any constant $c_{bw} \geq 0$. The initialization guarantees that $w_0^{(1)}b^{(1)} \leq 0$ and that $(w_0^{(1)} - w_{-1}^{(1)})b^{(1)} = 0 \leq -c_{bw}w_0^{(1)}b^{(1)}$. Suppose that (57) is true at iteration $t$. Consider iteration $t + 1$.

$$\begin{aligned}
w_{t+1}^{(1)}b^{(1)} &= (1 - \eta\lambda^{(1)}(A) - \rho\eta\|w_t\|)w_t^{(1)}b^{(1)} - \eta(b^{(1)})^2 + \beta(w_t^{(1)} - w_{t-1}^{(1)})b^{(1)} \\
&\leq (1 - \eta\lambda^{(1)}(A) - \rho\eta\|w_t\| - \beta c_{bw})w_t^{(1)}b^{(1)} - \eta(b^{(1)})^2 \\
&\leq 0,
\end{aligned} \tag{58}$$

where the first inequality is by induction at iteration $t$ and the second one is true if $(1 - \eta\lambda^{(1)}A - \rho\eta\|w_t\| - \beta c_{bw}) \geq 0$, which gives a constraints about $\eta$,

$$1 - \eta\lambda^{(1)}(A) - \rho\eta\|w_t\| \geq c_{bw}. \tag{59}$$

Now let us switch to show that $(w_{t+1}^{(1)} - w_t^{(1)})b^{(1)} \leq -c_{bw}w_{t+1}^{(1)}b^{(1)}$, which is equivalent to showing that $w_{t+1}^{(1)}b^{(1)} \leq \frac{1}{1+c_{bw}}w_t^{(1)}b^{(1)}$. From (58), it suffices to show that

$$(1 - \eta\lambda^{(1)}(A) - \rho\eta\|w_t\| - \beta c_{bw})w_t^{(1)}b^{(1)} - \eta(b^{(1)})^2 \leq \frac{1}{1+c_{bw}}w_t^{(1)}b^{(1)}. \tag{60}$$

Since $w_t^{(1)}b^{(1)} \leq 0$, a sufficient condition of the above inequality is

$$1 - \eta\lambda^{(1)}(A) - \rho\eta\|w_t\| - \beta c_{bw} - \frac{1}{1+c_{bw}} \geq 0. \tag{61}$$

Now using that $\rho\|w_t\| \leq \gamma - \delta$ and that $\frac{1}{1+x} \leq 1 - \frac{1}{2}x$ for $x \in [0, 1]$. It suffices to have that

$$1 - \eta\lambda^{(1)}(A) - \eta(\gamma-\delta) - \beta c_{bw} - \frac{1}{1+c_{bw}} \geq 1 + \eta\delta - \beta c_{bw} - 1 + \frac{1}{2}c_{bw} \geq \eta\delta - \frac{1}{2}c_{bw} \geq 0. \tag{62}$$

By setting $c_{bw} = 0$, we have that the inequality is satisfied. Substituting $c_{bw} = 0$ to (59), we have that

$$\eta \leq \frac{1}{\lambda^{(1)}(A) + \rho\|w_t\|},$$

Recall that $\rho\|w_t\| \leq \gamma - \delta$ for all $t \leq T_\delta$. So we have that $\eta \leq \frac{1}{\|A\|_2 + (\gamma-\delta)\cdot\mathbb{1}_{\gamma-\delta\geq 0}}$. Furthermore, by using that $\rho\|w_*\| > \gamma$, it suffices to have that $\eta \leq \frac{1}{\|A\|_2 + \rho\|w_*\|}$. We have completed the proof.

$\square$

Given Lemma 11, we are ready for proving Theorem 2.

*Proof.* (of Theorem 2) The lemma holds trivially when $\gamma \leq 0$, so let us assume $\gamma > 0$. Recall the notation that $w^{(1)}$ represents the projection of $w$ on the eigenvector $v_1$ of the least eigenvalue $\lambda^{(1)}(A)$, i.e. $w^{(1)} = \langle w, v_1 \rangle$. From the update rule, we have that

$$\frac{w_{t+1}^{(1)}}{-\eta b^{(1)}} = (I_d + \eta\gamma - \rho\eta\|w_t\|)\frac{w_t^{(1)}}{-\eta b^{(1)}} + 1 + \frac{\beta(w_t^{(1)} - w_{t-1}^{(1)})}{-\eta b^{(1)}}. \tag{63}$$

Denote $a_t := \frac{w_t^{(1)}}{-\eta b^{(1)}}$. We can rewrite (63) as

$$a_{t+1} = (1 + \eta\gamma - \rho\eta\|w_t\|)a_t + 1 + \beta(a_t - a_{t-1}) \geq (1 + \eta\delta)a_t + 1 + \beta(a_t - a_{t-1}), \quad (64)$$

where the inequality is due to that $\rho\|w_t\| \leq \gamma - \delta$ for $t \leq T_\delta$. Now we are going to show that, $a_{t+1} \geq (1 + \eta\delta + \frac{\eta\delta}{1+\eta\delta}\beta)a_t + 1$. For the above inequality to hold, it suffices to show that $a_t - a_{t-1} \geq \frac{\eta\delta}{1+\eta\delta}a_t$. That is, $a_t \geq \frac{1}{1-\eta\delta/(1+\eta\delta)}a_{t-1}$. The base case $t = 1$ holds because $a_1 \geq (1+\eta\delta)a_0 + 1 \geq \frac{1}{1-\eta\delta/(1+\eta\delta)}a_0$. Suppose that at iteration $t$, we have that $a_t \geq \frac{1}{1-\eta\delta/(1+\eta\delta)}a_{t-1}$. Consider $t + 1$, we have that $a_{t+1} \geq (1 + \eta\delta)a_t + 1 + \beta(a_t - a_{t-1}) \geq (1 + \eta\delta)a_t + 1 \geq \frac{1}{1-\eta\delta/(1+\eta\delta)}a_t$, where the second to last inequality is because $a_t \geq \frac{1}{1-\eta\delta/(1+\eta\delta)}a_{t-1}$ implies $a_t \geq a_{t-1}$. Therefore, we have completed the induction. So we have shown that $\frac{w_{t+1}^{(1)}}{-\eta b^{(1)}} = (I_d + \eta\gamma - \rho\eta\|w_t\|)\frac{w_t^{(1)}}{-\eta b^{(1)}} + 1 + \frac{\beta(w_t^{(1)} - w_{t-1}^{(1)})}{-\eta b^{(1)}} \geq (1 + \eta\delta + \frac{\eta\delta}{1+\eta\delta}\beta)\frac{w_t^{(1)}}{-\eta b^{(1)}} + 1$. Recursively expanding the inequality, we have that

$$\begin{aligned}
\frac{w_{t+1}^{(1)}}{-\eta b^{(1)}} &\geq (1 + \eta\delta + \frac{\eta\delta}{1+\eta\delta}\beta)\frac{w_t^{(1)}}{-\eta b^{(1)}} + 1 \geq (1 + \eta\delta + \frac{\eta\delta}{1+\eta\delta}\beta)^2 \frac{w_{t-1}^{(1)}}{-\eta b^{(1)}} + (1 + \eta\delta + \frac{\eta\delta}{1+\eta\delta}\beta) + 1 \\
&\geq \ldots \geq \frac{1}{\eta\delta(1+\frac{\beta}{1+\eta\delta})}\left((1 + \eta\delta + \frac{\eta\delta}{1+\eta\delta}\beta)^t - 1\right).
\end{aligned}$$

$$(65)$$

Therefore, $\frac{\gamma-\delta}{\rho} \overset{(a)}{\geq} \|w_{T_\delta}\| \geq |w_{T_\delta}^{(1)}| \overset{(b)}{\geq} \frac{|b^{(1)}|}{\delta+\delta\beta/(1+\eta\delta)}\left((1 + \eta\delta + \frac{\eta\delta}{1+\eta\delta}\beta)^{T_\delta} - 1\right)$ where (a) uses that for $t \leq T_\delta$, $\rho\|w_t\| \leq \gamma - \delta$, and (b) uses (65) and Lemma 11 that $w_t^{(1)}b^{(1)} \leq 0$. Consequently,

$$\begin{aligned}
T_\delta &\leq \frac{\log\left(1 + \frac{(\gamma-\delta)(\delta+\delta\beta/(1+\eta\delta))}{\rho|b^{(1)}|}\right)}{\log(1+\eta\delta+\frac{\eta\delta}{1+\eta\delta}\beta)} \overset{(a)}{\leq} \frac{2}{\eta\delta(1+\beta/(1+\eta\delta))}\log\left(1 + \frac{(\gamma-\delta)\delta(1+\beta/(1+\eta\delta))}{\rho|b^{(1)}|}\right) \\
&\overset{(b)}{\leq} \frac{2}{\eta\delta(1+\beta/(1+\eta\delta))}\log\left(1 + \frac{\gamma_+^2(1+\beta/(1+\eta\delta))}{4\rho|b^{(1)}|}\right),
\end{aligned}$$

$$(66)$$

where (a) uses that $\log(1 + x) \geq \frac{x}{2}$ for any $x \in [0, \sim 2.51]$, and (b) uses $\gamma\delta - \delta^2 \leq \frac{\gamma_+^2}{4}$. $\qquad\square$

## F  CONVERGENCE OF HB FOR THE CUBIC-REGULARIZED PROBLEM

**Theorem 3.** *Assume that the iterate stays in the benign region that exhibits one-point strong convexity to $w_*$, i.e. $\mathbb{B} := \{w \in \mathbb{R}^d : \rho\|w\| \geq \gamma - \delta\}$ for a number $\delta > 0$, once it enters the benign region. Denote $c^{converge} := 1 - \frac{2\tilde{c}_0}{1-2\eta c_\delta\left(\rho\|w_*\|-\gamma\right)}$, where $c_\delta$ and $\tilde{c}_0$ are some constants that satisfy $0.5 > c_\delta > 0$ and $\tilde{c}_0 > 0$. Suppose that there is a number $R$, such that the size of the iterate satisfies $\|w_t\| \leq R$ for all $t$ during the execution of the algorithm and that $\|w_*\| \leq R$. Denote $L := \|A\|_2 + 2\rho R$. Also, suppose that the momentum parameter $\beta \in [0, 0.65]$ and that the step size $\eta$ satisfies $\eta \leq \min\left(\frac{1}{4(\|A_*\|+26\rho R)}, \frac{\tilde{c}_0 c_\delta\left(\rho\|w_*\|-\gamma\right)}{(50L+2(26)^2)(\|A_*\|+\rho R)(1+\|A_*\|+\rho R)+1.3L}\right)$. Set $w_0 = w_{-1} = -r\frac{b}{\|b\|}$ for any sufficiently small $r > 0$. Then, in the benign region it takes at most*

$$t \lesssim \hat{T} := \frac{1}{\eta c_\delta(\rho\|w_*\|-\gamma)(1+ \boxed{\beta c^{converge}})}\log\left(\frac{(\|A\|_2+2\rho R)\tilde{c}}{2\epsilon}\right)$$

*number of iterations to reach an $\epsilon$-approximate error, where $\tilde{c} := 4R^2(1 + \eta\tilde{C})$ with $\tilde{C} = \frac{4}{3}\tilde{c}_0(\rho\|w_*\| - \gamma) + 3\left(\tilde{c}_0 c_\delta(\rho\|w_*\| - \gamma) + 10\max\{0, \gamma\}\right)$.*

Note that the constraint of $\eta$ ensures that $\eta c_\delta\left(\rho\|w_*\| - \gamma\right) \leq \frac{1}{8}$; as a consequence, $c^{converge} \leq 1$. Theorem 3 indicates that up to an upper-threshold, a larger value of $\beta$ reduces the number of iterations to linearly converge to an $\epsilon$-optimal point, and hence leads to a faster convergence.

Let us now make a few remarks. First, we want to emphasize that in the linear convergence rate regime, a constant factor improvement of the convergence rate (i.e. of $\hat{T}$ here) means that the slope of the curve in the log plot of optimization value vs. iteration is steeper. Our experimental result

(Figure 2) confirms this. In this figure, we can see that the curve corresponds to a larger momentum parameter $\beta$ has a steeper slope than that of the smaller ones. The slope is steeper as $\beta$ increases, which justifies the effectiveness of the momentum in the linear convergence regime. Our theoretical result also indicates that the acceleration due to the use of momentum is more evident for a small step size $\eta$. When $\eta$ is sufficiently small, the number $c_{converge}$ are close to 1, which means that the number of iterations can be reduced approximately by a factor of $1 + \beta$.

Secondly, from the theorem, one will need $|b^{(1)}|$ be non-zero, which can be guaranteed with a high probability by adding some Gaussian perturbation on $b$. We refer the readers to Section 4.2 of Carmon & Duchi (2019) for the technique.

To prove Theorem 3, we will need a series of lemmas which is given in the following subsection.

### F.1 Some supporting lemmas

**Lemma 12.** *Denote sequences* $d_t := \frac{\rho}{2}(\|w_*\| - \|w_t\|)\|w_t - w_*\|^2$, $e_t := -(w_t - w_*)^\top(A_* - 2\eta\tilde{c}_w A_*^2)(w_t - w_*)$, $g_t := -\frac{\rho}{2}(\|w_*\| - \|w_t\|)^2(\|w_*\| + \|w_t\| - 4\eta\rho\tilde{c}_w\|w_t\|^2)$, *where* $c_w := \frac{2\beta}{1-\beta}$ *and* $\tilde{c}_w := (1 + c_w) + (1 + c_w)^2$. *For all $t$, we have that*

$$\langle w_t - w_*, w_t - w_{t-1}\rangle + c_w\|w_t - w_{t-1}\|^2 \le \eta \sum_{s=1}^{t-1} \beta^{t-1-s}(d_s + e_s + g_s).$$

*Proof.* We use induction for the proof. The base case $t = 0$ holds, because $w_0 = w_{-1}$ by initialization and both sides of the inequality is 0. Let us assume that it holds at iteration $t$. That is, $\langle w_t - w_*, w_t - w_{t-1}\rangle + c_w\|w_t - w_{t-1}\|^2 \le \eta \sum_{s=1}^{t-1} \beta^{t-1-s}(d_s + e_s + g_s)$. Consider iteration $t+1$. We want to prove that

$$\langle w_{t+1} - w_*, w_{t+1} - w_t\rangle + c_w\|w_{t+1} - w_t\|^2 \le \eta \sum_{s=1}^{t} \beta^{t-s}(d_s + e_s + g_s).$$

Denote $\Delta := w_{t+1} - w_t$. It is equivalent to showing that $\langle \Delta, w_t + \Delta - w_*\rangle + c_w\|\Delta\|^2 \le \eta \sum_{s=1}^{t} \beta^{t-s}(d_s + e_s + g_s)$, or

$$\langle -\eta\nabla f(w_t) + \beta(w_t - w_{t-1}), w_t - w_* - \eta\nabla f(w_t) + \beta(w_t - w_{t-1})\rangle$$
$$+ c_w\| - \eta\nabla f(w_t) + \beta(w_t - w_{t-1})\|^2 \le \eta \sum_{s=1}^{t} \beta^{t-s}(d_s + e_s + g_s). \tag{67}$$

which is in turn equivalent to showing that

$$-\eta \underbrace{\langle \nabla f(w_t), w_t - w_*\rangle}_{(a)} + \eta^2(1 + c_w)\underbrace{\|\nabla f(w_t)\|^2}_{(b)} \underbrace{-2\eta\beta(1 + c_w)\langle \nabla f(w_t), w_t - w_{t-1}\rangle}_{(c)}$$

$$+ \beta\langle w_t - w_{t-1}, w_t - w_*\rangle + \beta^2\|w_t - w_{t-1}\|^2 + c_w\beta^2\|w_t - w_{t-1}\|^2 \tag{68}$$

$$\le \eta \sum_{s=1}^{t} \beta^{t-s}(d_s + e_s + g_s).$$

For term (a), we have that

$\langle w_t - w_*, \nabla f(w_t)\rangle$

$= (w_t - w_*)^\top A_*(w_t - w_*) + \rho(\|w_t\| - \|w_*\|)(\|w_t\|^2 - w_*^\top w_t)$

$= (w_t - w_*)^\top\big(A_* + \frac{\rho}{2}(\|w_t\| - \|w_*\|)I_d\big)(w_t - w_*) + \frac{\rho}{2}(\|w_*\| - \|w_t\|)^2(\|w_t\| + \|w_*\|).$

$= (w_t - w_*)^\top A_*(w_t - w_*) + \frac{\rho}{2}(\|w_t\| - \|w_*\|)\|w_t - w_*\|^2 + \frac{\rho}{2}(\|w_*\| - \|w_t\|)^2(\|w_t\| + \|w_*\|). \tag{69}$

Notice that $A_* \succeq -\gamma + \rho\|w_*\|I_d \succeq 0I_d$, as $\rho\|w_*\| \ge \gamma$. On the other hand, for term (b), we get that

$$\|\nabla f(w_t)\|^2 = \|A_*(w_t - w_*) - \rho(\|w_*\| - \|w_t\|)w_t\|^2$$
$$\le 2(w_t - w_*)^\top A_*^2(w_t - w_*) + 2\rho^2(\|w_*\| - \|w_t\|)^2\|w_t\|^2. \tag{70}$$

For term (c), we can bound it as

$$
\begin{aligned}
&- 2\eta\beta(1+c_w)\langle \nabla f(w_t), w_t - w_{t-1}\rangle \leq \|\sqrt{2}\eta(1+c_w)\nabla f(w_t)\|\|\sqrt{2}\beta(w_t - w_{t-1})\| \\
&\leq \frac{1}{2}\big(2\eta^2(1+c_w)^2\|\nabla f(w_t)\|^2 + 2\beta^2\|w_t - w_{t-1}\|^2\big) \\
&= \eta^2(1+c_w)^2\|\nabla f(w_t)\|^2 + \beta^2\|w_t - w_{t-1}\|^2.
\end{aligned}
\tag{71}
$$

Combining the above, we have that

$$
\begin{aligned}
&- \eta\langle \nabla f(w_t), w_t - w_*\rangle + \eta^2(1+c_w)\|\nabla f(w_t)\|^2 - 2\eta\beta(1+c_w)\langle \nabla f(w_t), w_t - w_{t-1}\rangle \\
&+ \beta\langle w_t - w_{t-1}, w_t - w_*\rangle + \beta^2\|w_t - w_{t-1}\|^2 + c_w\beta^2\|w_t - w_{t-1}\|^2 \\
&\leq -\eta\langle \nabla f(w_t), w_t - w_*\rangle + \eta^2\tilde{c}_w\|\nabla f(w_t)\|^2 + \beta\langle w_t - w_{t-1}, w_t - w_*\rangle + \beta^2(2+c_w)\|w_t - w_{t-1}\|^2 \\
&\leq -\eta(w_t - w_*)^\top\big(A_* - 2\eta\tilde{c}_w A_*^2\big)(w_t - w_*) \\
&- \eta\frac{\rho}{2}(\|w_*\| - \|w_t\|)^2\big(\|w_*\| + \|w_t\| - 4\eta\rho\tilde{c}_w\|w_t\|^2\big) + \eta\frac{\rho}{2}(\|w_*\| - \|w_t\|)\|w_t - w_*\|^2 \\
&+ \underbrace{\beta\langle w_t - w_{t-1}, w_t - w_*\rangle + \beta^2(2+c_w)\|w_t - w_{t-1}\|^2}_{=\beta\langle w_t - w_{t-1}, w_t - w_*\rangle + \beta c_w\|w_t - w_{t-1}\|^2} \\
&\leq \eta(d_t + e_t + g_t) + \eta\beta\sum_{s=1}^{t-1}\beta^{t-1-s}(d_s + e_s + g_s) := \eta\sum_{s=1}^{t}\beta^{t-s}(d_s + e_s + g_s).
\end{aligned}
\tag{72}
$$

where in the second to last inequality we used $\beta(2 + c_w) = c_w$ and the last inequality we used the assumption at iteration $t$. So we have completed the induction.

$\square$

**Lemma 13.** *Assume that for all $t$, $\|w_t\| \leq R$ for some number $R$. Following the notations used in Lemma 12, we denote sequences $d_t := \frac{\rho}{2}(\|w_*\| - \|w_t\|)\|w_t - w_*\|^2$, $e_t := -(w_t - w_*)^\top(A_* - 2\eta\tilde{c}_w A_*^2)(w_t - w_*)$, $g_t := -\frac{\rho}{2}(\|w_*\| - \|w_t\|)^2(\|w_*\| + \|w_t\| - 4\eta\rho\tilde{c}_w\|w_t\|^2)$, where $c_w := \frac{2\beta}{1-\beta}$ and $\tilde{c}_w := (1 + c_w) + (1 + c_w)^2$. Let us also denote $c_\beta := (2\beta^2 + 4\beta + L\beta)$, $L := \|A\|_2 + 2\rho R$, and $z_t := \sum_{s=0}^{t}\beta^{t-s+1}\nabla f(w_s)$. If $\eta$ satisfies: (1) $\eta \leq \frac{1+\beta}{2\rho R}$, and (2) $\eta \leq \frac{1+\beta}{4\|A_*\|_2}$, then we have that for all $t$,*

$$
\begin{aligned}
&\|w_{t+1} - w_*\|^2 \\
&\leq \big(1 - \eta(1+\beta)\big[\rho\|w_t\| - (\gamma - \frac{\rho\|w_*\| - \gamma}{2})\big]\big)\|w_t - w_*\|^2 \\
&+ \eta^2(w_{t-1} - w_*)^\top(2c_\beta A_*^2 + 2\beta^2 LI_d + 2c_\beta\rho^2 R^2 I_d)(w_{t-1} - w_*) + \eta^2(c_\beta - 2c_w)\|z_{t-2}\|^2 \\
&+ 2\eta\beta^2\sum_{s=0}^{t-2}\beta^{t-2-s}(d_s + e_s + g_s).
\end{aligned}
\tag{73}
$$

*Proof.* Recall that the update of heavy ball is

$$
w_{t+1} = w_t - \eta\nabla f(w_t) + \beta(w_t - w_{t-1}).
\tag{74}
$$

So the distance term can be decomposed as

$$
\begin{aligned}
\|w_{t+1} - w_*\|^2 = &\|w_t - \eta\nabla f(w_t) + \beta(w_t - w_{t-1}) - w_*\|^2 \\
= &\|w_t - w_*\|^2 - 2\eta\langle w_t - w_*, \nabla f(w_t)\rangle + \eta^2\|\nabla f(w_t)\|^2 \\
&+ \beta^2\|w_t - w_{t-1}\|^2 - 2\eta\beta(w_t - w_{t-1})^\top\nabla f(w_t) + 2\beta(w_t - w_*)^\top(w_t - w_{t-1}) \\
= &\|w_t - w_*\|^2 - 2\eta\langle w_t - w_*, \nabla f(w_t)\rangle + \eta^2\|\nabla f(w_t)\|^2 \\
&+ \beta^2\|w_t - w_{t-1}\|^2 - 2\eta\beta\langle w_t - w_*, \nabla f(w_t)\rangle - 2\eta\beta\langle w_* - w_{t-1}, \nabla f(w_t)\rangle \\
&+ 2\beta(w_t - w_*)^\top(w_t - w_{t-1}) \\
= &\|w_t - w_*\|^2 - 2\eta(1 + \beta)\underbrace{\langle w_t - w_*, \nabla f(w_t)\rangle}_{(a)} + \eta^2\underbrace{\|\nabla f(w_t)\|^2}_{(b)} \\
&+ (\beta^2 + 2\beta)\|w_t - w_{t-1}\|^2 - 2\eta\beta\langle w_* - w_{t-1}, \nabla f(w_t)\rangle + 2\beta(w_{t-1} - w_*)^\top(w_t - w_{t-1}).
\end{aligned}
\tag{75}
$$

For term (a), $\langle w_t - w_*, \nabla f(w_t)\rangle$, by using that $\nabla f(w) = A_*(w - w_*) - \rho(\|w_*\| - \|w\|)w$, we can bound it as

$$
\begin{aligned}
&\langle w_t - w_*, \nabla f(w_t)\rangle \\
&= (w_t - w_*)^\top A_*(w_t - w_*) + \rho(\|w_t\| - \|w_*\|)(\|w_t\|^2 - w_*^\top w_t) \\
&= (w_t - w_*)^\top\big(A_* + \frac{\rho}{2}(\|w_t\| - \|w_*\|)I_d\big)(w_t - w_*) + \frac{\rho}{2}\big(\|w_*\| - \|w_t\|\big)^2(\|w_t\| + \|w_*\|).
\end{aligned}
\tag{76}
$$

On the other hand, for term (b), $\|\nabla f(w_t)\|^2$, we get that

$$
\begin{aligned}
\|\nabla f(w_t)\|^2 &= \|A_*(w_t - w_*) - \rho(\|w_*\| - \|w_t\|)w_t\|^2 \\
&\leq 2(w_t - w_*)^\top A_*^2(w_t - w_*) + 2\rho^2\big(\|w_*\| - \|w_t\|\big)^2\|w_t\|^2.
\end{aligned}
\tag{77}
$$

By combining (75),(76), and (77), we can bound the distance term as

$$
\begin{aligned}
&\|w_{t+1} - w_*\|^2 = \|w_t - w_*\|^2 - 2\eta(1 + \beta)\langle w_t - w_*, \nabla f(w_t)\rangle + \eta^2\|\nabla f(w_t)\| \\
&+ (\beta^2 + 2\beta)\|w_t - w_{t-1}\|^2 - 2\eta\beta\langle w_* - w_{t-1}, \nabla f(w_t)\rangle + 2\beta(w_{t-1} - w_*)^\top(w_t - w_{t-1}) \\
&\leq \|w_t - w_*\|^2 \\
&- 2\eta(1 + \beta)\big((w_t - w_*)^\top\big(A_* + \frac{\rho}{2}(\|w_t\| - \|w_*\|)I_d\big)(w_t - w_*)\big) \\
&- \rho\eta(1 + \beta)\big(\|w_*\| - \|w_t\|\big)^2(\|w_t\| + \|w_*\|) \\
&+ \eta^2\big(2(w_t - w_*)^\top A_*^2(w_t - w_*) + 2\rho^2\big(\|w_*\| - \|w_t\|\big)^2\|w_t\|^2\big) \\
&+ (\beta^2 + 2\beta)\|w_t - w_{t-1}\|^2 - 2\eta\beta\langle w_* - w_{t-1}, \nabla f(w_t)\rangle + 2\beta(w_{t-1} - w_*)^\top(w_t - w_{t-1}).
\end{aligned}
\tag{78}
$$

Now let us bound the terms on the last line of (78). For the second to the last term, it is equal to

$$
\begin{aligned}
&- 2\eta\beta\langle w_* - w_{t-1}, \nabla f(w_t)\rangle \\
&= -2\eta\beta\langle w_* - w_{t-1}, \nabla f(w_{t-1})\rangle - 2\eta\beta\langle w_* - w_{t-1}, \nabla f(w_t) - \nabla f(w_{t-1})\rangle,
\end{aligned}
\tag{79}
$$

On the other hand, the last term of (78) is equal to

$$
2\beta(w_{t-1} - w_*)^\top(w_t - w_{t-1}) = -2\beta\eta\langle w_{t-1} - w_*, \nabla f(w_{t-1})\rangle + 2\beta^2\langle w_{t-1} - w_*, w_{t-1} - w_{t-2}\rangle.
\tag{80}
$$

By Lemma 12, for all $t$, we have that

$$
\langle w_{t-1} - w_*, w_{t-1} - w_{t-2}\rangle \leq -c_w\|w_{t-1} - w_{t-2}\|^2 + \eta\sum_{s=1}^{t-2}\beta^{t-2-s}(d_s + e_s + g_s).
\tag{81}
$$

Therefore, by combining (79), (80), and (81), we have that

$$
\begin{aligned}
&(\beta^2 + 2\beta)\|w_t - w_{t-1}\|^2 - 2\eta\beta\langle w_* - w_{t-1}, \nabla f(w_t)\rangle + 2\beta(w_{t-1} - w_*)^\top(w_t - w_{t-1}) \\
&\leq (\beta^2 + 2\beta)\|w_t - w_{t-1}\|^2 - 2\eta\beta\langle w_{t-1} - w_*, \nabla f(w_{t-1}) - \nabla f(w_t)\rangle - 2c_w\beta^2\|w_{t-1} - w_{t-2}\|^2 \\
&+ 2\eta\beta^2\sum_{s=1}^{t-2}\beta^{t-2-s}(d_s + e_s + g_s).
\end{aligned}
\tag{82}
$$

Combining (78) and (82) leads to the following,

$$
\begin{aligned}
\|w_{t+1} - w_*\|^2 \leq{} & \|w_t - w_*\|^2 - 2\eta(1+\beta)\big((w_t - w_*)^\top \big(A_* + \tfrac{\rho}{2}(\|w_t\| - \|w_*\|)I_d\big)(w_t - w_*)\big) \\
& - \rho\eta(1+\beta)\big(\|w_*\| - \|w_t\|\big)^2(\|w_t\| + \|w_*\|) \\
& + \eta^2\big(2(w_t - w_*)^\top A_*^2(w_t - w_*) + 2\rho^2\big(\|w_*\| - \|w_t\|\big)^2\|w_t\|^2\big) \\
& + (\beta^2 + 2\beta)\|w_t - w_{t-1}\|^2 - 2\eta\beta\langle w_{t-1} - w_*, \nabla f(w_{t-1}) - \nabla f(w_t)\rangle \\
& - 2c_w\beta^2\|w_{t-1} - w_{t-2}\|^2 + 2\eta\beta^2\sum_{s=1}^{t-2}\beta^{t-2-s}(d_s + e_s + g_s).
\end{aligned}
\tag{83}
$$

Let us now bound the terms on the second to the last line (83) above,

$$
(\beta^2 + 2\beta)\|w_t - w_{t-1}\|^2 - 2\eta\beta\langle w_{t-1} - w_*, \nabla f(w_{t-1}) - \nabla f(w_t)\rangle.
\tag{84}
$$

First, note that $\|\nabla^2 f(w)\| \leq \|A\|_2 + 2\rho\|w\|$. So we know that $f$ is $L := \|A\|_2 + 2\rho R$ smooth on $\{w : \|w\| \leq R\}$. Second, denote

$$
z_t := \sum_{s=0}^{t} \beta^{t-s+1}\nabla f(w_s),
\tag{85}
$$

we can bound $\|w_t - w_{t-1}\|$ as

$$
\begin{aligned}
\|w_t - w_{t-1}\| &= \| -\eta\sum_{s=0}^{t-1}\beta^{t-1-s}\nabla f(w_s)\| \leq \eta\|\nabla f(w_{t-1})\| + \eta\|\sum_{s=0}^{t-2}\beta^{t-1-s}\nabla f(w_s)\| \\
&:= \eta\|\nabla f(w_{t-1})\| + \eta\|z_{t-2}\|, \\
\|w_t - w_{t-1}\|^2 &= \| -\eta\sum_{s=0}^{t-1}\beta^{t-1-s}\nabla f(w_s)\|^2 \leq 2\eta^2\|\nabla f(w_{t-1})\|^2 + 2\eta^2\|\sum_{s=0}^{t-2}\beta^{t-1-s}\nabla f(w_s)\|^2 \\
&:= 2\eta^2\|\nabla f(w_{t-1})\|^2 + 2\eta^2\|z_{t-2}\|^2,
\end{aligned}
\tag{86}
$$

Using the results, we can bound (84) as

$$
\begin{aligned}
&(\beta^2 + 2\beta)\|w_t - w_{t-1}\|^2 - 2\eta\beta\langle w_{t-1} - w_*, \nabla f(w_{t-1}) - \nabla f(w_t)\rangle \\
&\overset{(86)}{\leq} 2\eta^2(\beta^2 + 2\beta)(\|\nabla f(w_{t-1})\|^2 + \|z_{t-2}\|^2) + 2\eta^2\beta L\|w_{t-1} - w_*\|(\|\nabla f(w_{t-1})\| + \|z_{t-2}\|) \\
&\leq 2\eta^2(\beta^2 + 2\beta)(\|\nabla f(w_{t-1})\|^2 + \|z_{t-2}\|^2) \\
&\quad + \eta^2\beta L(\|w_{t-1} - w_*\|^2 + \|\nabla f(w_{t-1})\|^2) + \eta^2\beta L(\|w_{t-1} - w_*\|^2 + \|z_{t-2}\|^2) \\
&= 2\eta^2(\beta^2 + 2\beta)(\|\nabla f(w_{t-1})\|^2 + \|z_{t-2}\|^2) + 2\eta^2\beta L\|w_{t-1} - w_*\|^2 + \eta^2\beta L\|\nabla f(w_{t-1})\|^2 \\
&\quad + \eta^2\beta L\|z_{t-2}\|^2.
\end{aligned}
\tag{87}
$$

Note that

$$
\begin{aligned}
\|\nabla f(w_{t-1})\|^2 &= \|A_*(w_{t-1} - w_*) - \rho(\|w_*\| - \|w_{t-1}\|)w_{t-1}\|^2 \\
&\leq 2(w_{t-1} - w_*)^\top A_*^2(w_{t-1} - w_*) + 2\rho^2\big(\|w_*\| - \|w_{t-1}\|\big)^2\|w_{t-1}\|^2.
\end{aligned}
\tag{88}
$$

Denote

$$
c_\beta := (2\beta^2 + 4\beta + L\beta).
\tag{89}
$$

By (87), (88), we have that

$$
\begin{aligned}
&(\beta^2 + 2\beta)\|w_t - w_{t-1}\|^2 - 2\eta\beta\langle w_{t-1} - w_*, \nabla f(w_{t-1}) - \nabla f(w_t)\rangle. \\
&\leq 2\eta^2 c_\beta(w_{t-1} - w_*)^\top A_*^2(w_{t-1} - w_*) + 2\eta^2 c_\beta\rho^2\big(\|w_*\| - \|w_{t-1}\|\big)^2\|w_{t-1}\|^2 + 2\eta^2\beta^2 L\|w_{t-1} - w_*\|^2 \\
&\quad + \eta^2 c_\beta\|z_{t-2}\|^2.
\end{aligned}
\tag{90}
$$

Let us summarize the results so far, by (83) and (90), we have that

$$\|w_{t+1} - w_*\|^2$$

$$\leq \|w_t - w_*\|^2 - 2\eta(1+\beta)\big((w_t - w_*)^\top \big(A_* + \frac{\rho}{2}(\|w_t\| - \|w_*\|)I_d\big)(w_t - w_*)\big)$$

$$- \rho\eta(1+\beta)\big(\|w_*\| - \|w_t\|\big)^2\big(\|w_t\| + \|w_*\|\big)$$

$$+ \eta^2\big(2(w_t - w_*)^\top A_*^2(w_t - w_*) + 2\rho^2\big(\|w_*\| - \|w_t\|\big)^2\|w_t\|^2\big)$$

$$+ 2\eta^2 c_\beta(w_{t-1} - w_*)^\top A_*^2(w_{t-1} - w_*) + 2\eta^2 c_\beta \rho^2\big(\|w_*\| - \|w_{t-1}\|\big)^2\|w_{t-1}\|^2 + 2\eta^2\beta^2 L\|w_{t-1} - w_*\|^2$$

$$+ \eta^2(c_\beta - 2c_w)\|z_{t-2}\|^2 + 2\eta\beta^2\sum_{s=1}^{t-2}\beta^{t-2-s}(d_s + e_s + g_s),$$

(91)

where we also used that $w_{t-1} - w_{t-2} = -\frac{\eta}{\beta}z_{t-2}$. We can rewrite the inequality above further as

$$\|w_{t+1} - w_*\|^2$$

$$\leq (w_t - w_*)^\top\big(I_d - 2\eta(1+\beta)A_*(I_d - \frac{\eta}{1+\beta}A_*) - \eta\rho(1+\beta)(\|w_t\| - \|w_*\|)I_d\big)(w_t - w_*)$$

$$\underbrace{-\eta\rho(1+\beta)(\|w_*\| - \|w_t\|)^2\big(\|w_t\|(1 - \frac{2\eta\rho\|w_t\|}{1+\beta}) + \|w_*\|\big)}_{\leq 0} + 2\eta^2 c_\beta\rho^2\big(\|w_*\| - \|w_{t-1}\|\big)^2\|w_{t-1}\|^2$$

$$+ \eta^2(w_{t-1} - w_*)^\top(2c_\beta A_*^2 + 2\beta^2 LI_d)(w_{t-1} - w_*)$$

$$+ \eta^2(c_\beta - 2c_w)\|z_{t-2}\|^2 + 2\eta\beta^2\sum_{s=1}^{t-2}\beta^{t-2-s}(d_s + e_s + g_s),$$

(92)

where we used that $-\eta\rho(1+\beta)(\|w_*\| - \|w_t\|)^2\big(\|w_t\|(1 - \frac{2\eta\rho\|w_t\|}{1+\beta}) + \|w_*\|\big) \leq 0$ for any $\eta$ that satisfies

$$\eta \leq \frac{1+\beta}{2\rho R},$$

(93)

as $\|w_t\| \leq R$ for all $t$. Let us simplify the inequality (92) further by writing it as

$$\|w_{t+1} - w_*\|^2$$

$$\overset{(a)}{\leq} (w_t - w_*)^\top\big(I_d - 2\eta(1+\beta)A_*(I_d - \frac{\eta}{1+\beta}A_*) - \eta\rho(1+\beta)(\|w_t\| - \|w_*\|)I_d\big)(w_t - w_*)$$

$$+ \eta^2(w_{t-1} - w_*)^\top(2c_\beta A_*^2 + 2\beta^2 LI_d + 2c_\beta\rho^2 R^2 I_d)(w_{t-1} - w_*) + \eta^2(c_\beta - 2c_w)\|z_{t-2}\|^2$$

$$+ 2\eta\beta^2\sum_{s=1}^{t-2}\beta^{t-2-s}(d_s + e_s + g_s).$$

$$\overset{(b)}{\leq} \big(1 - \eta(1+\beta)\big[\rho\|w_t\| - (\gamma - \frac{\rho\|w_*\| - \gamma}{2})\big]\big)\|w_t - w_*\|^2$$

$$+ \eta^2(w_{t-1} - w_*)^\top(2c_\beta A_*^2 + 2\beta^2 LI_d + 2c_\beta\rho^2 R^2 I_d)(w_{t-1} - w_*) + \eta^2(c_\beta - 2c_w)\|z_{t-2}\|^2$$

$$+ 2\eta\beta^2\sum_{s=1}^{t-2}\beta^{t-2-s}(d_s + e_s + g_s),$$

(94)

where (a) is because that $(\|w_*\| - \|w_{t-1}\|)^2\|w_{t-1}\|^2 \leq \|w_* - w_{t-1}\|^2\|w_{t-1}\|^2 \leq \|w_* - w_{t-1}\|^2 R^2$ as $\|w_t\| \leq R$ for all $t$, while (b) is by another constraint of $\eta$,

$$\eta \leq \frac{1+\beta}{4\|A_*\|_2},$$

(95)

so that $2\eta A_*(I_d - \frac{\eta}{1+\beta}A_*) \succeq \frac{3}{2}\eta A_* \succeq \frac{3}{2}\eta(-\gamma + \rho\|w_*\|)I_d$.

$\square$

**Lemma 14.** *Fix some numbers $c_0, c_1 > 0$. Denote $\omega^\beta_{t-2} := \sum_{s=0}^{t-2} \beta$. Following the notations and assumptions used in Lemma 12 and Lemma 13. If $\eta$ satisfies: (1) $\eta \leq \frac{c_0\beta}{2c_\beta\|A_*\|_2^2 + 2\beta^2 L + 2c_\beta\rho^2 R^2}$, (2)*

*$\eta \leq \frac{c_1}{\beta(2\tilde{c}_w + L\beta\omega^\beta_{t-2})\|A_*\|}$, (3) $\eta \leq \frac{c_1}{\rho\beta^2 L\omega^\beta_{t-2}R}$, (4) $\eta \leq \frac{1}{4\rho\tilde{c}_w R}$, and (5) $\eta \leq \frac{c_1\left(\|A_*\| + \rho R\right)}{L\beta^2\omega^\beta_{t-2}(\|A_*\|^2 + \rho^2 R^2)}$ for all $t$, then we have that for all $t$,*

$$\|w_{t+1} - w_*\|^2 \leq \left(1 - \eta(1 + \beta)\left[\rho\|w_t\| - (\gamma - \frac{\rho\|w_*\| - \gamma}{2})\right]\right)\|w_t - w_*\|^2 + \eta\beta c_0\|w_{t-1} - w_*\|^2$$

$$+ 2\eta\beta c_1\left(\|A_*\| + \rho R\right)\sum_{s=0}^{t-2}\beta^{t-2-s}\|w_s - w_*\|^2$$

$$- 2\eta\beta^2\sum_{s=1}^{t-2}\beta^{t-2-s}(w_s - w_*)^\top\left(A_* - \frac{\rho}{2}\left(\|w_*\| - \|w_t\|\right)I_d\right)(w_s - w_*).$$

*Proof.* From Lemma 13, we have that

$$\|w_{t+1} - w_*\|^2$$

$$\leq \left(1 - \eta(1 + \beta)\left[\rho\|w_t\| - (\gamma - \frac{\rho\|w_*\| - \gamma}{2})\right]\right)\|w_t - w_*\|^2$$

$$+ \eta^2(w_{t-1} - w_*)^\top(2c_\beta A_*^2 + 2\beta^2 LI_d + 2c_\beta\rho^2 R^2 I_d)(w_{t-1} - w_*) + \eta^2(c_\beta - 2c_w)\|z_{t-2}\|^2$$

$$+ 2\eta\beta^2\sum_{s=1}^{t-2}\beta^{t-2-s}(d_s + e_s + g_s),$$

$$\overset{(a)}{\leq} \left(1 - \eta(1 + \beta)\left[\rho\|w_t\| - (\gamma - \frac{\rho\|w_*\| - \gamma}{2})\right]\right)\|w_t - w_*\|^2 + \eta\beta c_0\|w_{t-1} - w_*\|^2 + \eta^2(c_\beta - 2c_w)\|z_{t-2}\|^2$$

$$+ 2\eta\beta^2\sum_{s=1}^{t-2}\beta^{t-2-s}(d_s + e_s + g_s),$$

$$\overset{(b)}{\leq} \left(1 - \eta(1 + \beta)\left[\rho\|w_t\| - (\gamma - \frac{\rho\|w_*\| - \gamma}{2})\right]\right)\|w_t - w_*\|^2 + \eta\beta c_0\|w_{t-1} - w_*\|^2$$

$$+ 2\eta^2(c_\beta - 2c_w)\beta^2\omega^\beta_{t-2}\sum_{s=0}^{t-2}\beta^{t-2-s}(w_s - w_*)^\top(A_*^2 + \rho^2 R^2 I_d)(w_s - w_*)$$

$$+ 2\eta\beta^2\sum_{s=1}^{t-2}\beta^{t-2-s}(d_s + e_s + g_s),$$

$$\tag{96}$$

where (a) is by a constraint of $\eta$ so that $\eta(2c_\beta\|A_*\|_2^2 + 2\beta^2 L + 2c_\beta\rho^2 R^2) \leq c_0\beta$, and (b) is due to the following, denote $\omega^\beta_t := \sum_{s=0}^t \beta$, we have that

$$\|z_t\|^2 := \|\sum_{s=0}^t \beta^{t-s+1}\nabla f(w_s)\|^2 = \beta^2(\omega^\beta_t)^2\|\sum_{s=0}^t \frac{\beta^{t-s}}{\omega^\beta_t}\nabla f(w_s)\|^2$$

$$\leq \beta^2(\omega^\beta_t)^2\left(\sum_{s=0}^t \frac{\beta^{t-s}}{\omega^\beta_t}\|\nabla f(w_s)\|^2\right) \tag{97}$$

$$\leq 2\beta^2\omega^\beta_t\left(\sum_{s=0}^t \beta^{t-s}(w_s - w_*)^\top(A_*^2 + \rho^2 R^2 I_d)(w_s - w_*)\right).$$

where the first inequality of (97) is due to Jensen's inequality and the second inequality of (97) is due to the following upper-bound of the gradient norm, $\|\nabla f(w_s)\|^2 = \|A_*(w_s - w_*) - \rho(\|w_*\| - \|w_s\|)w_s\|^2 \leq 2(w_s - w_*)^\top A_*^2(w_s - w_*) + 2\rho^2(\|w_*\| - \|w_s\|)^2\|w_s\|^2 \leq 2(w_s - w_*)^\top A_*^2(w_s -$

$w_*) + 2\rho^2 \|w_* - w_s\|^2 R^2$. By the definitions of $d_s, e_s, g_s$, we further have that

$$
\begin{aligned}
\|w_{t+1} - w_*\|^2 \leq & \left(1 - \eta(1+\beta)\left[\rho\|w_t\| - (\gamma - \frac{\rho\|w_*\| - \gamma}{2})\right]\right)\|w_s - w_*\|^2 + \eta\beta c_0\|w_{t-1} - w_*\|^2 \\
& - 2\eta\beta^2 \sum_{s=1}^{t-2} \beta^{t-2-s}(w_s - w_*)^\top \left(A_* - \frac{\rho}{2}(\|w_*\| - \|w_s\|)I_d\right)(w_s - w_*) \\
& + 2\eta\beta^2 \sum_{s=1}^{t-2} \beta^{t-2-s}(w_s - w_*)^\top \left(\eta(2\tilde{c}_w + (c_\beta - 2c_w)\omega_{t-2}^\beta)A_*^2\right)(w_s - w_*) \\
& + 2\eta^2\rho^2 R^2 \beta^2 \omega_{t-2}^\beta (c_\beta - 2c_w) \sum_{s=1}^{t-2} \beta^{t-2-s}\|w_s - w_*\|^2 \\
& - \eta\beta^2\rho \sum_{s=1}^{t-2} \beta^{t-2-s}(\|w_*\| - \|w_s\|)^2 \left(\|w_*\| + \|w_s\| - 4\eta\rho\tilde{c}_w\|w_s\|^2\right) \\
& + \eta^2(c_\beta - 2c_w)2\beta^t\omega_{t-2}^\beta\left(\|A_*\|_2^2 + \rho^2 R^2\right)\|w_0 - w_*\|^2 \\
\leq & \left(1 - \eta(1+\beta)\left[\rho\|w_t\| - (\gamma - \frac{\rho\|w_*\| - \gamma}{2})\right]\right)\|w_t - w_*\|^2 + \eta\beta c_0\|w_{t-1} - w_*\|^2 \\
& - 2\eta\beta^2 \sum_{s=1}^{t-2} \beta^{t-2-s}(w_s - w_*)^\top \left(A_* - \frac{\rho}{2}(\|w_*\| - \|w_s\|)I_d\right)(w_s - w_*) \\
& + 2\eta\beta c_1 \sum_{s=1}^{t-2} \beta^{t-2-s}(w_s - w_*)^\top (A_* + \rho R I_d)(w_s - w_*) \\
& - \eta\beta^2\rho \sum_{s=1}^{t-2} \beta^{t-2-s}(\|w_*\| - \|w_s\|)^2 \left(\|w_*\| + \|w_s\| - 4\eta\rho\tilde{c}_w\|w_s\|^2\right) \\
& + \eta^2(c_\beta - 2c_w)2\beta^t\omega_{t-2}^\beta\left(\|A_*\|_2^2 + \rho^2 R^2\right)\|w_0 - w_*\|^2,
\end{aligned}
$$
(98)

where the last inequality is due to (1): $2\eta^2\beta^2(2\tilde{c}_w + (c_\beta - 2c_w)\omega_{t-2}^\beta)A_*^2 \preceq 2\eta^2\beta^2(2\tilde{c}_w + L\beta\omega_{t-2}^\beta)A_*^2 \preceq 2\eta\beta c_1 A_*$, as $c_\beta - 2c_w \leq L\beta$ and that $\eta \leq \frac{c_1}{\beta(2\tilde{c}_w + L\beta\omega_{t-2}^\beta)\|A_*\|}$, and (2) that $2\eta^2\rho^2 R^2\beta^2\omega_{t-2}^\beta(c_\beta - 2c_w) \leq 2\eta^2\rho^2 R^2 L\beta^3\omega_{t-2}^\beta \leq 2\eta\rho\beta c_1 R$, as $\eta \leq \frac{c_1}{\rho\beta^2 L\omega_{t-2}^\beta R}$.

To continue, let us bound the last two terms on (98). For the second to the last term, by using that $\|w_t\| \leq R$ for all $t$ and that $\eta \leq \frac{1}{4\rho\tilde{c}_w R}$, we have that $\left(\|w_*\| + \|w_s\| - 4\eta\rho\tilde{c}_w\|w_s\|^2\right) \geq \|w_*\|$. Therefore, we have that the second to last term on (98) is non-positive, namely, $-\eta\beta^2\rho \sum_{s=1}^{t-2} \beta^{t-2-s}(\|w_*\| - \|w_s\|)^2\left(\|w_*\| + \|w_s\| - 4\eta\rho\tilde{c}_w\|w_s\|^2\right) \leq 0$. For the last term on (98), by using that $\eta \leq \frac{c_1\left(\|A_*\| + \rho R\right)}{L\beta^2\omega_{t-2}^\beta(\|A_*\|^2 + \rho^2 R^2)}$, we have that $\eta^2(c_\beta - 2c_w)2\beta^t\omega_{t-2}^\beta\left(\|A_*\|_2^2 + \rho^2 R^2\right)\|w_0 - w_*\|^2 \leq \eta^2 2L\beta^{t+1}\omega_{t-2}^\beta\left(\|A_*\|_2^2 + \rho^2 R^2\right)\|w_0 - w_*\|^2 \leq 2\eta\beta^{t-1}c_1\left(\|A_*\| + \rho R\right)\|w_0 - w_*\|^2$.

Combining the above results, we have that

$$
\begin{aligned}
\|w_{t+1} - w_*\|^2 \leq & \left(1 - \eta(1+\beta)\left[\rho\|w_t\| - (\gamma - \frac{\rho\|w_*\| - \gamma}{2})\right]\right)\|w_t - w_*\|^2 + \eta\beta c_0\|w_{t-1} - w_*\|^2 \\
& + 2\eta\beta c_1\left(\|A_*\| + \rho R\right) \sum_{s=0}^{t-2} \beta^{t-2-s}\|w_s - w_*\|^2 \\
& - 2\eta\beta^2 \sum_{s=1}^{t-2} \beta^{t-2-s}(w_s - w_*)^\top \left(A_* - \frac{\rho}{2}(\|w_*\| - \|w_s\|)I_d\right)(w_s - w_*).
\end{aligned}
$$
(99)

$\square$

The following lemma will be used for getting the iteration complexity from Lemma 14.

**Lemma 15.** *For a non-negative sequence $\{y_t\}_{t \geq 0}$, suppose that it satisfies*

$$y_{t+1} \leq p y_t + q y_{t-1} + r \beta \bar{y}_{t-2} + x \beta \sum_{s=\tau}^{t-2} \beta^{t-2-s} y_s + z \beta^{t-\tau+1}, \text{ for all } t \geq \tau, \tag{100}$$

*for non-negative numbers $p, q, r, x, z \geq 0$ and $\beta \in [0, 1)$, where we denote*

$$\bar{y}_t := \sum_{s=0}^{t} \beta^{t-s} y_s. \tag{101}$$

*Fix some numbers $\phi \in (0, 1)$ and $\psi > 0$. Define $\theta = \frac{-p + \sqrt{p^2 + 4(q+\phi)}}{2}$. Suppose that*

$$\beta \leq \frac{\frac{p + \sqrt{p^2 + 4(q+\phi)}}{2} \phi}{r + \phi + x} \text{ and } \beta \leq \frac{p + \sqrt{p^2 + 4(q + \phi)}}{2} - \frac{1}{\psi}.$$

*then we have that for all $t \geq \tau$,*

$$y_t \leq \Big( \frac{p + \sqrt{p^2 + 4(q + \phi)}}{2} \Big)^{t-\tau} c_{\tau,\beta}, \tag{102}$$

*where $c_{\tau,\beta}$ is an upper bound that satisfies, for any $t \leq \tau$,*

$$y_t + \theta y_{t-1} + \phi \bar{y}_{t-2} + \beta \psi z \leq c_{\tau,\beta}, \tag{103}$$

*Proof.* For a non-negative sequence $\{y_t\}_{t \geq 0}$, suppose that it satisfies

$$y_{t+1} \leq p y_t + q y_{t-1} + r \beta \bar{y}_{t-2} + x \beta \sum_{s=\tau}^{t-2} \beta^{t-2-s} y_s + z \beta^{t-\tau+1}, \text{ for all } t \geq \tau, \tag{104}$$

for non-negative numbers $p, q, r, x, z$ and $\beta \in [0, 1)$, where we denote

$$\bar{y}_t := \sum_{s=0}^{t} \beta^{t-s} y_s. \tag{105}$$

$y_{t+1} + \theta y_t + \phi \bar{y}_{t-1} + \psi z \beta^{t-\tau+2}$

$\overset{(a)}{\leq} (p + \theta) y_t + q y_{t-1} + r \beta \bar{y}_{t-2} + \phi \bar{y}_{t-1} + x \beta \sum_{s=\tau}^{t-2} \beta^{t-2-s} y_s + (\psi \beta + 1) z \beta^{t-\tau+1}$

$\overset{(b)}{=} (p + \theta) y_t + (q + \phi) y_{t-1} + (r \beta + \phi \beta) \bar{y}_{t-2} + x \beta \sum_{s=\tau}^{t-2} \beta^{t-2-s} y_s + (\psi \beta + 1) z \beta^{t-\tau+1}$

$\leq (p + \theta)(y_t + \frac{(q + \phi)}{p + \theta} y_{t-1}) + \beta(r + \phi + x) \bar{y}_{t-2} + (\psi \beta + 1) z \beta^{t-\tau+1}$

$\overset{(c)}{\leq} (p + \theta)(y_t + \theta y_{t-1}) + \beta(r + \phi + x) \bar{y}_{t-2} + (\psi \beta + 1) z \beta^{t-\tau+1}$

$\overset{(d)}{\leq} (p + \theta)(y_t + \theta y_{t-1} + \phi \bar{y}_{t-2} + \psi z \beta^{t-\tau+1}) \leq (p + \theta)^2 (y_{t-1} + \theta y_{t-2} + \phi \bar{y}_{t-3} + \psi z \beta^{t-\tau-1}) \leq \dots$

$:= (p + \theta)^{t-\tau+1} c_{\tau,\beta}.$

$$\tag{106}$$

where $(a)$ is due to the dynamics (100), $(b)$ is because $\bar{y}_{t-1} = y_{t-1} + \beta \bar{y}_{t-2}$, $(c)$ is by,

$$\frac{q + \phi}{p + \theta} \leq \theta, \tag{107}$$

and $(d)$ is by

$$\beta \leq \frac{(p + \theta)\phi}{r + \phi + x} \text{ and } \beta \leq p + \theta - \frac{1}{\psi}. \tag{108}$$

Note that (107) holds if,

$$\theta \geq \frac{-p + \sqrt{p^2 + 4(q + \phi)}}{2}. \tag{109}$$

Let us choose the minimal $\theta = \frac{-p + \sqrt{p^2 + 4(q + \phi)}}{2}$. So we have $p + \theta = \frac{p + \sqrt{p^2 + 4(q + \phi)}}{2}$, which completes the proof. $\square$

**Lemma 16.** *Assume that for all $t$, $\|w_t\| \leq R$ for some number $R$. Fix the numbers $c_0, c_1 > 0$ in Lemma 14 so that $c_1 \leq \frac{c_0}{40(\|A_*\| + \rho R)}$. Suppose that the step size $\eta$ satisfies (1) $\eta \leq \frac{c_0 \beta}{2c_\beta \|A_*\|_2^2 + 2\beta^2 L + 2c_\beta \rho^2 R^2}$, (2) $\eta \leq \frac{c_1}{\beta(2\tilde{c}_w + L\beta \omega_{t-2}^\beta)\|A_*\|}$, (3) $\eta \leq \frac{c_1}{\rho \beta^2 L \omega_{t-2}^\beta R}$, (4) $\eta \leq \frac{1}{4\rho \tilde{c}_w R}$, (5) $\eta \leq \frac{c_1(\|A_*\| + \rho R)}{L\beta^2 \omega_{t-2}^\beta(\|A_*\|^2 + \rho^2 R^2)}$, (6) $\eta \leq \frac{1+\beta}{2\rho R}$, and (7) $\eta \leq \frac{1+\beta}{4\|A_*\|_2}$ for all $t$, where $c_\beta := (2\beta^2 + 4\beta + L\beta)$, $L := \|A\|_2 + 2\rho R$, $c_w := \frac{2\beta}{1-\beta}$, $\tilde{c}_w := (1 + c_w) + (1 + c_w)^2$ and $\omega_{t-2}^\beta := \sum_{s=0}^{t-2} \beta$. Furthermore, suppose that the momentum parameter $\beta$ satisfies $\beta \leq \min\{\frac{10}{11}(1 - \eta(1+\beta)\delta), 1 - \eta(1+\beta)\delta - 0.1\}$. Fix numbers $\delta, \delta'$ so that $\frac{1}{2}(\rho\|w_*\| - \gamma) > \delta > 0$ and $\frac{c_0}{20} \geq \delta' > 0$. Assume that $\|w_t\|$ is non-decreasing. If $\rho\|w_t\| \geq \gamma - \frac{1}{2}(\rho\|w_*\| - \gamma) + \delta$ and $\rho\|w_t\| \geq \gamma - \delta'$ for some $t \geq t_*$, then we have that*

$$\|w_t - w_*\|^2 \leq \left(1 - \eta(1+\beta)\delta + \frac{2\eta\beta c_0}{1 - \eta(1+\beta)\delta}\right)^{t - t_*} c_{\beta, t_*}$$

*where $c_{\beta, t_*}$ is a number that satisfies for any $t \leq t_*$, $\|w_t - w_*\|^2 + \frac{2\eta c_0 \beta}{1 - \eta(1+\beta)\delta}\|w_{t-1} - w_*\|^2 + \eta\beta c_0 \sum_{s=0}^{t-2} \beta^{t-2-s}\|w_s - w_*\|^2 + \max\{0, \beta 20\eta \sum_{s=1}^{t_*-1} \beta^{t_*-1-s}(\frac{\gamma}{2} - \frac{\rho}{2}\|w_s\|)\|w_s - w_*\|^2\} \leq c_{\beta, t_*}$*

.

*Proof.* From Lemma 14, we have that

$$\|w_{t+1} - w_*\|^2 \leq \left(1 - \eta(1+\beta)\left[\rho\|w_t\| - (\gamma - \frac{\rho\|w_*\| - \gamma}{2})\right]\right)\|w_t - w_*\|^2 + \eta\beta c_0\|w_{t-1} - w_*\|^2$$

$$+ 2\eta\beta c_1(\|A\|_* + \rho R) \sum_{s=0}^{t-2} \beta^{t-2-s}\|w_s - w_*\|^2$$

$$- 2\eta\beta^2 \sum_{s=1}^{t-2} \beta^{t-2-s}(w_s - w_*)^\top \left(A_* - \frac{\rho}{2}(\|w_*\| - \|w_s\|)I_d\right)(w_s - w_*).$$

Using that for all $t \geq t_*$, $\rho\|w_t\| \geq \gamma - \frac{1}{2}(\rho\|w_*\| - \gamma) + \delta$, we have that

$$\|w_{t+1} - w_*\|^2 \leq \left(1 - \eta(1+\beta)\delta\right)\|w_t - w_*\|^2 + \eta\beta c_0\|w_{t-1} - w_*\|^2$$

$$+ 2\eta\beta c_1(\|A\|_* + \rho R) \sum_{s=0}^{t-2} \beta^{t-2-s}\|w_s - w_*\|^2$$

$$- 2\eta\beta^2 \sum_{s=1}^{t-2} \beta^{t-2-s}(w_s - w_*)^\top \left(A_* - \frac{\rho}{2}(\|w_*\| - \|w_s\|)I_d\right)(w_s - w_*)$$

$$= \left(1 - \eta(1+\beta)\delta\right)\|w_t - w_*\|^2 + \eta\beta c_0\|w_{t-1} - w_*\|^2 \tag{110}$$

$$+ 2\eta\beta c_1(\|A\|_* + \rho R) \sum_{s=0}^{t-2} \beta^{t-2-s}\|w_s - w_*\|^2$$

$$- 2\eta\beta^2 \sum_{s=t_*}^{t-2} \beta^{t-2-s}(w_s - w_*)^\top \left(A_* - \frac{\rho}{2}(\|w_*\| - \|w_s\|)I_d\right)(w_s - w_*)$$

$$- 2\eta\beta^2 \sum_{s=1}^{t_*-1} \beta^{t-2-s}(w_s - w_*)^\top \left(A_* - \frac{\rho}{2}(\|w_*\| - \|w_s\|)I_d\right)(w_s - w_*).$$

We bound the second to last term of (110) as follows. Note that for $s \geq t_*$, we have that $\rho\|w_s\| \geq \gamma - \delta'$ by the assumption that $\|w_t\|$ is non-decreasing. Therefore, once $\rho\|w_t\|$ exceeds any level

blow $\gamma - \delta'$, it will not fall below $\gamma - \delta'$. So we have that

$$
\begin{aligned}
A_* - \frac{\rho}{2}\big(\|w_*\| - \|w_s\|\big)I_d &\overset{(a)}{\succeq} A_* - \frac{\rho}{2}\|w_*\|I_d + \frac{\gamma - \delta'}{2}I_d \\
&\overset{(b)}{\succeq} (-\gamma + \rho\|w_*\|)I_d - \frac{\rho}{2}\|w_*\|I_d + \frac{\gamma - \delta'}{2}I_d \\
&\overset{(c)}{\succeq} -\frac{\delta'}{2}I_d
\end{aligned}
\tag{111}
$$

where (a) uses that $\rho\|w_s\| \geq \gamma - \delta'$ for some number $\delta' > 0$, (b) uses the fact that $A_* \succeq (-\gamma + \rho\|w_*\|)I_d$, and (c) uses the fact that $\rho\|w_*\| \geq \gamma$. Using the result, we can bound the second to the last term of (110) as

$$
\begin{aligned}
&- 2\eta\beta^2 \sum_{s=t_*}^{t-2} \beta^{t-2-s}(w_s - w_*)^\top \big(A_* - \frac{\rho}{2}\big(\|w_*\| - \|w_s\|\big)I_d\big)(w_s - w_*) \\
&\leq \eta\beta^2\delta' \sum_{s=t_*}^{t-2} \beta^{t-2-s}\|w_s - w_*\|^2.
\end{aligned}
\tag{112}
$$

Now let us switch to the last term of (110). Using the fact that $A_* \succeq (-\gamma + \rho\|w_*\|)I_d$, and that $\rho\|w_*\| \geq \gamma$ by the characterization of the optimizer $\|w_*\|$, we have that

$$
A_* - \frac{\rho}{2}\big(\|w_*\| - \|w_s\|\big)I_d \succeq \frac{\rho}{2}\big(\|w_*\| + \|w_s\|\big)I_d - \gamma \succeq \big(\frac{\rho}{2}\|w_s\| - \frac{\gamma}{2}\big)I_d.
\tag{113}
$$

So we can bound the last term as

$$
\begin{aligned}
&- 2\eta\beta^2 \sum_{s=1}^{t_*-1} \beta^{t-2-s}(w_s - w_*)^\top \big(A_* - \frac{\rho}{2}\big(\|w_*\| - \|w_s\|\big)I_d\big)(w_s - w_*) \\
&\leq 2\eta\beta^{t-t_*+1} \sum_{s=1}^{t_*-1} \beta^{t_*-1-s}(w_s - w_*)^\top \big(\frac{\gamma}{2} - \frac{\rho}{2}\|w_s\|\big)I_d(w_s - w_*) := 2\eta\beta^{t-t_*+1}D_{t*},
\end{aligned}
\tag{114}
$$

where we denote $D_{t*} := \sum_{s=1}^{t_*-1} \beta^{t_*-1-s}(w_s - w_*)^\top \big(\frac{\gamma}{2} - \frac{\rho}{2}\|w_s\|\big)I_d(w_s - w_*)$. Combining (110), (112), (114), we have that

$$
\begin{aligned}
\|w_{t+1} - w_*\|^2 \leq{}& \big(1 - \eta(1 + \beta)\delta\big)\|w_t - w_*\|^2 + \eta\beta c_0\|w_{t-1} - w_*\|^2 \\
&+ 2\eta\beta c_1(\|A\|_* + \rho R) \sum_{s=0}^{t-2} \beta^{t-2-s}\|w_s - w_*\|^2 \\
&+ \eta\beta^2\delta' \sum_{s=t_*}^{t-2} \beta^{t-2-s}\|w_s - w_*\|^2 + 2\eta\beta^{t-t_*+1}D_{t*}.
\end{aligned}
\tag{115}
$$

Now we are ready to use Lemma 15. Set $p, q, r, x, z$ in Lemma 15 as follows.

- $p = 1 - \eta(1 + \beta)\delta$

- $q = \eta\beta c_0$

- $r = 2\eta(\|A_*\| + \rho R)c_1$

- $x = \eta\beta\delta'$

- $z = 2\eta D_{t*}\mathbb{1}\{D_{t*} \geq 0\}$.

- $\phi = q = \eta\beta c_0$

- $\psi = 10$.

we have that

$$
\|w_t - w_*\|^2 \leq \Big( \frac{1 - \eta(1+\beta)\delta + \sqrt{(1 - \eta(1+\beta)\delta)^2 + 4(2\eta\beta c_0)}}{2} \Big)^{t-t_*} c_{\beta,t_*}
$$
$$
\leq \Big( 1 - \eta(1+\beta)\delta + \frac{2\eta\beta c_0}{1 - \eta(1+\beta)\delta} \Big)^{t-t_*} c_{\beta,t_*}
$$
(116)

where we use that $\frac{a + a\sqrt{1+4b/a^2}}{2} \leq \frac{a + a(1 + \frac{2b}{a^2})}{2} = a + \frac{b}{a}$ for number $a, b$ that satisfy $4b/a^2 \geq -1$. Now let us check the conditions of Lemma 15 to see if $\beta \leq \frac{\frac{p + \sqrt{p^2 + 4(q+\phi)}}{2}\phi}{r + \phi + x}$ and $\beta \leq \frac{p + \sqrt{p^2 + 4(q+\phi)}}{2} - \frac{1}{\psi}$. For the first condition, $\beta(r + \phi + x) \leq \frac{p + \sqrt{p^2 + 4(q+\phi)}}{2}\phi$ is equivalent to $\beta \big( 2\eta(\|A_*\| + \rho R)c_1 + \eta\beta c_0 + \eta\beta\delta' \big) \leq \frac{p + \sqrt{p^2 + 4(q+\phi)}}{2}\eta\beta c_0$, which can be satisfied by $c_1 \leq \frac{c_0}{40(\|A_*\| + \rho R)}$ and $\delta' \leq \frac{c_0}{20}$, leading to an upper bound of $\beta$, which is $\beta \leq \frac{10}{11}\frac{p + \sqrt{p^2 + 4(q+\phi)}}{2}$. Using the expression of $p, q, \phi,$ and $\psi$, it suffices to have that $\beta \leq \frac{10}{11}\big( 1 - \eta(1+\beta)\delta \big)$. On the other hand, for the second condition, $\beta \leq \frac{p + \sqrt{p^2 + 4(q+\phi)}}{2} - \frac{1}{\psi}$, by using the expression of $p, q, \phi,$ and $\psi$, it suffices to have that $\beta \leq 1 - \eta(1+\beta)\delta - 0.1$.

$\square$

## F.2 PROOF OF THEOREM 3

*Proof.* Let $\delta = c_\delta \big( \rho\|w_*\| - \gamma \big)$ for some number $c_\delta < 0.5$. Denote $c_{\eta\delta}^{converge} := 1 - \frac{2\tilde{c}_0}{1 - 2\eta\delta}$ for some number $\tilde{c}_0 > 0$. By Lemma 16, we have that

$$
\|w_t - w_*\|^2 \leq \Big( 1 - \eta(1+\beta)\delta + \frac{2\eta\beta\tilde{c}_0\delta}{1 - \eta(1+\beta)\delta} \Big)^{t-t_*} c_{\beta,t_*}
$$
$$
\leq \big( 1 - \eta\delta(1 + \beta c_{\eta\delta}^{converge}) \big)^{t-t_*} c_{\beta,t_*}
$$
(117)

where $c_{\beta,t_*}$ is a number that satisfies for any $t \leq t_*$, $\|w_t - w_*\|^2 + \frac{2\eta c_0 \beta}{1 - \eta(1+\beta)\delta}\|w_{t-1} - w_*\|^2 + \eta\beta c_0 \sum_{s=0}^{t-2}\beta^{t-2-s}\|w_s - w_*\|^2 + \max\{0, \beta 20\eta \sum_{s=1}^{t_*-1}\beta^{t_*-1-s}(\frac{\gamma}{2} - \frac{\rho}{2}\|w_s\|)\|w_s - w_*\|^2\} \leq c_{\beta,t_*}$. Note that we can obtain a trivial upper-bound $c_{\beta,t}$ for any $t$ as follows. Using that $\|w_t - w_*\|^2 \leq 4R^2$ and that $c_0 \leftarrow \tilde{c}_0\delta$ and that $\delta \leftarrow c_\delta\big( \rho\|w_*\| - \gamma \big)$, we can upper-bound the term as

$$
c_{\beta,t} \leq 4R^2 \Big( 1 + \frac{2\eta\tilde{c}_0 c_\delta\big( \rho\|w_*\| - \gamma \big)\beta}{1 - \eta(1+\beta)c_\delta\big( \rho\|w_*\| - \gamma \big)} + \eta\beta\tilde{c}_0 c_\delta\big( \rho\|w_*\| - \gamma \big)\frac{1 - \beta^t}{1 - \beta} + \beta 10\eta\max\{0, \gamma\}\frac{1 - \beta^t}{1 - \beta} \Big)
$$
$$
\leq 4R^2(1 + \eta\beta\tilde{C}_\beta)
$$
$$
:= \tilde{c}_\beta.
$$
(118)

where we define $\tilde{C}_\beta := \frac{2\tilde{c}_0 c_\delta\big( \rho\|w_*\| - \gamma \big)}{1 - \eta(1+\beta)c_\delta\big( \rho\|w_*\| - \gamma \big)} + \frac{1}{1-\beta}\big( \tilde{c}_0 c_\delta(\rho\|w_*\| - \gamma) + 10\max\{0, \gamma\} \big)$ and $\tilde{c}_\beta := 4R^2(1 + \eta\beta\tilde{C}_\beta)$.

Now denote $c^{converge} := 1 - \frac{2\tilde{c}_0}{1 - 2\eta c_\delta\big( \rho\|w_*\| - \gamma \big)}$. We have that

$$
f(w_{t_*+t}) - f(w_*) \overset{(a)}{\leq} \frac{\|A\|_2 + 2\rho R}{2}\|w_{t_*+t} - w_*\|^2
$$
$$
\overset{(b)}{\leq} \frac{\|A\|_2 + 2\rho R}{2}\tilde{c}_\beta \exp\big( -\eta c_\delta(\rho\|w_*\| - \gamma)(1 + \beta c^{converge})t \big)
$$
(119)

where (a) uses the $(\|A\|_2 + 2\rho R)$-smoothness of function $f(\cdot)$ in the region of $\{w : \|w\| \leq R\}$, and (b) uses (117), (118) and that $\delta := c_\delta\big( \rho\|w_*\| - \gamma \big)$. So we see that the number of iterations in the linear convergence regime is at most

$$
t \leq \hat{T} := \frac{1}{\eta c_\delta(\rho\|w_*\| - \gamma)(1 + \beta c^{converge})} \log\Big( \frac{(\|A\|_2 + 2\rho R)\tilde{c}_\beta}{2\epsilon} \Big).
$$
(120)

Lastly, let us check if the step size $\eta$ satisfies the constraints of Lemma 16. Recall the notations that $c_w := \frac{2\beta}{1-\beta}$, $\tilde{c}_w := (1+c_w)+(1+c_w)^2$, $c_\beta := (2\beta^2+4\beta+L\beta)$, and $L := \|A\|_2+2\rho R$. Lemma 16 has the following constraints, (1) $\eta \leq \frac{c_0\beta}{2c_\beta\|A_*\|_2^2+2\beta^2 L+2c_\beta\rho^2 R^2}$, (2) $\eta \leq \frac{c_1}{\beta(2\tilde{c}_w+L\beta/(1-\beta))\|A_*\|}$, (3) $\eta \leq \frac{c_1}{\rho\beta^2 LR/(1-\beta)}$, (4) $\eta \leq \frac{1}{4\rho\tilde{c}_w R}$, (5) $\eta \leq \frac{c_1(\|A_*\|+\rho R)}{L\beta^2(\|A_*\|^2+\rho^2 R^2)/(1-\beta)}$, (6) $\eta \leq \frac{1+\beta}{2\rho R}$, and (7) $\eta \leq \frac{1+\beta}{4\|A_*\|_2}$. For the constraints of (1), using $c_0 = \tilde{c}_0\delta$ and that $\delta = c_\delta(\rho\|w_*\| - \gamma)$, it can be rewritten as

$$\eta \leq \frac{\tilde{c}_0 c_\delta(\rho\|w_*\| - \gamma)}{2(2\beta + 4 + L)(\|A_*\|_2^2 + \rho^2 R^2) + 2\beta L}. \tag{121}$$

For the constraints of (2), using $c_1 \leq \frac{\tilde{c}_0\delta}{40(\|A_*\|+\rho R)}$ and that $\delta = c_\delta(\rho\|w_*\| - \gamma)$, it can be rewritten as

$$\eta \leq \frac{\tilde{c}_0 c_\delta(\rho\|w_*\| - \gamma)}{40\beta(\|A_*\|^2 + \rho R\|A_*\|)(2\tilde{c}_w + L\beta/(1-\beta))}. \tag{122}$$

The constraints of (3) can be written as, using $c_1 \leq \frac{\tilde{c}_0\delta}{40(\|A_*\|+\rho R)}$ and that $\delta = c_\delta(\rho\|w_*\| - \gamma)$,

$$\eta \leq \frac{\tilde{c}_0 c_\delta(\rho\|w_*\| - \gamma)}{40\rho R(\|A_*\| + \rho R)\beta^2 L/(1-\beta)}. \tag{123}$$

The constraints of (5) translates into, using $c_1 \leq \frac{\tilde{c}_0\delta}{40(\|A_*\|+\rho R)}$ and that $\delta = c_\delta(\rho\|w_*\| - \gamma)$,

$$\eta \leq \frac{\tilde{c}_0 c_\delta(\rho\|w_*\| - \gamma)}{L\beta^2(\|A_*\|^2 + \rho^2 R^2)/(1-\beta)}. \tag{124}$$

Considering all the above constraints, it suffices to let $\eta$ satisfies

$$\eta \leq \min\left(\frac{1}{4(\|A_*\| + \tilde{c}_w\rho R)}, \frac{\tilde{c}_0 c_\delta(\rho\|w_*\| - \gamma)}{C_\beta(\|A_*\| + \rho R)(1 + \|A_*\| + \rho R) + 2\beta L}\right), \tag{125}$$

where $C_\beta := \max\left(4\beta + 8 + 2L, \frac{40\beta^2 L}{1-\beta} + 80\beta\tilde{c}_w, 4\tilde{c}_w\right)$.

Note that the constraint of $\eta$ satisfies $\eta c_\delta(\rho\|w_*\| - \gamma) \leq \frac{1}{4}c_\delta \leq \frac{1}{8}$. Using this inequality, we can simplify the constraint regarding the parameter $\beta$ in Lemma 16, which leads to $\beta \in [0, 0.65]$. Consequently, we can simplify and upper bound the constants $\tilde{c}_w$, $C_\beta$ and $\tilde{C}_\beta$, which leads to the theorem statement.

Thus, we have completed the proof.

□

# G   MORE DISCUSSIONS

Recall the discussion in the main text, we showed that the iterate $w_{t+1}$ generated by HB satisfies

$$\langle w_{t+1}, u_i\rangle = (1 + \eta\lambda_i)\langle w_t, u_i\rangle + \beta(\langle w_t, u_i\rangle - \langle w_{t-1}, u_i\rangle), \tag{126}$$

which is in the form of dynamics shown in Lemma 1. Hence, one might be able to show that with the use of the momentum, the growth rate of the projection on the eigenvector $u_i$ (i.e. $|\langle w_{t+1}, u_i\rangle|$) is faster as the momentum parameter $\beta$ increases. Furthermore, the top eigenvector projection $|\langle w_{t+1}, u_1\rangle|$ is the one that grows at the fastest rate. As the result, after normalization (i.e. $\frac{w_T}{|w_T|}$), the normalized solution will converge to the top eigenvector after a few iterations $T$.

However, we know that power iteration or Lanczos method are the standard, specialized, state-of-the-art algorithms for computing the top eigenvector. It is true that HB is outperformed by these methods. But in the next subsection, we will show an implication of the acceleration result, compared to vanilla gradient descent, of top eigenvector computations.

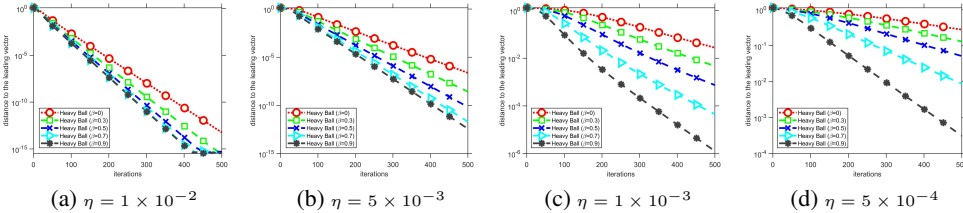

(a) $\eta = 1 \times 10^{-2}$    (b) $\eta = 5 \times 10^{-3}$    (c) $\eta = 1 \times 10^{-3}$    (d) $\eta = 5 \times 10^{-4}$

Figure 3: Distance to the top leading eigenvector vs. iteration when applying HB for solving $\min_w \frac{1}{2} w^\top A w$. The acceleration effect due to the use of momentum is more evident for small $\eta$. Here we construct the matrix $A = BB^\top \in \mathbb{R}^{10 \times 10}$ with each entry of $B \in \mathbb{R}^{10 \times 10}$ sampled from $\mathcal{N}(0, 1)$.

### G.1 IMPLICATION: ESCAPE SADDLE POINTS FASTER

Recent years there is a growing trend in designing algorithms to quickly find a second order stationary point in non-convex optimization (e.g. Carmon et al. (2018); Agarwal et al. (2017); Allen-Zhu & Li (2018); Xu et al. (2018); Ge et al. (2015); Levy (2016); Fang et al. (2019); Jin et al. (2017; 2018; 2019); Daneshmand et al. (2018); Staib et al. (2019); Wang et al. (2020)). The common assumptions are that the gradient is $L$-Lipschitz: $\|\nabla f(x) - \nabla f(y)\| \leq L\|x - y\|$ and that the Hessian is $\rho$-Lipschitz: $\|\nabla^2 f(x) - \nabla^2 f(y)\| \leq \rho\|x - y\|$, while some related works make some additional assumptions. All the related works agree that if the current iterate $w_{t_0}$ is in the region of strict saddle points, defined as that the gradient is small (i.e. $\|\nabla f(w_{t_0})\| \leq \epsilon_g$ ) but the least eigenvalue of the Hessian is strictly negative (i.e. $\lambda_{\min}(\nabla^2 f(w_{t_0})) \preceq -\epsilon_h I_d$), then the eigenvector corresponding the least eigenvalue $\lambda_{\min}(\nabla^2 f(w_{t_0}))$ is the escape direction. To elaborate, by $\rho$-Lipschitzness of the Hessian, $f(w_{t_0+t}) - f(w_{t_0}) \leq \langle \nabla f(w_{t_0}), w_{t_0+t} - w_{t_0} \rangle + \underbrace{\frac{1}{2}(w_{t_0+t} - w_{t_0})^\top \nabla^2 f(w_{t_0})(w_{t_0+t} - w_{t_0})}_{\text{exhibit negative curvature}} + \frac{\rho}{3}\|w_{t_0+t} - w_{t_0}\|^3$. So if $w_{t_0+t} - w_{t_0}$ is in the direction of the bottom eigenvector of $\nabla^2 f(w_{t_0})$, then $\frac{1}{2}(w_{t_0+t} - w_{t_0})^\top \nabla^2 f(w_{t_0})(w_{t_0+t} - w_{t_0}) \leq -c'\epsilon_h$ for some $c' > 0$. Together with the fact that the gradient is small when in the region of saddle points and the use of a sufficiently small step size can guarantee that the function value decreases sufficiently (i.e. $f(w_{t_0+t}) - f(w_{t_0}) \leq -c\epsilon_h$ for some $c > 0$). Therefore, many related works design fast algorithms by leveraging the problem structure to quickly compute the bottom eigenvector of the Hessian (see e.g. Carmon et al. (2018); Agarwal et al. (2017); Allen-Zhu & Li (2018); Xu et al. (2018)).

An interesting question is as follows "*If the Heavy Ball algorithm is used directly to solve a non-convex optimization problem, can it escape possible saddle points faster than gradient descent?*" Before answering the question, let us first conduct an experiment to see if the Heavy Ball algorithm can accelerate the process of escaping saddle points. Specifically, we consider a problem that was consider by Staib et al. (2019); Reddi et al. (2018); Wang et al. (2020) for the challenge of escaping saddle points. The problem is

$$\min_w f(w) := \frac{1}{n} \sum_{i=1}^{n} \left( \frac{1}{2} w^\top H w + x_i^\top w + \|w\|_{10}^{10} \right) \tag{127}$$

with $H := \begin{bmatrix} 1 & 0 \\ 0 & -0.1 \end{bmatrix}$. where $x_i \sim \mathcal{N}(0, \text{diag}([0.1, 0.001]))$ and the small variance in the second component will provide smaller component of gradient in the escape direction. At the origin, we have that the gradient is small but that the Hessian exhibits a negative curvature. For this problem, Wang et al. (2020) observe that SGD with momentum escapes the saddle points faster but they make strong assumptions in their analysis. We instead consider the Heavy Ball algorithm (i.e. Algorithm 1, the deterministic version of SGD with momentum). Figure 4 shows the result and we see that the higher the momentum parameter $\beta$, the faster the process of escaping the saddle points.

We are going to argue that the observation can be explained by our theoretical result that the Heavy Ball algorithm computes the top eigenvector faster than gradient descent. Let us denote $h(w) := \frac{1}{n} \sum_{i=1}^{n} \left( x_i^\top w + \|w\|_{10}^{10} \right)$. We can rewrite the objective (127) as $f(w) := \frac{1}{2} w^\top H w + h(w)$. Then,

the Heavy Ball algorithm generates the iterate according to

$$w_{t+1} = \underbrace{\begin{bmatrix} 1-\eta & 0 \\ 0 & 1+\frac{\eta}{10} \end{bmatrix}}_{:=A} w_t + \beta(w_t - w_{t-1}) - \eta\nabla h(w_t). \tag{128}$$

By setting $\eta \leq 1$, we have that the top eigenvector of $A$ is $u_1 = e_2$, which is the escape direction. Now observe the similarity (if one ignores the term $\eta\nabla h(w_t)$) between the update (128) and 19. It suggests that a similar analysis could be used for explaining the escape process. In Appendix G.2, we provide a detailed analysis of the observation. However, problem (127) has a fixed Hessian and is a synthetic objective function. *For general smooth non-convex optimization, can we have a similar explanation?* Consider applying the Heavy Ball algorithm to $\min_w f(w)$ and suppose that at time $t_0$, the iterate is in the region of strict saddle points. We have that $w_{t+1+t_0} - w_{t_0} = (I_d - \eta\nabla^2 f(w_{t_0}))(w_{t+t_0} - w_{t_0}) + \beta((w_{t+t_0} - w_{t_0}) - (w_{t+t_0-1} - w_{t_0})) + \eta(\underbrace{\nabla^2 f(w_{t_0})(w_{t+t_0} - w_{t_0}) - \nabla f(w_{t+t_0})}_{\text{deviation}})$.

By setting $\eta \leq \frac{1}{L}$ with $L$ being the smoothness constant of the problem, we have that the top eigenvector $(I_d - \eta\nabla^2 f(w_{t_0}))$ is the eigenvector that corresponds to the smallest eigenvalue of the Hessian $\nabla^2 f(w_{t_0})$, which is an escape direction. Therefore, if the deviation term can be controlled, the dynamics of the Heavy Ball algorithm can be viewed as implicitly and approximately computing the eigenvector, and hence we should expect that higher values of momentum parameter accelerate the process. To control $\nabla^2 f(w_{t_0})(w_{t+t_0} - w_{t_0}) - \nabla f(w_{t+t_0})$, one might want to exploit the $\rho$-Lipschitzness assumption of the Hessian and might need further mild assumptions, as Du et al. (2017) provide examples showing that gradient descent can take exponential time $T = \Omega(\exp(d))$ to escape saddles points. Previous work of Wang et al. (2020) makes strong assumptions to avoid the result of exponential time to escape.

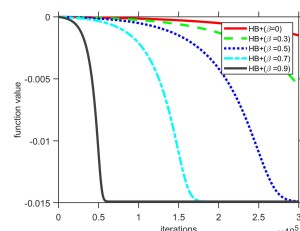

Figure 4: Solving (127) (with $n = 10$) by the Heavy Ball algorithm with different $\beta$.

On the other hand, Lee et al. (2019) show that first order methods can escape strict saddle points almost surely. So we conjecture that under additional mild conditions, if gradient descent can escape in polynomial time, then using Heavy Ball momentum ($\beta \neq 0$) will accelerate the process of escape. We leave it as a future work.

## G.2 ESCAPING SADDLE POINTS OF (127)

Recall the objective is

$$\min_w f(w) := \frac{1}{n}\sum_{i=1}^{n}\left(\frac{1}{2}w^\top Hw + x_i^\top w + \|w\|_{10}^{10}\right) \tag{129}$$

with

$$H := \begin{bmatrix} 1 & 0 \\ 0 & -0.1 \end{bmatrix}. \tag{130}$$

where $x_i \sim \mathcal{N}(0, \text{diag}([0.1, 0.001]))$ and the small variance in the second component will provide smaller component of gradient in the escape direction. At the origin, we have that the gradient is small but that the Hessian exhibits a negative curvature. Let us denote $h(w) := \frac{1}{n}\sum_{i=1}^{n}(x_i^\top w + \|w\|_{10}^{10})$. We can rewrite the objective as $f(w) := \frac{1}{2}w^\top Hw + h(w)$. Then, the Heavy Ball algorithm generates the iterate according to

$$w_{t+1} = w_t - \eta Hw_t + \beta(w_t - w_{t-1}) - \eta\nabla h(w_t)$$
$$= \begin{bmatrix} 1-\eta & 0 \\ 0 & 1+0.1\eta \end{bmatrix} w_t + \beta(w_t - w_{t-1}) - \eta\nabla h(w_t) \tag{131}$$
$$:= Aw_t + \beta(w_t - w_{t-1}) - \eta\nabla h(w_t),$$

where the matrix $A$ is defined as $A := \begin{bmatrix} 1-\eta & 0 \\ 0 & 1+0.1\eta \end{bmatrix}$. By setting $\eta \leq 1$, we have that the top eigen-vector of $A$ is $u_1 = e_2 = \begin{bmatrix} 0 \\ 1 \end{bmatrix}$, which is the escape direction. In the following, let us

denote $u_2 := e_1 = \begin{bmatrix} 1 \\ 0 \end{bmatrix}$ and denote $\bar{x} = \frac{1}{n} \sum_{i=1}^{n} x_i$. Since $\|w\|_p^p := \left( \sum_{i=1}^{d} |w_i|^p \right)$ and that $\frac{\partial \|w\|_p^p}{\partial w_j} := p|w_j|^{p-1} \frac{\partial |w_j|}{\partial w_j} = p|w_j|^{p-2} w_j$, we have that

$$\nabla h(w) = \bar{x} + 10 abs(w)^8 \circ w, \tag{132}$$

where $\circ$ denotes element-wise product and $abs(\cdot)$ denotes the absolution value of its argument in the element-wise way.

By initialization $w_0 = w_{-1} = 0$, we have that $(w_0^\top u_1) = (w_{-1}^\top u_1) = (w_0^\top u_2) = (w_{-1}^\top u_2) = 0$ and the dynamics

$$
\begin{aligned}
w_{t+1}^\top u_2 &= (1 - \eta) w_t^\top u_2 + \beta(w_t^\top u_2 - w_{t-1}^\top u_2) - \eta u_2^\top \nabla h(w_t) \\
w_{t+1}^\top u_1 &= (1 + \frac{\eta}{10}) w_t^\top u_1 + \beta(w_t^\top u_1 - w_{t-1}^\top u_1) - \eta u_1^\top \nabla h(w_t),
\end{aligned}
\tag{133}
$$

while we also have the initial condition that

$$
\begin{aligned}
w_1^\top u_2 &= -\eta u_2^\top \nabla h(w_0) = -\eta u_2^\top \nabla h(\begin{bmatrix} 0 \\ 0 \end{bmatrix}) = -\eta u_2^\top \bar{x} = -\eta \bar{x}[1] \\
w_1^\top u_1 &= -\eta u_1^\top \nabla h(w_0) = -\eta u_1^\top \nabla h(\begin{bmatrix} 0 \\ 0 \end{bmatrix}) = -\eta u_1^\top \bar{x} = -\eta \bar{x}[2].
\end{aligned}
\tag{134}
$$

That is,

$$w_1 = -\eta \bar{x}. \tag{135}$$

Using (132), we can rewrite (133) as

$$
\begin{aligned}
w_{t+1}^\top u_2 &= (1 - \eta) w_t^\top u_2 + \beta(w_t^\top u_2 - w_{t-1}^\top u_2) - \eta \bar{x}[1] - 10\eta(w_{t+1}^\top u_2)^9 \\
w_{t+1}^\top u_1 &= (1 + \frac{\eta}{10}) w_t^\top u_1 + \beta(w_t^\top u_1 - w_{t-1}^\top u_1) - \eta \bar{x}[2] - 10\eta(w_{t+1}^\top u_1)^9.
\end{aligned}
\tag{136}
$$

Note that we have that

$$
\begin{aligned}
\nabla f(w) &:= Hw + \nabla h(w) = Hw + \bar{x} + 10 abs(w)^8 \circ w \\
&= \begin{bmatrix} 1 + 10|w[1]|^8 & 0 \\ 0 & -0.1 + 10|w[2]|^8 \end{bmatrix} w + \bar{x}
\end{aligned}
\tag{137}
$$

So the stationary points of the objective function satisfy

$$
\begin{aligned}
(1 + 10|w[1]|^8)w[1] + \bar{x}[1] &= 0 \\
(-0.1 + 10|w[2]|^8)w[2] + \bar{x}[2] &= 0
\end{aligned}
\tag{138}
$$

To check if the stationary point is a local minimum, we can use the expression of the Hessian

$$
\begin{aligned}
\nabla^2 f(w) &:= H + \nabla^2 h(w) \\
&= \begin{bmatrix} 1 + 90|w[1]|^8 & 0 \\ 0 & -0.1 + 90|w[2]|^8 \end{bmatrix}.
\end{aligned}
\tag{139}
$$

Therefore, the Hessian at a stationary point is positive semi-definite as long as $|w[2]|^8 \geq \frac{1}{900}$, which can be guaranteed by some realizations of $\bar{x}[2]$ according to (138).

Now let us illustrate why higher momentum $\beta$ leads to the faster convergence in a high level way, From (138), we see that one can specify the stationary point by determining $\bar{x}[1]$ and $\bar{x}[2]$. Furthermore, from (139), once the iterate $w_t$ satisfies $|w_t[2]|^8 > \frac{1}{900}$, the iterate enters the locally strongly convex and smooth region, for which the local convergence of the Heavy Ball algorithm is known (Ghadimi et al. (2015)). W.l.o.g, let us assume that $\bar{x}[2]$ is negative. From (134), we have that $w_1^\top u_1 > 0$ when $\bar{x}[2]$ is negative. Moreover, from (136), if, before the iterate satisfies $|w_t[2]|^8 > \frac{1}{900}$, we have that $\frac{1}{10}(w_t^\top u_1) - 10(w_t^\top u_1)^9 - \bar{x}[2] > 0$, then the contribution of the projection on the escape direction (i.e. $w_{t+1}^\top u_1$) due to the momentum term $\beta(w_t^\top u_1 - w_{t-1}^\top u_1)$ is positive, which also implies that the larger $\beta$, the larger the contribution and hence the faster the growing rate of $|w_t[2]|$. Now let us check the condition, $\frac{1}{10}(w_t^\top u_1) - 10(w_t^\top u_1)^9 - \bar{x}[2] > 0$ before $|w_t[2]|^8 > \frac{1}{900}$. A sufficient condition is that $(\frac{1}{10} - \frac{1}{90})(w_t^\top u_1) - \bar{x}[2] > 0$, which we immediately see that it is true given that $\bar{x}[2]$ is negative and that $w_1^\top u_1 > 0$ and that the magnitude of $w_t^\top u_1$ is increasing. A similar reasoning can be conducted on the other coordinate $w_{t+1}[1] := w_{t+1}^\top u_2$.

# H EMPIRICAL RESULTS

## H.1 PHASE RETRIEVAL

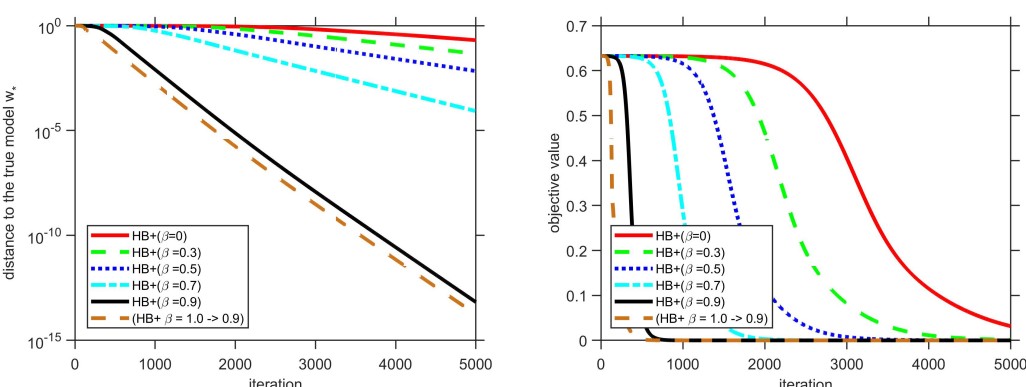

Figure 5: Performance of gradient descent with Heavy Ball momentum with different values of $\beta = \{0, 0.3, 0.5, 0.7, 0.9, 1.0 \to 0.9\}$ for solving the phase retrieval problem (1). The case of $\beta = 0$ corresponds to the standard gradient descent. Left: we plot the convergence to the true model $w_*$, defined as $\min(\|w_t - w_*\|, \|w_t + w^*\|)$, as the global sign of the objective equation 1 is unrecoverable. Right: we plot the objective value (1) vs. iteration $t$.

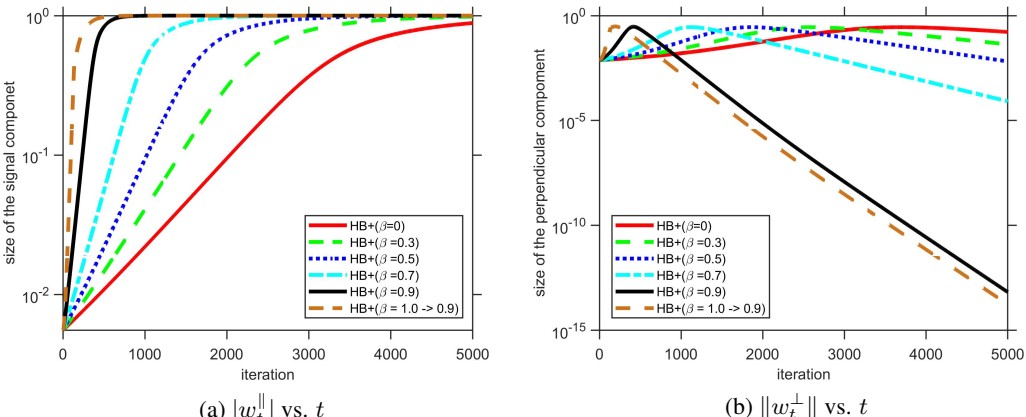

(a) $|w_t^{\|}|$ vs. $t$        (b) $\|w_t^{\perp}\|$ vs. $t$

Figure 6: Performance of HB with different $\beta = \{0, 0.3, 0.5, 0.7, 0.9, 1.0 \to 0.9\}$ for phase retrieval. (a): The size of projection of $w_t$ on $w_*$ over iterations (i.e. $|w_t^{\|}|$ vs. $t$), which is non-decreasing throughout the iterations until reaching an optimal point (here, $\|w_*\| = 1$). (b): The size of the perpendicular component over iterations (i.e. $\|w_t^{\perp}\|$ vs. $t$), which is increasing in the beginning and then it is decreasing towards zero after some point. We see that the slope of the curve corresponding to a larger momentum parameter $\beta$ is steeper than that of a smaller one, which confirms Lemma 1 and Lemma 3.

All the lines are obtained by initializing the iterate at the same point $w_0 \sim \mathcal{N}(0, \mathcal{I}_d/(10000d))$ and using the same step size $\eta = 5 \times 10^{-4}$. Here we set $w_* = e_1$ and sample $x_i \sim \mathcal{N}(0, \mathcal{I}_d)$ with dimension $d = 10$ and number of samples $n = 200$. We see that the higher the momentum parameter $\beta$, the faster the algorithm enters the linear convergence regime. For the line represented by HB+($\beta = 1.0 \to 0.9$), it means switching to the use of $\beta = 0.9$ from $\beta = 1$ after some iterations. Below, Algorithm 3 and Algorithm 4, we show two equivalent presentations of this

practice. In our experiment, for the ease of implementation, we let the criteria of the switch be $\mathbb{1}\{\frac{f(w_1)-f(w_t)}{f(w_1)} \geq 0.5\}$, i.e. if the relative change of objective value compared to the initial value has been increased to $50\%$.

---

**Algorithm 3:** Switching $\beta = 1$ to $\beta < 1$

---

1: Required: step size $\eta$ and momentum parameter $\beta \in [0, 1)$.
2: Init: $u = v = w_0 \in \mathbb{R}^d$ and $\hat{\beta} = 1$.
3: **for** $t = 0$ to $T$ **do**
4:     Update iterate $w_{t+1} = w_t - \eta\nabla f(w_t) + \hat{\beta}(u - v)$.
5:     Update auxiliary iterate $u = v$
6:     Update auxiliary iterate $v = w_t$.
7:     **If** { Criteria is met }
8:         $\hat{\beta} = \beta$.
9:         $u = v = w_{t+1}$     # (reset momentum)
10:     **end**
11: **end for**

---

---

**Algorithm 4:** Switching $\beta = 1$ to $\beta < 1$.

---

1: Required: step size $\eta$ and momentum parameter $\beta \in [0, 1)$.
2: Init: $w_0 \in \mathbb{R}^d$, $m_{-1} = 0_d$, $\hat{\beta} = 1$.
3: **for** $t = 0$ to $T$ **do**
4:     Update momentum $m_t := \hat{\beta}m_{t-1} + \nabla f(w_t)$.
5:     Update iterate $w_{t+1} := w_t - \eta m_t$.
6:     **If** { Criteria is met }
7:         $\hat{\beta} = \beta$.
8:         $m_t = 0$.     # (reset momentum)
9:     **end**
10: **end for**

---

### H.2 Cubic-regularized problem

Figure 2 shows empirical results of solving the cubic-regularized problem by Heavy Ball with different values of momentum parameter $\beta$. Subfigure (a) shows that larger momentum parameter $\beta$ results in a faster growth rate of $\|w_t\|$, which confirms Lemma 2 and shows that it enters the benign region $\mathbb{B}$ faster with larger $\beta$. Note that here we have that $\|w_*\| = 1$. It suggests that the norm is non-decreasing during the execution of the algorithm for a wide range of $\beta$ except very large $\beta$ For $\beta = 0.9$, the norm starts decreasing only after it arises above $\|w_*\|$. Subfigure (b) show that higher $\beta$ also accelerates the linear convergence, as one can see that the slope of a line that corresponds to a higher $\beta$ is steeper than that of the lower one (e.g. compared to $\beta = 0$), which verifies Theorem 3. We also observe a very interesting phenomenon: when $\beta$ is set to a very large value (e.g. 0.9 here), the pattern is intrinsically different from the smaller ones. The convergence is not monotone and its behavior (bump and overshoots when decreasing); furthermore, the norm of $\|w_t\|$ generated by the high $\beta$ is larger than $\|w_*\|$ of the minimizer at some time during the execution of the algorithm, which is different from the behavior due to using smaller values of $\beta$ (i.e. non-decreasing of the norm until the convergence). Our theoretical results cannot explain the behavior of such high $\beta$, as such value of $\beta$ exceeds the upper-threshold required by the theorem. An investigation and understanding of the observation might be needed in the future.

Now let us switch to describe the setup of the experiment. We first set step size $\eta = 0.01$, dimension $d = 4$, $\rho = \|w_*\| = \|A\|_2 = 1$, $\gamma = 0.2$ and **gap** $= 5 \times 10^{-3}$. Then we set $A = \text{diag}([-\gamma; -\gamma + \textbf{gap}; a_{33}; a_{44}])$, where the entries $a_{33}$ and $a_{44}$ are sampled uniformly random in $[-\gamma + \textbf{gap}; \|A\|_2]$. We draw $\tilde{w} = (A + \rho\|w_*\|I_d)^{-\xi}\theta$, where $\theta \sim \mathcal{N}(0; I_d)$ and $\log_2 \xi$ is uniform on $[-1, 1]$. We set $w_* = \frac{\|w_*\|}{\|\tilde{w}\|}\tilde{w}$ and $b = -(A + \rho\|w_*\|I_d)w_*$. The procedure makes $w_*$ the global minimizer of problem instance $(A, b, \rho)$. Patterns shown on this figure exhibit for other random problem instances as well.

