# OpenReview forum: "Quickly Finding a Benign Region via Heavy Ball Momentum in Non-Convex Optimization"
_ICLR.cc/2021/Conference — Reject_

### Official Review · AnonReviewer1 · 2020-10-26
**relavant contribution but clarifications are needed**

**Rating:** 6
**Confidence:** 3

**Review:**

The authors analysed the dynamics heavy ball momentum in non-convex optimization settings (namely the phase retrieval and the cubic-regularized problems). The results show that the algorithm leads to a faster convergence rate in comparison with gradient descent without inertial contributions. Although there are results in the context of convex optimization, the result in non-convex setting is novel and very interesting.

In the phase retrieval case, following an approach analogous to [Chen et al. (2018)] the authors show that the dynamics can be reduced to the study of the parallel and perpendicular components of the weights with respect to the ground truth.
However in that case, the landscape is non-convex anymore since the number of samples is infinity and the spurious minima disappear (as the authors reported in the introduction [Davis et al., 2018; Soltanolkotabi, 2014; Sun et al., 2016; White et al., 2016]).
In order to analysed the case with a finite number of datapoints, the authors introduced perturbation terms in the dynamical equations and hypothesized that the perturbation is bounded by a constant c_n that the they ask to be smaller than zeta.
My main concerns on this part are the following :
* is the landscape still convex under this constraint?
* is there the possibility that for some regimes for which the hypothesis break and heavy ball momentum will behave poorly compared to gradient descent? In the paper it seems that it always speed up the performances, but this is not clear from the theorem.

The authors in thm.1 and 2, give a characterization of the speed up given by the momentum parameter beta. The authors should check whether the bound is strict in simulations, and verify numerically that in fact the speed is correctly captured by the theory.

**Other comments :**

* There is a change of notation between the one used for Fig.1 and the one reported in the appendix (d<-m, n<-m). Furthermore those details are important and should put forward in the caption.
* The figures need larger fonts. It is almost impossible read them in the printed paper.

---

> ### Author Response · Authors · 2020-11-19
> **Response to Reviewer 1**
>
> We thank the reviewer for the positive review and the feedback.
>
> We first want to be clear that even if there are an infinite number of samples, the optimization landscape is still non-convex, since there are two isolated global minimum $\pm w_*$, and at least a local maximum, and a saddle point as Chen et al. 2019 show.
>
> In the finite-sample case, the goal is still to get into the neighborhood of $\pm w_*$. The approximated dynamic is used for analyzing the first stage, i.e. before reaching a neighborhood. We show that, with an isotropic random initialization, as long as the approximated dynamic closely follows the population dynamic, Heavy Ball will get into the neighborhood of $w_*$ or $-w_*$ faster than vanilla gradient descent. This implies that Heavy Ball can avoid getting trapped into stationary points (e.g. saddle points), as long as the deviation from the population dynamic of Heavy Ball is small. A similar result of vanilla gradient descent was established by Chen et al. 2019, which suggests that given enough samples, the approximate dynamic closely follows the population dynamic with high probability. Therefore, GD with random initialization avoids getting trapped into those undesired stationary points with high probability.
>
> One can ensure that the deviation of the population dynamic is small with high probability if there are a sufficient number of samples, as the variance of the finite-sample gradient decays when the number of samples increases. Specifically, a careful treatment of analyzing the required number of samples is possible by following the techniques of (Chen et al. 2019), but is not in the scope of this paper. As for the second stage in the finite-sample regime, if the number of samples is sufficiently large, $n \asymp d \log d$, the local strong convexity in the neighborhood of $\pm w_*$ is still preserved (Ma et al. 2017) with high probability, and one can use prior results of HB (e.g. Saunders 2018. Xiong et al. 2020) in the second stage.
>
> On the other hand, if the deviation is large or the number of samples is not enough, then the guarantee might be broken and we cannot say too much about how Heavy Ball will behave.
>
> Our result shows that momentum can help to reduce the number of iterations (of getting the benign region) required by gradient descent by a factor that can be as large as approximately $1+\beta$. The simulation supports the theoretical result. However, we know that the theoretical result is for the worst-case guarantee and the empirical performance can be better.
>
> As for other comments of the reviewer, we have followed the suggestion and updated the paper accordingly. We thank the reviewer for thoroughly reading the paper. We find the comments very helpful.

---

### Official Review · AnonReviewer2 · 2020-10-28
**Interesting analysis showing constant-factor improvements with momentum.**

**Rating:** 7
**Confidence:** 3

**Review:**

This paper analyzes Polyak momentum in the deterministic case for two simple but important non-convex problems: phase retrieval and finding the cubic-regularized newton step. It is shown that in both cases the problems posses a “benign region” in which the objective “looks” a bit convex. In the phase retrieval case, assuming that the data is generated from a normal distribution, it is shown that having a non-zero momentum parameter in fact allows the momentum method to find the benign region faster, after which standard convex analysis can take over to find the optimal point.

For the cubic regularization problem, a similar result showing faster convergence to the benign region is shown. The key steps appear to be observing that in both instances, the benign region is essentially just asking for the norm of the iterate to reach sufficient magnitude. Then, a technical lemma (Lemma 1) and careful analysis of the gradients of the individual losses shows that the iterates increase in magnitude very quickly, with the momentum parameter accelerating the increase.

Overall I think these are interesting results. I think the major weakness (as admitted by the authors) is that for the cubic-regularization problem, there is no guarantee that the iterates will in fact stay in the benign region once they have entered it. Although it seems plausible that such behavior is in fact guaranteed, this does somewhat weaken the theoretical impact. The other issue is that as far as I can see, these results improve by at most a constant factor of about 2 over the number of iterations required by ordinary gradient descent, which is not a huge amount. However, constants are important in practice, and many analyses of Polyak momentum in the non-convex setting actually get worse when there is momentum, so at least this is going in the right direction :)

I think the cubic-regularization result seems the most promising for future development as it may have some application to generally showing faster a convergence on second-order-smooth non-convex problems.

Minor nit:
In Lemma 1, what does “the effective dynamics satisfies...” mean? It sounds cool, but I don’t actually know what effective dynamics are. Maybe you could just say “a_t satisfies...”. Similarly for the other few places that effective dynamics shows up.

---

> ### Author Response · Authors · 2020-11-19
> **Response to Reviewer 2**
>
> We thank the reviewer for the encouraging feedback and the positive review. We also appreciate the reviewer for neatly summarizing our paper.
>
> ("effective dynamics"): We thank the reviewer for the suggestion and we have updated the paper accordingly.

---

### Official Review · AnonReviewer3 · 2020-10-29
**Summary**

**Rating:** 4
**Confidence:** 4

**Review:**

#########################
Summary:
This paper studies heavy ball momentum in non-convex optimization. It has provided some theory to explain the good performance of heavy ball method in some type of nonconvex problems, i.e., the phase retrieval problem and cubic-regularized problem.

#########################
Reasons for score:
Overall, I vote for rejecting. Although this paper has some contribution to explain the superiority of heavy ball method over vanilla SGD, the problem has very special structure, which makes its not very much likely to be generalized to general problems and applications.

##########################
Pros:
1. This paper has proven the superiority of heavy ball method in some cases by showing that heavy ball momentum helps the iterate to enter a benign region that contains a global optimal point faster.


##########################
Cons:
1. The considered problems have very special structures, i.e., the phase retrieval problem and cubic-regularized problem, which would prevent the generalization to a broad family of problems. Even though that the authors have tried to generalize it to solve the problem of top eigenvector computation and saddle point escape problem, they still focus on special subproblems of these two.

2. In terms of writing, the authors did not give enough background of the considered phase retrieval problem and cubic-regularized problem. I do not feel that the ICLR community would be familiar with these two problems.

##########################
Questions:
1. Why are considered problem important on their own? And how can they be connected to problems that the ICLR community would be interested in?

---

> ### Author Response · Authors · 2020-11-19
> **Response to Reviewer 3**
>
> We thank the reviewer for the feedback.
>
> Our work falls into a broad paradigm of understanding momentum, e.g. (Kidambi et al. ICLR 2019) and (Liu and Belkin ICLR 2020). Theoretically understanding when and why Heavy Ball momentum works in non-convex optimization is a big open question in the community. In our paper, we identify a common dynamic and build a theoretical result of Heavy Ball momentum for solving a family of non-convex problems. We think our result is exciting and other reviewers seem to agree that our result is interesting and novel.
>
> We want to point out that in the convex world, strongly convex, smooth, twice differentiable are the only known example such that Heavy Ball has provable
> acceleration. There are even some negative examples of divergence in convex optimization (see e.g. Lessard et al. 2016). Similarly, in the non-convex world, there is little theoretical evidence supporting the use of Heavy Ball momentum. As Reviewer 2 points out, many analyses of Heavy Ball momentum in non-convex optimization actually get worse when there is momentum. Given the current progress of the literature, generalizing our results beyond the family of problems which we considered might be too much for this paper.
>
>
> Furthermore, we note that (Kidambi et al. ICLR 2019) and (Liu and Belkin. ICLR 2020) only focuses on the classical linear regression $F(w) = \frac{1}{2} \sum_i (w^\top x_i - y_i)^2$ and/or convex optimization.  But that doesn't necessarily make their papers weak.
>
> Ref:
>
> [1] Rahul Kidambi, Praneeth Netrapalli, Prateek Jain, and Sham M. Kakade. On the insufficiency of existing momentum schemes for stochastic optimization. ICLR 2019
>
> [2] Chaoyue Liu, Mikhail Belkin
> Accelerating SGD with momentum for over-parameterized learning
> ICLR 2020
>
> *** phase retrieval ***
>
> We briefly mentioned the applications of phase retrieval in the introduction.
> On page 1 of the paper, we also cite two tutorial papers (Fannjiang and Strohmer, 2020) and (Shechtman et al., 2015) about phase retrieval.
>
> We note that there are some efforts in integrating the technique of generative models and phase retrieval, which could help the task of image recovery (e.g. Hand et al. NeruIPS 2019).
>
> Furthermore, phase retrieval might also be a good entry point of understanding some observations in optimization and neural net training (e.g. Mannellia et al. NeurIPS 2020).
>
>
> Ref:
>
> [3] Fannjiang and Strohmer. The Numerics of Phase Retrieval.
>  arXiv:2004.05788. 2020
>
> [4] Phase Retrieval Under a Generative Prior. Paul Hand and Oscar Leong and and Vladislav Voroninsk. NeurIPS 2019.
>
> [5] Stefano Sarao Mannellia, Giulio Birolib, Chiara Cammarotac, Florent Krzakalab, Pierfrancesco Urbania, and Lenka Zdeborova. Complex dynamics in simple neural networks: Understanding gradient flow in phase retrieval.
> NeuriPS 2020.
>
> *** cubic-regularized problem ***
>
> As we mentioned in the paper, solving the cubic-regularized problem is a critical subroutine of Nesterov-Polyak's algorithm of finding a second-order stationary point in general smooth non-convex problems.
>
> It is known that for solving some non-convex optimization problems, it suffices to find a second-order stationary point. These problems include dictionary learning (Sun et al. 2015), matrix completion (Chi et al. 2019), robust PCA (Ge et al. 2017) etc.
>
> Furthermore, there are some recent works at ICLR highlighting the importance of quickly finding a second-order stationary point for learning a neural network.
>
> (Ge et al. ICLR 2019) consider learning a class of one-layer neural net where the optimization landscape has no spurious local minima and finding an approximately second-order stationary point is equivalent to finding an approximately global minimum.
>
> (Bai and Lee ICLR 2020) consider training a neural network such that "the randomized neural net loss exhibits a nice optimization landscape in that every second-order stationary point has training loss not much higher than the best quadratic model''.
>
> Ref:
>
> [6] Rong Ge, Tengyu Ma, and Jason D. Lee. Learning One-hidden-layer Neural Networks with Landscape Design. ICLR 2019.
>
> [7] Yu Bai and Jason D. Lee. Beyond Linearization: On Quadratic and Higher-Order Approximation of Wide Neural Networks. ICLR 2020.
>
> [8] Yuejie Chi and Yue M. Lu and Yuxin Chen. Nonconvex Optimization Meets Low-Rank Matrix Factorization: An Overview. 2019.
>
> [9] Rong Ge and Chi Jin and Yi Zheng. No Spurious Local Minima in Nonconvex Low Rank Problems: A Unified Geometric Analysis. ICML 2017
>
> [10] Ju Sun and Qing Qu and John Wright. Complete Dictionary Recovery over the Sphere. arXiv:1504.06785. 2015
>
> *** " the authors did not give enough background...''  ***
>
> We note that other reviewers do not have such concern.  It would be more helpful if the reviewer can be more explicit on what is lacking, and we will add.

---

### Official Review · AnonReviewer4 · 2020-10-29
**Interesting work but the results are not very clear**

**Rating:** 6
**Confidence:** 4

**Review:**

This paper studies the deterministic Heavy Ball (HB) method compared with vanilla gradient descent (GD) in nonconvex optimization.
In particular, they consider two optimization problems, i.e., phase retrieval and cubic-regularized problem, and try to show HB is better than GD, and the larger momentum parameter $\beta$, the better benefit of HB.

They consider two stages for these two problems. The first stage is to find/fall into a benign region (strongly-convex region) and then the second stage can directly use previous HB results for strongly convex functions (i.e., linear convergence).
Thus the main part/contribution is to show how HB could perform better than GD in the first stage. In this paper, they show this by providing Theorem 1 and 2 for these two problems respectively.
However, I have some questions/concerns regarding Theorem 1 and 2.

For Theorem 1 (phase retrieval):
1a. They first analyze the HB using the population update (i.e., Eq (5)), then they consider Eq (11) for dealing with finite samples case. The author should show/prove that Eq (11) is the update form for finite case and also satisfies the corresponding assumption in Theorem 1.
1b. Why the initial point $\||w_0\||$ is not equal to 1 in Theorem 1, i.e., sampling from the unit sphere? Since they assume that $\||w_*\||=1$.
1c. They claim that HB is better than GD by a factor of $1+c_\eta \beta$ according to Theorem 1. But what’s the result for GD, is it $T_\zeta \lesssim\log d$? It looks a bit strange as it does not depend on the step-size $\eta$.

For Theorem 2 (cubic regularized)
2a. In general, the value of step-size $\eta$ varies for different algorithms. So only according to their Theorem 2, I am not convinced that HB is better than GD as the authors claimed since GD may choose different step-size due to nonexistence of momentum.
2b. Beyond Theorem 2, I have a question regarding Figure 2b. Why there are oscillations for the last two curves in Figure 2b?

---

> ### Author Response · Authors · 2020-11-19
> **Response to Reviewer 4**
>
> We thank the reviewer for the positive review and the feedback. We also thank the reviewer for thoroughly reading the paper and we believe the comments help to improve the presentation of this paper.
>
> *** === 1 (a) === ***
>
> We thank the reviewer for the suggestion. We propose to use the approximated dynamic (11) for modeling the case when only a finite number of samples are available. When $\beta=0$, we note that the approximated dynamic (11) reduces to the one used by (Chen et al. 2019) for analyzing vanilla gradient descent in the finite-sample regime. Therefore, we think it is reasonable to use (11) for modeling and analyzing the case when only a finite number of samples are available. As long as there are enough samples, the approximated dynamic should closely follow the population dynamic with high probability and careful treatment of analyzing the required number of samples is possible by following the techniques of (Chen et al. 2019). However, as we mentioned in the introduction, we are not aiming at providing a state-of-the-art algorithm for solving phase retrieval. We treat phase retrieval as one of the starting points of understanding Heavy Ball momentum in non-convex optimization and were be able to identify an accelerated mechanism of Heavy Ball for a family of non-convex problems. We leave the analysis of the sample complexity as a potential future work.
>
> *** === 1 (b) === ***
>
> In Remark 1, we provide a discussion of how to ensure the initial point $w_0$ to satisfy $|w_0^\parallel| \gtrsim \frac{1}{ \sqrt{d \log d} }$ and $\\| w_0 \\| < \frac{1}{3}$, which are the conditions required by the theorem. The conditions state that the initial point has a non-zero projection on $w_*$, and that the size of the initial point is (sufficiently) smaller than $\\| w_* \\|$ so that the (undesired) perpendicular component is also sufficiently small initially.
>
> The first condition can be satisfied by sampling $w_0$ from a gaussian or sampling uniformly from a sphere with high probability. The second condition can then be satisfied by scaling the size appropriately.
>
> We follow (Chen et al. 2019) to analyze the case that $\\| w_* \\| = 1$ and we want to emphasize that this is without loss of generality. Our analysis can be extended to other cases when $\\| w_* \\|$ is a different number.
>
>
> We thank the reviewer for raising this concern and we have polished the writing of Remark 1.
>
> *** === 1(c) === ***
>
> The notation $\lesssim$ hides the step size $\eta$. We were following the presentation of (Chen et al. 2019), which states that vanilla gradient descent takes $T_{\zeta} \lesssim \log d$ number of iterations for getting in a benign region. Their presentation hides the dependency on $\eta$ (i.e. $T_{\zeta} \lesssim \frac{\log d}{\eta}$ of GD).
> But we don't have to stick to their presentation. We have updated our presentation and have made the dependency on $\eta$ explicit, i.e. $T_{\zeta} \lesssim \frac{\log d}{ \eta \left(1 + c_{\eta} \beta \right)} $.
>
> We thank the reviewer for raising this point.
>
> *** === 2(a) === ***
>
> (Carmon and Duchi 2019) show that vanilla gradient descent with the step size
> satisfying
>
> $
> \eta \leq
> \frac{1}{4 (  \rho \\| w_* \\| + \\| A \\|_2   ) }
> $
>
> needs
>
> $
> T_{\delta} \leq
> \frac{2}{ \eta \delta  } \log \big( 1 +
> \frac{ \gamma_+^2   }{4 \rho | b^{(1)} |} \big).
> $
>
> number of iterations required to get into the benign region.
>
> On other other hand, we show that for Heavy Ball with the step size satisfying (a larger range of)
>
> $
> \eta \leq \frac{1}{  \rho \\|w_* \\| + \\| A \\|_2  },
> $
>
> the number of iterations required to get into the benign region is
>
> $
> T_{\delta} \leq
> \frac{2}{ \eta \delta ( 1 + \beta / ( 1+ \eta \delta) ) } \log \big( 1 +
> \frac{ \gamma_+^2  (1 + \beta/ (1+\eta \delta))  }{4 \rho | b^{(1)} |} \big).
> $
>
> Simply ignoring the log factor on both bounds, we can see that Heavy Ball momentum reduces the number of iterations by a factor that can be as large as approximately $(1+\beta)$.
>
> Furthermore, Lemma 1 in our paper should also show a strict separateness of the progress between the case under a non-zero momentum parameter $\beta$ and the case without momentum.
>
>
> *** === 2(b) === ***
>
> In convex optimization, it is known that under certain tuning of the momentum parameter $\beta$, the convergence can be non-monotone (e.g. Flammarion and Bach 2015; Saunder 2019).
>
> Regarding the observation (bump and overshoots), we conjecture that a similar behavior/mechanism of using a certain value of the momentum parameter happens when the iterate is in the benign region of the cubic-regularized problem. Understanding the oscillatory behavior in the second phase is important but it is not in the scope of this paper.
>
> Ref:
>
> Nicolas Flammarion, Francis Bach. From Averaging to Acceleration, There is Only a Step-size. COLT, 2015.
>
> Michael Saunders. Notes on first-order methods for minimizing smooth functions. Lecture note. 2019.

---

### Decision · Program_Chairs · 2021-01-07
**Final Decision**

**Decision:**

Reject

**Comment:**

This paper considers the analysis of momentum for nonconvex problems. While this is a worthy direction, as some reviewers pointed out, the examples considered are rather specialized and one can argue they are mostly "convex-like". Therefore it is not clear that these results are generalizable and whether the analysis offers insights about how momentum helps (including faster escape of saddle in nonconvex regime, and faster convergence in the convex regime). In the current form, these results have limited scopes.